# Causal Dynamic Variational Autoencoder for Counterfactual Regression in Longitudinal Data

**Mouad El Bouchattaoui[1,2]**                                   *mouad.elbouchattaoui@gmail.com*
**Myriam Tami[1]**
**Benoit Lepetit[2]**
**Paul-Henry Cournède[1]**
[1]*Paris-Saclay University, CentraleSupélec, MICS Lab, Gif-sur-Yvette, France*
[2]*Saint-Gobain, France*

**Reviewed on OpenReview:** *https://openreview.net/forum?id=atf9q49DeF*

## Abstract

Accurately estimating treatment effects over time is crucial in fields such as precision medicine, epidemiology, economics, and marketing. Many current methods for estimating treatment effects over time assume that all confounders are observed or attempt to infer unobserved ones. In contrast, our approach focuses on unobserved adjustment variables—variables that specifically have a causal effect on the outcome sequence. Under the assumption of unconfoundedness, we address estimating Conditional Average Treatment Effects (CATEs) while accounting for unobserved heterogeneity in response to treatment due to these unobserved adjustment variables. Our proposed Causal Dynamic Variational Autoencoder (CDVAE) is grounded in theoretical guarantees concerning the validity of latent adjustment variables and generalization bounds on CATEs estimation error. Extensive evaluations on synthetic and real-world datasets show that CDVAE outperforms existing baselines. Moreover, we demonstrate that state-of-the-art models significantly improve their CATE estimates when augmented with the latent substitutes learned by CDVAE—approaching oracle-level performance without direct access to the true adjustment variables. [1]

## 1 Introduction

Estimating *Conditional Average Treatment Effects (CATEs)* helps us understand how individuals uniquely respond to the same treatment, thereby enabling more personalized and effective decision-making. For example, in healthcare, two patients receiving the same drug might experience considerably different outcomes due to underlying genetic or lifestyle differences (Atan et al., 2018; Shalit, 2020; Mueller & Pearl, 2023). The exact curriculum might yield more remarkable improvement for one student than another in education, depending on background factors like socioeconomic status (Morgan, 2013; Imbens & Rubin, 2015). Likewise, in marketing, identical promotions might drive purchases in one customer segment but not another (Hair Jr & Sarstedt, 2021; Fang et al., 2023).

*Longitudinal data* arise naturally in these domains. Consider, for instance, a medical dataset recording blood pressure, treatments (e.g., vasopressors), and vitals for each patient at regular intervals. Or a retail dataset tracking weekly customer purchases following commercial campaigns. Each of these longitudinal data describes a sequence of treatments, covariates, and responses causally interacting through time. However, this setting brings unique challenges: (1) *Time-dependent confounding*: Confounders influenced by past treatment can impact subsequent treatments and responses (Platt et al., 2009); (2) *Selection bias*: Time-varying covariates exhibit imbalanced distributions across treatment regimes which should be accounted for to estimate treatment effects accurately (Robins et al., 2000; Schisterman et al., 2009; Lim, 2018);

---

[1]The implementation is available at `https://github.com/moad-lihoconf/cdvae`

(3) *Long-term dependencies*: A treatment effect may unfold over extended periods, requiring models to capture complex long-range interactions between covariates, treatments, and responses (Choi et al., 2016; Pham et al., 2017); (4) *Missing covariates*: Some variables crucial for predicting outcomes—like genetic predispositions or environmental exposures—introduce bias and less personalized treatment effect estimation unless properly accounted for.

**Assumptions over Confounders and Existing Approaches**  The existing literature primarily addresses the first three challenges under the assumption of *sequential ignorability*, where all confounders, whether static or time-varying, are fully observed. Methods such as Marginal Structural Models (MSMs) (Robins & Hernán, 2009a), Recurrent Marginal Structural Models (RMSM) (Lim, 2018), Counterfactual Recurrent Networks (CRN) (Bica et al., 2020a), G-Net (Li et al., 2021), Causal Transformers (CT) (Melnychuk et al., 2022), and Causal CPC (Bouchattaoui et al., 2024) have been developed based on this assumption. When the fourth challenge—missing covariates—is addressed, it is often in the form of *missing confounders*, leading to the violation of sequential ignorability. Existing approaches either condition on *observed proxies* of confounders to infer a latent representation of the unobserved confounders (Kuroki & Pearl, 2014; Miao et al., 2016; Louizos et al., 2017; Cheng et al., 2021) or apply the *deconfounding technique*, which involves imposing a factor model over the treatment assignment to mitigate hidden confounding (Lopez & Gutman, 2017; Ranganath & Perotte, 2018; Wang & Blei, 2019a; Zhang et al., 2019; Bica et al., 2020b; Hatt & Feuerriegel, 2024).

**Our Focus**  In contrast to missing confounders, we study for the first time the presence of *unobserved static adjustment variables*—factors that affect only the outcome sequence and remain *time-invariant*. In a medical context, these could include genetic factors, environmental conditions, or lifestyle attributes that impact treatment response but are not directly observed (Sadowski et al., 2024). The absence of such variables can lead to a loss of heterogeneity in the estimated treatment effect. This phenomenon can also be understood through the lens of *population structure*, which arises from distinct subgroups within a population that share characteristics such as geography, socioeconomic status, or cultural practices. Having knowledge of, or being able to accurately infer, population structure allows for a more precise estimation of CATEs by capturing variations in treatment responses across subgroups. In genomics (Laird & Lange, 2011; Peter et al., 2020), for instance, such structure arises due to evolutionary or migration histories. This concept extends beyond genetics to fields such as economics, healthcare, and education, where differences among subgroups stem from factors such as socioeconomic conditions or environmental influences. Moreover, population structure often acts as an *effect modifier* (Hernán & Robins, 2020), altering treatment effects across subgroups without directly affecting treatment assignment (Hyun et al., 2024). By accounting for such effect modifiers through stratified or interaction analyses, treatment effects can be estimated more accurately, even in randomized trials (Schochet, 2024).

**Adjustment Variables vs. Confounders.**  We distinguish *confounders*, pre-treatment variables that affect both treatment and outcome, whose omission biases estimates, from *adjustment variables*, pre-treatment variables that affect the outcome and may modify treatment effects but not treatment assignment (Hernán MA, 2020).  Adjusting for the latter clarifies further treatment-effect heterogeneity, even when confounding variables are fully observed. For example, in healthcare, pharmacogenetic variants (VKORC1, CYP2C9) markedly alter warfarin (a blood thinner) response yet typically do not determine treatment assignment. Modeling these variants (observed or represented) improves precision and personalization of effects (McClain et al., 2008; Schwarz et al., 2008). In education, Baseline test scores and socioeconomic indicators are important adjustment variables and often are effect modifiers. Covariate adjustment in school-level RCTs thus substantially increases precision and clarifies subgroup treatment effects (Bloom et al., 2007; Steingrimsson et al., 2017). In online experiments/ marketing, pre-period behavior (preceding the A/B test) consists of important adjustment variables that do not affect the randomized assignment, yet adjusting for them leads to faster detection of heterogeneity in treatment effect and even a reduction of the required sample size to achieve a given statistical power (Deng et al., 2013; Benkeser et al., 2021).

In this work, we specifically address the challenge of missing covariates by focusing on estimating the *contemporaneous treatment effect*—the effect of the current treatment on the subsequent response, given the history of confounding processes (see Remark 1). We consider the estimation of an *Augmented Conditional Average*

*Treatment Effect (ACATE)* that depends not only on the confounding variables but also on adjustment variables. We aim to achieve *near-oracle performance* in treatment effect estimation by leveraging the learned representation of unobserved adjustment variables. Specifically, we prove that for all baselines satisfying sequential ignorability, treatment effect estimation is substantially improved when these models are augmented with the learned representation of unobserved adjustment variables produced by our model—ultimately delivering performance that approaches the oracle scenario, where the true unobserved adjustment variables are directly fed into the baselines. Extensive experiments on synthetic data and semi-synthetic data derived from real-world datasets such as MIMIC-III (Johnson et al., 2016) validate our approach.

**Our Approach**  We treat the unobserved adjustment variables as latent variables and leverage a probabilistic modeling approach based on the *Dynamic Variational Autoencoder (DVAE)* framework (Girin et al., 2021) to estimate the ACATE, where we augment the CATE with the learned representation of unobserved adjustment variables, which we term *substitutes*. We address the covariate imbalance induced by selection bias in longitudinal data using a weighted empirical risk minimization strategy, where the weights are a function of the propensity scores. Additionally, we derive a generalization bound for the error in estimating ACATE and use its approximation as a loss function for our model, which we refer to as *Causal DVAE (CDVAE)*. To account for potential population structure induced by unobserved adjustment variables, we define a flexible prior over the latent variables using a learnable Gaussian Mixture Model. We ensure the causal validity of the learned substitutes by imposing a finite-order Conditional Markov Model (CMM) on the response series. The causal validity follows from our proof that, once the finite-order CMM holds, the learned substitutes account for all relevant adjustment variables (Theorem 6) and that the ACATE leveraging the substitutes is identifiable. To experimentally support the theoretical analysis, we discuss the relevance of all CDVAE components through an ablation study.

Furthermore, since we adopt a probabilistic approach, we represent the substitute for unobserved adjustment variables as a *stochastic* latent variable. Consequently, the choice of the substitute given individual longitudinal data is not unique. As a novel contribution, we study Causal DVAE in the *near-deterministic regime*, where we reduce the variance of the responses (outputs of the Causal DVAE) to near zero. This process indirectly pushes the covariance matrix of the substitutes toward zero, preventing their distribution from becoming overly diffuse. An intuitive consequence of this approach is that any sampled substitute of the adjustment variables yields the same treatment effect as the mean substitute. We demonstrate that as the response variance approaches zero, the treatment effect remains consistent regardless of the specific instance of the substitutes. The near-deterministic regime of VAEs has not been previously explored in the context of causal inference, making this a novel contribution. To draw an analogy, in the deconfounding literature (Lopez & Gutman, 2017; Ranganath & Perotte, 2018; Wang & Blei, 2019a; Zhang et al., 2019; Bica et al., 2020b; Hatt & Feuerriegel, 2024), substitutes—referring to learned representations of missing confounders—are theoretically assumed to follow a Dirac posterior. This enables deterministic selection and facilitates treatment effect identifiability. However, the training of these models is based on a non-degenerate posterior, creating a disconnect between theoretical assumptions and practical implementation. Our approach bridges this gap by achieving near-deterministic behavior while maintaining a probabilistic framework.

**Contributions**  Our contributions can be summarized as follows: (1) We propose a principled approach to infer a latent representation of unobserved adjustment variables (§ 4).(2) We prove the theoretical validity of the learned adjustment variables by modeling the response series as a finite-order conditional Markov process, ensuring that ACATEs conditioned on learned substitutes are identifiable (§ 4.1). (3) We study for the first time the near-deterministic regime of VAEs for causal inference and show that reducing the variance of response predictions enforces consistency in treatment effect estimation under stochastic modeling of adjustment variables (§ 4.3). (4) Numerical experiments confirm the effectiveness of CDVAE in estimating ACATEs, showing that our method consistently outperforms state-of-the-art (SOTA) baselines (§ 5), as further analyzed through an ablation study. (5) We show that for SOTA baselines, augmenting their input covariates with the inferred representation of adjustment variables given by CDVAE substantially enhances their accuracy in estimating ACATEs (§ 5) and even provides near-oracle performances.

## 2 Related Work

**Causal Inference in Time-Varying Settings**  Assuming sequential ignorability, MSMs (Robins et al., 2000) are widely used to adjust for time-varying confounders in causal inference, employing inverse probability of treatment weighting (Robins & Hernán, 2009a) to balance covariates by estimating treatment probabilities based on past treatment and confounders. However, MSMs often yield high-variance estimates, and their effectiveness heavily relies on the accurate specification of the treatment assignment mechanism—a challenge in complex, high-dimensional settings. Recent neural network-based models address some of these challenges. The RMSM (Lim, 2018) uses Recurrent Neural Networks (RNNs) in propensity and outcome modeling for multi-step outcome forecasting and outperforms traditional MSMs. Furthermore, the CRN (Bica et al., 2020a) uses adversarial domain training to learn a treatment-invariant representation space to mitigate bias from time-varying confounders. G-Net (Li et al., 2021) combines g-computation (Vansteelandt & Joffe, 2014) with sequential deep models for multi-timestep counterfactual prediction. Causal Transformer (Melnychuk et al., 2022) employs self-attention to capture temporal dynamics and to learn treatment-invariant representations using an adversarial approach, similar to Bica et al. (2020a). G-Transformer (Xiong et al., 2024) extends g-computation by using a Transformer to model covariate dynamics and Monte-Carlo rollouts for counterfactuals under dynamic regimes and is thus conceptually a combination of Causal Transformer and G-Net. Additionally, Causal CPC (Bouchattaoui et al., 2024) leverages temporal dynamics through contrastive predictive coding and information maximization and learns a balanced representation using an adversarial training strategy. Our approach enhances these SOTA baselines by augmenting their input covariates with our inferred substitute of adjustment variables, achieving a more accurate estimation of ACATEs.

**Combining Weighting and Representation Learning**  Representation learning in causal inference helps correct covariate imbalance in high-dimensional settings by balancing learned features and reducing covariate shift. When combined with weighting methods (Zubizarreta, 2015; Li et al., 2018; Johansson et al., 2018; Hassanpour & Greiner, 2019b), these approaches become more effective—showing that carefully chosen weights can improve causal estimation. However, there is a trade-off between covariate balance and predictive performance: overly strict balancing can lead to the loss of valuable features, increasing bias in treatment effect estimates. Zhang et al. (2020) and Johansson et al. (2019) noted that such regularization might lead to a loss of ignorability in the representation and suggested learning representations in which the context information remains preserved but where treatment groups overlap. To this end, Assaad et al. (2021) introduced the Balancing Weights Counterfactual Regression (BWCFR) method, which aims to achieve balance *within the reweighted covariate* representations instead of directly balancing the covariates representations. Assaad et al. (2021) argues that BWCFR provides bounds on the degree of imbalance as a function of the propensity model and offers theoretical guarantees for estimating the CATE using the overlapping weight. Johansson et al. (2022) built on the previous work on sampling weighting for counterfactual regression and representation learning (Johansson et al., 2016; Shalit et al., 2017; Kallus, 2020; Jung et al., 2020; Assaad et al., 2021), and provide a comprehensive theory for weighted risk minimization for CATE for a learned representation from the data. In this paper, we adapt the weighted empirical risk minimization (WERM) framework traditionally developed for static settings to the longitudinal context. Our approach emphasizes *contemporaneous treatments effects* over sequences of interventions (see Remark 1), allowing a natural adaptation of WERM to time-varying data.

**Probabilistic Modeling in Causal Inference**  In scenarios where confounders are unobserved but proxy variables are available, probabilistic models are used to infer a representation for the unobserved confounding given the proxy variables (Kuroki & Pearl, 2014; Miao et al., 2016; Louizos et al., 2017; Cheng et al., 2021). While our work assumes the presence of observed confounding, it draws inspiration from the theory of deconfounding (Lopez & Gutman, 2017; Ranganath & Perotte, 2018; Wang & Blei, 2019a), and its extensions to time-varying settings (Bica et al., 2020b; Hatt & Feuerriegel, 2024). The deconfounding involves applying a factor model over the treatment assignment, where each treatment becomes conditionally independent given latent variables that serve as substitutes for the unobserved confounders. Bica et al. (2020b) extended the application of the confounding method to sequential settings with multiple treatments to infer time-varying confounders. They assumed that the joint distribution of causes at each time step, conditioned on latent variables and observed confounders, could be decomposed into the product of the conditional distribution

of each treatment. For a single treatment per time step, Hatt & Feuerriegel (2024) assumed a conditional Markov model over the *treatment sequences* given the sequence of confounders and the latent variables. In this work, we demonstrate that the core idea of the factor model can be applied to learn valid substitutes for unobserved adjustment variables in the time-varying domain. Unlike previous deconfounder works, we show that assuming a higher-order conditional Markov model for the *sequence of responses* is sufficient to infer a valid representation of the unobserved adjustment variables.

## 3 Problem Definition

Following the Potential Outcome (PO) framework (Robins & Hernán, 2009b), we consider a cohort of individuals indexed as $i \in \{1, 2, \ldots, n\}$, observed over $T$ time steps. At each time point $t \in \{1, 2, \ldots, T\}$, we define: (1) A **binary treatment** $W_{it} \in \mathcal{W} = \{0, 1\}$, such as whether a cancer patient receives radiotherapy or not; (2) An **outcome** $Y_{it} \in \mathcal{Y} \subset \mathbb{R}$, which represents the response to the treatment (e.g., tumor volume); (3) A **context** $\mathbf{X}_{it} \in \mathcal{X} \subset \mathbb{R}^{d_x}$, a time-varying $d_x$-dimensional vector capturing confounders, such as health records and patient measurements; (4) **Partially observed potential outcomes** $Y_{it}(1)$ and $Y_{it}(0) \in \mathcal{Y} \subset \mathbb{R}$, which represent the outcomes that *would* be observed under treatments $W_{it} = 1$ and $W_{it} = 0$, respectively; (5) **Unobserved adjustment variables** $\mathbf{U}_i \in \mathcal{U} \subset \mathbb{R}^{d_u}$, which are static variables affecting the response series $Y_{i1}, Y_{i2}, \ldots, Y_{iT}$.

**Remark 1** *We could have defined the PO for an individual given their treatment history up to time $t$ as $Y_{it}(\omega_{i,\leq t})$, where $\omega_{i,\leq t} := (\omega_{i,1}, \omega_{i,2}, \ldots, \omega_{i,t}) \in \mathcal{W}^t$. Here, we focus on the contemporaneous treatment effect, that relates to the current treatment $W_{it}$. We assume the treatment history up to $t-1$, $\omega_{i,<t}$, to be consistent with the observed history, that is, $\omega_{i,<t} = W_{i,<t}$.*

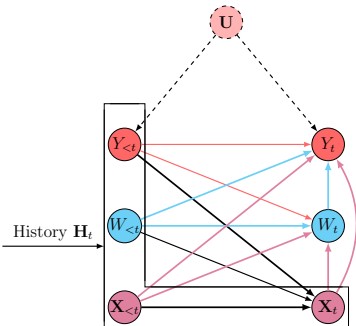

Figure 1: A simplified representation of the DGP at time $t$. Edges between $Y_{<t}$, $W_{<t}$, and $\mathbf{X}_{<t}$ are omitted for simplicity.

We define the *confounding history process* $\mathbf{H}_{it} = [\mathbf{X}_{i,\leq t}, W_{i,<t}, Y_{i,<t}]$, capturing information up to the assignment of $W_{it}$ depicted in Figure 1 which also compactly represents the Data Generating Process (DGP) for which we can estimate the CATE, that is

$$\tau_t(\mathbf{h}_t) := \mathbb{E}(Y_t(1) - Y_t(0) \mid \mathbf{H}_t = \mathbf{h}_t). \tag{1}$$

We assume the CATE identifiability from the observed data distribution using sequential ignorability (Robins & Hernán, 2009b; Lim, 2018; Bica et al., 2020a) represented in Assumptions 2.

**Assumption 2 (CATE Identifiability)** *We make three key identifiability assumptions at each time step $t$, for any potential treatment $\omega$, and given the realization $\mathbf{h}_{it}$ of the confounding history $\mathbf{H}_{it}$:*

1. ***Consistency.*** *If a unit $i$ receives treatment $\omega$, then the observed outcome corresponds to the potential outcome under $\omega$. Formally, $W_{it} = \omega \implies Y_{it} = Y_{it}(\omega)$.*
2. ***Unconfoundedness/Ignorability.*** *The potential outcomes $Y_{it}(\omega)$ are independent of the treatment assignment $W_t$, given the confounding history $\mathbf{H}_{it} = \mathbf{h}_{it}$. Thus, $Y_{it}(\omega) \perp\!\!\!\perp W_t \mid \mathbf{H}_{it} = \mathbf{h}_{it}$.*

*3.* ***Overlap/Positivity.*** *For any confounding context $\mathbf{h}_t$ where $p(\mathbf{h}_t) \neq 0$, there is a non-zero probability of observing each treatment regime. Formally, $p(W_t = \omega \mid \mathbf{h}_t) > 0$.*

With Assumptions 2, we ensure that the CATE is identifiable. Specifically, we have:

$$\tau_t(\mathbf{h}_t) = \mathbb{E}(Y_t \mid \mathbf{H}_t = \mathbf{h}_t, W_t = 1) - \mathbb{E}(Y_t \mid \mathbf{H}_t = \mathbf{h}_t, W_t = 0). \tag{2}$$

So far, we expressed the CATE as a function of the confounding history $\mathbf{H}_t$. Now, assuming we **observe adjustment variables $\mathbf{U}$** and aim to estimate a heterogeneous treatment effect depending on both $\mathbf{H}_t$ and $\mathbf{U}$. We define the *Augmented CATE (ACATE)* as follows:

$$\tau_t(\mathbf{h}_t, \mathbf{u}) := \mathbb{E}(Y_t(1) - Y_t(0) \mid \mathbf{H}_t = \mathbf{h}_t, \mathbf{U} = \mathbf{u}). \tag{3}$$

Since the adjustment variables $\mathbf{U}$ influence only the response series and do not act as confounders in the treatment-response relationship at any time step, the ignorability of treatment given $\mathbf{H}_t$ suffices for identifying the ACATE, as expressed below:

$$\tau_t(\mathbf{h}_t, \mathbf{u}) = \mathbb{E}(Y_t \mid \mathbf{H}_t = \mathbf{h}_t, \mathbf{U} = \mathbf{u}, W_t = 1) - \mathbb{E}(Y_t \mid \mathbf{H}_t = \mathbf{h}_t, \mathbf{U} = \mathbf{u}, W_t = 0). \tag{4}$$

**Inference Problem** Deriving the ACATE becomes straightforward when both confounding and adjustment variables are fully observed. However, when $\mathbf{U}$ is missing, unobserved heterogeneity in the treatment effect arises, leading to incomplete or biased conclusions about treatment effects. This raises the following key inference question: *Under what conditions can a substitute $\mathbf{Z}$ for the unobserved adjustment variables $\mathbf{U}$ ensure the identifiability of the ACATE by replacing $\mathbf{U}$ with $\mathbf{Z}$, specifically guaranteeing that the following equality (i) holds?*

$$\tau_t(\mathbf{h}_t, \mathbf{z}) = \mathbb{E}(Y_t(1) - Y_t(0) \mid \mathbf{H}_t = \mathbf{h}_t, \mathbf{Z} = \mathbf{z}) \overset{(i)}{=} \mathbb{E}(Y_t \mid \mathbf{H}_t = \mathbf{h}_t, \mathbf{Z} = \mathbf{z}, W_t = 1) - \mathbb{E}(Y_t \mid \mathbf{H}_t = \mathbf{h}_t, \mathbf{Z} = \mathbf{z}, W_t = 0).$$

## 4 Causal DVAE

### 4.1 When does a Latent Representation Act as a Valid Substitute for Unobserved Adjustment Variables?

To address the causal inference problem defined in Section 3, we treat $\mathbf{Z}$ as a latent variable to be learned from the observed data distribution $p(\mathbf{X}_{\leq T}, W_{\leq T}, Y_{\leq T})$ within a probabilistic model defined over the conditional responses as follows:

**Assumption 3 (CMM($p$): Conditional Markov Model of order $p$)** *We say that a latent variable $\mathbf{Z}$ follows a Conditional Markov Model of order $p$, written $\mathbf{Z} \sim CMM(p)$, if there exists a fixed order $p \in \mathbb{N}^*$ and a parameter vector $\theta$ such that the response distribution factorizes as:*

$$p_\theta(y_{\leq T}, \mathbf{z} \mid \mathbf{x}_{\leq T}, \omega_{\leq T}) = p(\mathbf{z}) \prod_{t=1}^{T} p_\theta\left(y_t \mid y_{t-1:t-p}, \mathbf{x}_{\leq t}, \omega_{\leq t}, \mathbf{z}\right).$$

*For $t < 1$, we set $y_t = \emptyset$ by convention.*

The fact that $\mathbf{Z}$ is set only with a prior in Assumption 3 mimics the role of $\mathbf{U}$ because $\mathbf{U}$ consists of static adjustment variables that are parentless, as illustrated in the causal graph (Figure 1). The bounded memory assumption over the sequence of responses, that is, the direct causal effect of past responses on future ones stops at an arbitrary order $p$, is a technical condition ensuring the causal validity of $\mathbf{Z}$ as in the Theorem 4 where we demonstrate the sequential ignorability property when augmenting the history process with $\mathbf{Z}$, replicating the same result as if the true adjustment variables $\mathbf{U}$ were available. All the proofs are deferred to Appendix A.

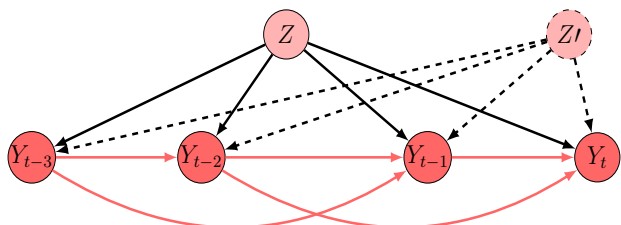

Figure 2: A simplified causal graph for the sketch of the proof for Theorem 6. We do not represent $W_{\leq t}, \mathbf{X}_{\leq t}$ for simplicity.

**Theorem 4 (Sequential Ignorability with Augmented History)** *Let $\mathbf{Z}$ be a latent variable verifying CMM(p). Assume the response domain $\mathcal{Y}$ is a Borel subset of a compact interval. Therefore, sequential ignorability holds when augmenting the history process with $\mathbf{Z}$:*

$$Y_t(\omega) \perp\!\!\!\perp W_t | \mathbf{H}_t = \mathbf{h}_t, \mathbf{Z} = \mathbf{z} \quad \forall(\omega, \mathbf{h}_t, \mathbf{z}),$$

*where $\mathbf{H}_t$ represents the history process up to time t.*

The ignorability result of Theorem 4 is the first step into answering the inference problem of 3 as it allows us to establish the identifiability of the ACATE when $\mathbf{Z}$ replaces the true unobserved variable $\mathbf{U}$ in Eq. (4), summarized in the following corollary:

**Corollary 5 (Identifiability of ACATE with Z)** *Let $\mathbf{Z}$ be a latent variable satisfying CMM(p). The Augmented CATE, when augmented with $\mathbf{Z}$ instead of $\mathbf{U}$ as defined in Equation 4, is identifiable. Specifically:*

$$\tau_t(\mathbf{h}_t, \mathbf{z}) = \mathbb{E}(Y_t | \mathbf{H}_t = \mathbf{h}_t, \mathbf{Z} = \mathbf{z}, W_t = 1) - \mathbb{E}(Y_t | \mathbf{H}_t = \mathbf{h}_t, \mathbf{Z} = \mathbf{z}, W_t = 0) \tag{5}$$

As a result, Corollary 5 addresses the identifiability problem and its conditions as raised in Section 3: A CMM of arbitrary order over the conditional distribution of responses, along with a mild regularity condition on the response domain as stated in Theorem 4, constitutes a sufficient condition to ensure the identifiability of the ACATE. To further emphasize the relevance of $\mathbf{Z} \sim CMM(p)$, we show that any two substitutes satisfying $CMM(p)$ must necessarily be related through a measurable map given the entire process history $\mathbf{H}_T$:

**Theorem 6** *Let $\mathbf{Z}$ be a latent variable such that $\mathbf{Z} \sim CMM(p)$, and assume the domain $\mathcal{Y}$ is a Borel subset of a compact interval. Then, any static adjustment variable that influences the entire series of responses in the panel must be measurable with respect to $(\mathbf{Z}, \mathbf{H}_T)$.*

**Intuition behind CMM(p)** We provide an intuition behind the CMM(p) assumption by sketching a proof of Theorem 6 using d-separation properties, assuming the data generation process follows the graph depicted in Figure 1. Let $\mathbf{Z}'$ be an unobserved risk variable independent of $\mathbf{Z}$, as illustrated in Figure 2. Given that $\mathbf{Z} \sim CMM(p)$ and for $m > p$, $Y_t$ and $Y_{t-m}$ are d-separated given $\{Y_{t-1:t-m+1}, \mathbf{Z}, W_{\leq t}, \mathbf{X}_{\leq t}\}$ implying that $Y_t \perp\!\!\!\perp Y_{t-m} \mid Y_{t-1:t-m+1}, \mathbf{Z}, W_{\leq t}, \mathbf{X}_{\leq t}$. Now, suppose $\mathbf{Z}'$ influences the entire series of responses similarly to $\mathbf{Z}$. In this case, $\mathbf{Z}'$ acts as a common parent for both $Y_t$ and $Y_{t-m}$, creating a path $Y_{t-m} \leftarrow \mathbf{Z}' \rightarrow Y_t$ that cannot be blocked by conditioning on $\{Y_{t-1:t-m+1}, \mathbf{Z}, W_{\leq t}, \mathbf{X}_{\leq t}\}$. Consequently, $Y_t \not\perp\!\!\!\perp Y_{t-m} \mid Y_{t-1:t-m+1}, \mathbf{Z}, W_{\leq t}, \mathbf{X}_{\leq t}$, which contradicts the implications of the conditional Markov assumption.

## 4.2 Definition of the Probabilistic Model

We define in the section the probabilistic model to learn a substitute $\mathbf{Z}$ building on the DVAE approach. We break down the approach into three steps, where we first deal with selection bias and then incorporate the

inductive bias of population structure, defining a GMM over the substitutes and finally define the suitable Evidence Lower Bound (ELBO) to be maximized.

**Step 1: Handling Selection Bias** In a purely factual setting, modeling data with a DVAE involves maximizing:

$$\log p_\theta(y_{\leq T} \mid \mathbf{x}_{\leq T}, \omega_{\leq T}) = \sum_{t=1}^{T} \log p_\theta(y_t \mid y_{<t}, \mathbf{x}_{\leq t}, \omega_{\leq t}) = \sum_{t=1}^{T} \log p_\theta(y_t \mid \mathbf{h}_t, \omega_t)$$

based on the observed data distribution. However, this model cannot perform counterfactual regression—switching treatment assignment during inference to estimate counterfactual responses—nor does it allow for causal inference. This limitation arises in observational studies where treatment assignment is typically non-random and depends on observed covariates, resulting in a selection bias. We adopt the importance sampling strategy to reduce the selection bias by reweighing the likelihood of the factual model (Shimodaira, 2000). The key idea is to assign weights $\{\alpha(\mathbf{h}_t, \omega_t)\}_{t=1}^{T}$ to units in the population such that the resulting distribution in a given treatment regime matches that of the entire population. Therefore, we seek to maximize the likelihood $L$ defined as:

$$L := \sum_{t=1}^{T} \mathbb{E}_{\mathbf{H}_t, W_t} \left[ \alpha(\mathbf{H}_t, W_t) \mathbb{E}_{Y_t \mid \mathbf{H}_t, W_t} \log p_\theta(y_t \mid \mathbf{H}_t, W_t) \right].$$

A more convenient formulation expresses $L$ as an expectation over all repeated cross-sectional data of a population:

$$L = \sum_{t=1}^{T} \mathbb{E}_{\mathcal{D}_T} \left[ \alpha(\mathbf{H}_t, W_t) \log p_\theta(Y_t \mid \mathbf{H}_t, W_t) \right], \tag{6}$$

with $\mathcal{D}_T = \{Y_{\leq T}, \mathbf{X}_{\leq T}, W_{\leq T}\}$. The formulation of a *conditional* DVAE model specifies the conditional generative model $p_\theta(y_{\leq T}, \mathbf{z} \mid \mathbf{x}_{\leq T}, \omega_{\leq T})$, which factorizes as: $p_\theta(y_{\leq T}, \mathbf{z} \mid \mathbf{x}_{\leq T}, \omega_{\leq T}) = \prod_{t=1}^{T} \left[ p_\theta(y_t \mid y_{<t}, \mathbf{x}_{\leq t}, \omega_{\leq t}, \mathbf{z}) \right] p(\mathbf{z})$.

**Step 2: Incorporating Population Structure** To account for an eventual population structure induced by unobserved adjustment variables, we place a Gaussian mixture prior over the latent variable $\mathbf{Z}$, with $K$ components:

$$p(\mathbf{z}) = \sum_{c=1}^{K} \pi_c \mathcal{N}(\mathbf{z} \mid \mu_c, \Sigma_c). \tag{7}$$

This prior corresponds to first generating a cluster index $C \sim \mathrm{Cat}(\pi)$, followed by generating $\mathbf{Z} \mid C \sim \mathcal{N}(\mu_C, \Sigma_C)$. Here, $\pi$ represents the probability distribution over cluster assignments, verifying $\sum_{c=1}^{K} \pi_c = 1$, with $\mathrm{Cat}(\pi)$ denoting the categorical distribution parameterized by $\pi$. Each cluster $c$ is associated with a learnable Gaussian distribution of mean $\mu_c$ and covariance $\Sigma_c$. The complete generative model becomes:

$$p_\theta(y_{\leq T}, \mathbf{z}, c \mid \mathbf{x}_{\leq T}, \omega_{\leq T}) = \prod_{t=1}^{T} \left[ p_\theta(y_t \mid y_{<t}, \mathbf{x}_{\leq t}, \omega_{\leq t}, \mathbf{z}) \right] p(\mathbf{z} \mid c) p(c). \tag{8}$$

We approximate the true posterior over $(\mathbf{z}, c)$ using a factorized variational distribution

$$q_\phi(\mathbf{z}, c \mid \mathcal{D}_T) = q_{\phi_z}(\mathbf{z} \mid \mathcal{D}_T) q_{\phi_c}(c \mid \mathcal{D}_T), \tag{9}$$

where $\phi = \phi_z \cup \phi_c$ collects the variational parameters. The following theorem characterizes the corresponding variational bound.

**Theorem 7 (Weighted ELBO Decomposition)** *Under the generative model defined in Eq. (8) and the approximate posterior factorization (Eq. (9)), the weighted likelihood in Eq. (6) can be decomposed as:*

$$L = \mathbb{E}_{\mathcal{D}_T} \left[ \mathrm{ELBO}_0(\mathcal{D}_T; \theta, \phi) \right] + \mathbb{E}_{\mathcal{D}_T} \left[ \Delta_0(\mathcal{D}_T; \theta, \phi) \right], \tag{10}$$

*where the individual gap term $\Delta_0$ is given by:*

$$\Delta_0(\mathcal{D}_T; \theta, \phi) := \sum_{t=1}^{T} \alpha(\mathbf{h}_t, \omega_t) D_{KL}(q_\phi(\mathbf{z}, c \mid \mathcal{D}_T) \parallel p_\theta(\mathbf{z}, c \mid \mathcal{D}_T)), \tag{11}$$

*and the individual ELBO termed* $\mathrm{ELBO}_0$ *is expressed as:*

$$\mathrm{ELBO}_0(\mathcal{D}_T; \theta, \phi) = \sum_{t=1}^{T} \mathbb{E}_{\mathbf{Z} \sim q_{\phi_z}(\cdot \mid \mathcal{D}_T)} \left[ \alpha(\mathbf{h}_t, \omega_t) \log p_\theta(y_t \mid \mathbf{h}_t, \omega_t, \mathbf{Z}) \right] \tag{12}$$

$$- \left( \sum_{t=1}^{T} \alpha(\mathbf{h}_t, \omega_t) \right) \left\{ D_{KL}(q_{\phi_z}(\mathbf{z} \mid \mathcal{D}_T) \parallel p(\mathbf{z})) + \mathbb{E}_{\mathbf{Z} \sim q_{\phi_z}(\cdot \mid \mathcal{D}_T)} D_{KL}(q_{\phi_c}(c \mid \mathcal{D}_T) \parallel p(c \mid \mathbf{Z})) \right\}.$$

The individual gap $\Delta_0$ is a positive term that quantifies the inaccuracy of the variational approximation of $L$ using $\mathrm{ELBO}_0$ (Eq. (12)). The ELBO term, $\mathrm{ELBO}_0$, comprises three components: (1) A weighted sum of the conditional log-likelihoods of the responses (reconstruction term). (2) A KL divergence between the approximate posterior of the continuous latent variables $q_{\phi_z}(\cdot \mid \mathcal{D}_T)$, which serve as substitutes for adjustment variables, and the unconditional prior $p(\mathbf{z})$. (3) A KL divergence between the approximate posterior of the discrete latents $q_{\phi_c}(\cdot \mid \mathcal{D}_T)$ and the true conditional posterior $p(\mathbf{C} \mid \mathbf{Z})$, averaged over $q_{\phi_z}(\cdot \mid \mathcal{D}_T)$. The decoupling of these two approximate posteriors arises from the assumed factorization over the joint approximate posterior as in Eq. (9).

**Step 3: Addressing the Discrete Latent Variable** $c$   The approximate posterior $q_\phi(\mathbf{z}, c \mid \mathcal{D}_T)$ involves both continuous and discrete latent variables, $\mathbf{z}$ and $c$, respectively. Training deep generative models with discrete latent variables presents challenges due to difficulties in reparameterization and handling high cardinality, even when using classical approaches like Gumbel-Softmax (Jang et al., 2017; Tucker et al., 2017; Huijben et al., 2022). To avoid designing an additional inference network for $q_{\phi_c}(c \mid \mathcal{D}_T)$ and the associated computational cost, we follow Jiang et al. (2017); Falck et al. (2021) and define a *Bayes-optimal posterior* for the discrete latent variable. Specifically, we leverage the decomposition of Eq. (9) and choose $q_{\phi_c}(c \mid \mathcal{D}_T)$ in such a way that the third term in the ELBO of Eq. (12) is minimized by definition:

$$\min_{q_{\phi_c}(\cdot \mid \mathcal{D}_T)} \mathbb{E}_{\mathbf{Z} \sim q_{\phi_z}(\cdot \mid \mathcal{D}_T)} D_{KL}(q_{\phi_c}(c \mid \mathcal{D}_T) \parallel p(c \mid \mathbf{Z})). \tag{13}$$

The minimal value of this optimization problem is

$$\min_{q_{\phi_c}(\cdot \mid \mathcal{D}_T)} \mathbb{E}_{\mathbf{Z} \sim q_{\phi_z}(\cdot \mid \mathcal{D}_T)} D_{KL}(q_{\phi_c}(\cdot \mid \mathcal{D}_T) \parallel p(\cdot \mid \mathbf{Z})) = -\log \mathrm{Const}(q_{\phi_z}(\cdot \mid \mathcal{D}_T)), \tag{14}$$

where $\mathrm{Const}(q_{\phi_z}(\cdot \mid \mathcal{D}_T))$ is a constant that depends only on the approximate posterior over the continuous latents:

$$\mathrm{Const}(q_{\phi_z}(\cdot \mid \mathcal{D}_T)) = \sum_{c=1}^{K} \exp(\mathbb{E}_{\mathbf{Z} \sim q_{\phi_z}(\cdot \mid \mathcal{D}_T)} \log p(c \mid \mathbf{Z})).$$

The minimizer is given by:

$$\pi(c \mid q_{\phi_z}(\cdot \mid \mathcal{D}_T)) = \frac{\exp(\mathbb{E}_{\mathbf{Z} \sim q_{\phi_z}(\cdot \mid \mathcal{D}_T)} \log p(c \mid \mathbf{Z}))}{Cst(q_{\phi_z}(\cdot \mid \mathcal{D}_T))}. \tag{15}$$

Moreover, the minimal value is:

$$\min_{q_{\phi_c}(\cdot \mid \mathcal{D}_T)} \mathbb{E}_{\mathbf{Z} \sim q_{\phi_z}(\cdot \mid \mathcal{D}_T)} D_{KL}(q_{\phi_c}(\cdot \mid \mathcal{D}_T) \parallel p(c \mid \mathbf{Z})) = -\log Z(q_{\phi_z}(\cdot \mid \mathcal{D}_T)). \tag{16}$$

Thus, we update the original $\mathrm{ELBO}_0$ in Eq. (12) to obtain a *modified* ELBO by replacing $q_{\phi_c}(\cdot \mid \mathcal{D}_T)$ with the minimizer $\pi(c \mid q_{\phi_z}(\cdot \mid \mathcal{D}_T))$ and plugging in the minimal value from Eq. (16):

$$\mathrm{ELBO}(\mathcal{D}_T; \theta, \phi) = \sum_{t=1}^{T} \mathbb{E}_{\mathbf{Z} \sim q_{\phi_z}(\cdot \mid \mathcal{D}_T)} \left[ \alpha(\mathbf{h}_t, \omega_t) \log p_\theta(y_t \mid \mathbf{H}_t, W_t, \mathbf{Z}) \right] \tag{17}$$

$$- \left[ \sum_{t=1}^{T} \alpha(\mathbf{h}_t, \omega_t) \right] \left\{ D_{KL}(q_{\phi_z}(\cdot \mid \mathcal{D}_T) \parallel p(\mathbf{z})) - \log Z(q_{\phi_z}(\cdot \mid \mathcal{D}_T)) \right\}.$$

As a result, the modified ELBO is computationally more efficient than the original one, as it bypasses the need for applying a reparameterization trick to discrete latents and eliminates the necessity of training over specific parameters $\phi_c$, all while remaining a valid lower bound for the weighted likelihood $L$. This validity holds since Theorem 7 applies to any arbitrary approximate posterior $q_{\phi_c}(\cdot \mid \mathcal{D}_T)$, and in particular, to $\pi(c \mid q_{\phi_z}(\cdot \mid \mathcal{D}_T))$, which is the minimizer of Eq. (13). We henceforth use $\phi$ to denote $\phi_z$.

### 4.3 CDVAE in the Near-Deterministic Regime

In this section, we explore the behavior of Causal DVAE in the *near-deterministic regime*, where the variance of the generative model approaches zero. We examine how this regime affects both inference and causal identifiability, demonstrating that it leads to a non-diffuse posterior suitable for treatment effect estimation.

**Model Setup**   For the remainder of our discussion, we define the DVAE model parameters as $\mathcal{M}_{vae} = \{\theta, \phi\}$ and assume the approximate posterior to be Gaussian $q_\phi(\mathbf{z} \mid \mathcal{D}_T) \sim \mathcal{N}(\mathbf{z}|\mu_{\mathbf{z}}, \Sigma_{\mathbf{z}})$, where the mean is defined as $\mu_{\mathbf{z}} = f_{\mu_{\mathbf{z}}}(\mathcal{D}_T, \phi)$, and the covariance matrix is given by $\Sigma_{\mathbf{z}} = S_{\mathbf{z}} S_{\mathbf{z}}^\top$, with $S_{\mathbf{z}} = f_{S_{\mathbf{z}}}(\mathcal{D}_T, \phi)$. Moreover, we assume the conditional generative model to be Gaussian such that:

$$p_\theta(y_t \mid y_{<t}, \mathbf{x}_{\leq t}, \omega_{\leq t}, \mathbf{z}) = \mathcal{N}(f_t(\mathbf{h}_t, \omega_t, \mathbf{z}; \theta), \sigma^2), \tag{18}$$

with $\sigma > 0$, a spatially and temporally uniform scale parameter.

**Motivation**   In probabilistic modeling, our inference model for the substitute adjustment variable is *stochastic*. However, to calculate an individualized ACATE for each individual $i$, that is

$$\tau_t(\mathbf{h}_{it}, \mathbf{z}_i) = \mathbb{E}(Y_t|\mathbf{H}_{it} = \mathbf{h}_{it}, \mathbf{Z} = \mathbf{z}_i, W_t = 1) - \mathbb{E}(Y_t|\mathbf{H}_t = \mathbf{h}_{it}, \mathbf{Z} = \mathbf{z}_i, W_t = 0),$$

we need to choose a unique instance of the latent variable representation. Otherwise, infinitely many ACATEs for the same individual $i$ could be generated by sampling repeatedly from $q_{\phi_z}(\cdot \mid \mathcal{D}_{iT})$. A natural choice for this deterministic representation is the *mean* of the posterior. Yet, because we sample from the approximate posterior during training rather than using the mean, it is essential to prevent the posterior from becoming *overly diffuse*—ensuring that the mean remains a valid representation during causal inference. Previous studies (Dai & Wipf, 2019; Takida et al., 2022) show that operating in the *near-deterministic regime* of the VAE decoder—where $\sigma \to 0^+$—leads the encoder's covariance matrix toward zero, resulting in near-deterministic behavior. We, therefore, explore the near-deterministic regime in our Causal DVAE by controlling the decoder's variance to induce a non-diffuse, stable posterior distribution suitable for causal inference in the latent space.

**Side Benefit: Preventing Posterior Collapse**   A well-known challenge in training VAEs is *posterior collapse*, where the approximate posterior converges to the uninformative prior, making the latent space irrelevant (Bowman et al., 2015; Sønderby et al., 2016; Higgins et al., 2017; Dai & Wipf, 2019; Fu et al., 2019; Lucas et al., 2019; Wang et al., 2021; Takida et al., 2022). A notable advantage of the near-deterministic regime is its ability to *avoid posterior collapse* within the latent space, a benefit demonstrated in *static VAEs* (Lucas et al., 2019; Takida et al., 2022). Specifically, Takida et al. (2022) showed that the decoder's output variance and covariance could influence the latent space by causing over-smoothing by affecting the gradient regularization strength, which in turn leads to posterior collapse. By controlling variance in the near-deterministic regime, we mitigate this collapse and thereby preserve meaningful latent representations.

We advance the study of *dynamic* VAE models by investigating their behavior in the near-deterministic regime within the context of causal inference. This work is the first to explore dynamic VAEs in this regime for causal applications, addressing a gap in the literature, as such behavior has yet to be thoroughly examined even in non-causal, factual settings. For the remainder, and unless otherwise stated, we omit dependence on the individual $i$ for notational ease.

We prove that in our framework, replacing the original lower bound $\text{ELBO}_0(\mathcal{D}_T; \theta, \phi)$ with the modified $\text{ELBO}(\mathcal{D}_T; \theta, \phi)$ still ensures the existence of a parameterization under which the approximate likelihood of responses converges to the true likelihood. Specifically, Theorem 8 demonstrates that, for a sequence of

inference and generative models parameterized by $\sigma$, the gap between the modified and true importance-weighted likelihoods asymptotically approaches zero.

**Theorem 8 (Asymptotic Likelihood Recovery)** *Suppose $d_{\mathbf{z}} \leq T$. There exists a family of autoencoders $\{\phi_\sigma, \theta_\sigma\}_{\sigma>0}$ such that*

$$\lim_{\sigma \to 0^+} p_{\theta_\sigma}(y_{\leq T} \mid \mathbf{x}_{\leq T}, \omega_{\leq T}) = p(y_{\leq T} \mid \mathbf{x}_{\leq T}, \omega_{\leq T}),$$

*and when operating under the Bayes-optimal posterior over the discrete latents, i.e.,*

$$q_\phi(c \mid \mathcal{D}_T) = \pi(c \mid q_\phi(\cdot \mid \mathcal{D}_T)),$$

*the gap $\Delta(\mathcal{D}_T; \theta_\sigma, \phi_\sigma)$ between the modified ELBO (Eq. (17)) and the true weighted likelihood $L$ (Eq. (6)) satisfies*

$$\Delta(\mathcal{D}_T; \theta_\sigma, \phi_\sigma) = \text{ELBO}(\mathcal{D}_T; \theta_\sigma, \phi_\sigma) - L(\mathcal{D}_T; \theta_\sigma)$$

*and converges to zero:*

$$\lim_{\sigma \to 0^+} \Delta(\mathcal{D}_T; \theta_\sigma, \phi_\sigma) = 0.$$

This result generalizes prior findings for static VAEs (Dai & Wipf, 2019) to the more complex dynamic setting with GMM priors and weighted objectives. It also addresses an oversight in Dai & Wipf (2019) regarding the proof of gap convergence.

More importantly, we establish a critical property in our setting within the near-deterministic regime: the causal effect remains asymptotically consistent whether one uses an arbitrary realization of the latent variable $\mathbf{z}$ or the posterior mean $\mu_{\mathbf{z}}$. Theorem 9 formalizes this insight as follows:

**Theorem 9 (Realization-Invariant Causal Consistency)** *For a fixed $\sigma > 0$, let a parametrization of Causal DVAE $\mathcal{M}^*_{vae}(\sigma) = \{\theta^*_\sigma, \phi^*_\sigma\}$ be an optimal solution to the ELBO defined in Eq. (17). Then, for any substitute realization $\mathbf{z} \in \mathbb{R}^{d_z}$, the conditional expected outcome satisfies the following consistency in the near-deterministic regime:*

$$\lim_{\sigma \to 0^+} \mathbb{E}_{\mathcal{M}^*_{vae}(\sigma)} \left[ Y_t \mid \mathbf{H}_t, W_t = \omega, \mathbf{Z} = \mathbf{z} \right] = \lim_{\sigma \to 0^+} \mathbb{E}_{\mathcal{M}^*_{vae}(\sigma)} \left[ Y_t \mid \mathbf{H}_t, W_t = \omega, \mathbf{Z} = \mu_{\mathbf{z}} \right]. \tag{19}$$

*As a result of Corollary 5, the Augmented CATE satisfies the same consistency in the near-deterministic regime:*

$$\lim_{\sigma \to 0^+} \tau_{\mathcal{M}^*_{vae}(\sigma)}(\mathbf{H}_t, \mathbf{z}) = \lim_{\sigma \to 0^+} \tau_{\mathcal{M}^*_{vae}(\sigma)}(\mathbf{H}_t, \mu_{\mathbf{z}}). \tag{20}$$

Theorem 9 demonstrates that as the variance parameter $\sigma$ of the generative model approaches zero, both the expected outcome and the estimated causal effect remain consistent, regardless of whether one relies on an arbitrary realization of the latent variable $\mathbf{z}$ or its posterior mean $\mu_{\mathbf{z}}$. Such consistency ensures that, in the near-deterministic regime, the causal effect is robust to the specific realization of the latent variable.

## 4.4 Generalization Bound over Treatment Effect Estimation and Derivation of Model Loss

We establish in this section a theoretical generalization bound for the error in estimating ACATE, which serves as a foundation for deriving the training loss of CDVAE. We begin by defining a suitable representation function to address the high dimensionality of the history $\mathbf{H}_t$. Next, we introduce a weighted risk minimization framework that ensures covariate balance between treatment groups. We then derive a generalization bound for treatment effect estimation, relating the error in estimating ACATE to the model's weighted risk. Finally, we construct the CDVAE training objective by integrating these theoretical insights into a structured loss function.

**Step 1: Representation Learning** To address the high dimensionality of the history $\mathbf{H}_t$, we define a representation function $\Phi$ that reduces dimensionality while preserving essential information in a latent space. Assuming $\Phi$ is invertible, we ensure sequential ignorability holds in the latent space if and only if it holds in the original space. In practice, the outcome and treatment models share the same representation $\Phi(\mathbf{H}_t)$, capturing information predictive of both treatment and response. While invertibility is theoretically assumed for identifiability, it is not strictly enforced in implementation. Instead, we prioritize ensuring $\Phi(\mathbf{H}_t)$ remains predictive across treatment regimes, applying regularization techniques to balance representations as in Shalit et al. (2017); Lim (2018); Bica et al. (2020a); Assaad et al. (2021); Melnychuk et al. (2022); Johansson et al. (2022).

**Remark 10** *Following Eq. (18), the generative model for the responses is defined as $Y_t = f_t(\mathbf{H}_t, W_t, \mathbf{Z}) + \epsilon_t$, where $\epsilon_t \sim \mathcal{N}(0, \sigma^2)$ is the error term. A common modeling (Johansson et al., 2016; Shalit et al., 2017; Shi et al., 2019; Johansson et al., 2022) approach for $f_t$ is to assume*

$$f_t(\mathbf{H}_t, W_t, \mathbf{Z}) := W_t f_{t,1}(\mathbf{H}_t, \mathbf{Z}) + (1 - W_t) f_{t,0}(\mathbf{H}_t, \mathbf{Z}),$$

*where $f_{t,0}$ and $f_{t,1}$ denote the mappings to the response under each treatment regime $W_t = 0, 1$.*

*The representation function $\Phi$ over $\{\mathbf{H}_t\}_{t=1}^T$ produces lower-dimensional vectors $\{\mathbf{r}_t\}_{t=1}^T$, living in a space $\mathcal{R} \subset \mathbb{R}^r$. Instead of defining time-dependent generative functions $f_t$, we define a single hypothesis function*

$$f : \mathcal{R} \times \mathcal{W} \times \mathbb{R}^{d_\mathbf{z}} \to \mathcal{Y},$$

*such that:*
$$Y_t = f(\Phi(\mathbf{H}_t), W_t, \mathbf{Z}) + \epsilon_t = W_t f_1(\Phi(\mathbf{H}_t), \mathbf{Z}) + (1 - W_t) f_0(\Phi(\mathbf{H}_t), \mathbf{Z}) + \epsilon_t.$$

**Step 2: Weighted Risk Minimization** To integrate causal inference into the DVAE framework, we employ weighted risk minimization techniques (Kallus, 2020; Johansson et al., 2022). First, we justify the choice of weights $\alpha(\mathbf{H}_t, W_t)$ used in the definition of the weighted ELBO in Section 4.2, Eq. (17). In general, these weights can be any arbitrary mapping $\alpha : \mathbf{H}_t \times \mathcal{W} \to \mathbb{R}^+$ such that for every $\omega \in \mathcal{W}$, $\mathbb{E}_{\mathbf{H}_t | W_t = \omega} [\alpha(\mathbf{H}_t, \omega)] = 1$. The weights $\alpha(\mathbf{h}_t, \omega)$ induce, therefore, a weighted probability *target distribution* $g$ over the population s.t $g(\mathbf{h}_t \mid W_t = \omega) := \alpha(\mathbf{h}_t, \omega) p(\mathbf{h}_t \mid W_t = \omega)$. Choosing an appropriate weighting strategy $\alpha(\mathbf{h}_t, \omega)$ is essential for achieving covariate balance between treatment groups, specifically $g(\mathbf{h}_t \mid W_t = 1) = g(\mathbf{h}_t \mid W_t = 0)$. We use *overlap weights* (Li et al., 2018; Assaad et al., 2021), defined as:

$$\alpha(\mathbf{h}_t, \omega) \propto \frac{e(\Phi(\mathbf{h}_t))\,(1 - e(\Phi(\mathbf{h}_t)))}{\omega\, e(\Phi(\mathbf{h}_t)) + (1 - \omega)\,(1 - e(\Phi(\mathbf{h}_t)))},$$

where $e(\Phi(\mathbf{h}_t)) = p(W_t = 1 \mid \Phi(\mathbf{h}_t))$ is the propensity score within the representation space. Overlap weights emphasize units with propensity scores near 0.5, concentrating on regions where treated and control groups have the most overlap, thereby enhancing comparability and improving covariate balance (Li et al., 2018).

We now define the *weighted population risk* for a given treatment regime $W_t = \omega$, measuring the expected risk for a given $(f, \Phi)$ at time $t$ and over the weighted population distributed according to $g$, as:

$$R_{t,g}^\omega(f, \Phi) := \mathbb{E}_{\mathbf{H}_t | W_t} [\alpha(\mathbf{H}_t, W_t)\, \ell_{f,\Phi}(\mathbf{H}_t, W_t) \mid W_t = \omega],$$

where $\ell_{f,\Phi}(\mathbf{h}_t, \omega)$ is the expected pointwise loss associated with the hypothesis $f$:

$$\ell_{f,\Phi}(\mathbf{h}_t, \omega) := \mathbb{E}_{\mathbf{Z} \sim q_\phi(\mathbf{Z} | \mathcal{D}_{\leq t-1})} \mathbb{E}_{Y_t(\omega) | \mathbf{H}_t, \mathbf{Z}} [L(Y_t(\omega), f(\Phi(\mathbf{H}_t), \mathbf{Z}, W_t)) \mid \mathbf{H}_t = \mathbf{h}_t, \mathbf{Z}]. \tag{21}$$

In the definition of the expected pointwise loss, we respect the temporal order by conditioning the approximate posterior of the continuous latents only on the longitudinal data up to time step $t$. In line with our probabilistic modeling in Section 4.2 and the distributional assumptions in Section 4.3, we define the loss as the negative log-likelihood:

$$L(Y_t(\omega), f(\Phi(\mathbf{H}_t), \mathbf{Z}, W_t)) = -\log \mathcal{N}(Y_t(\omega); f(\Phi(\mathbf{H}_t), \mathbf{Z}, W_t), \sigma^2).$$

**Step 3: Generalization Bound** Our aim during training is to minimize the factual quantities $\{R_{t,g}^{\omega}(f, \Phi)\}_{t=1}^{T}$ to reduce the error in estimating treatment effects. To formalize and theoretically justify this objective, we establish a generalization bound for the *Precision in Estimation of Heterogeneous Effects (PEHE)* (Hill, 2011), which accounts for our weighted population risks $\{R_{t,g}^{\omega}(f, \Phi)\}_{t=1}^{T}$. PEHE measures the mean squared error (MSE) between the true and estimated ACATE at time $t$ as [2]

$$\epsilon_{\text{PEHE}_t} = \mathbb{E}_{\mathbf{H}_t} \mathbb{E}_{\mathbf{Z} \sim q_\phi(\mathbf{Z}|\mathcal{D}_{\leq t-1})} \left[ (\tau(\mathbf{H}_t, \mathbf{Z}) - \hat{\tau}_{f,\Phi}(\mathbf{H}_t, \mathbf{Z}))^2 \right],$$

where $\tau(\mathbf{H}_t, \mathbf{Z})$ is the true ACATE and $\hat{\tau}_{f,\Phi}(\mathbf{H}_t, \mathbf{Z})$ is the estimated ACATE. Similarly, we define the weighted PEHE relative to the target distribution $g$:

$$\epsilon_{\text{PEHE}_{t,g}} = \mathbb{E}_{\mathbf{H}_t \sim g} \mathbb{E}_{\mathbf{Z} \sim q_\phi(\mathbf{Z}|\mathcal{D}_{\leq t-1})} \left[ (\tau(\mathbf{H}_t, \mathbf{Z}) - \hat{\tau}_{f,\Phi}(\mathbf{H}_t, \mathbf{Z}))^2 \right].$$

We provide an upper bound for the weighted PEHE for CDVAE similar to Assaad et al. (2021). This bound consists of three key components: the weighted factual prediction error, a term capturing the discrepancy between treatment and control distributions in the confounders representation space, and a term that accounts for the variance of the generative model over responses.

**Theorem 11 (Generalization Bound for Weighted PEHE)** *Let $\mathcal{M}_{VAE}(\sigma) = \{\theta_\sigma, \phi_\sigma\}$ be the VAE model defined in Section 4.2. For a given class of functions $G$, assume there exists a constant $B_\Phi$ such that $\frac{\ell_{f,\Phi}}{B_\Phi} \in G$. Assume the representation $\Phi$ is invertible. Then, the error in estimating the treatment effect at time $t$ for a weighted population is upper bounded by:*

$$\epsilon_{\text{PEHE}_{t,g}} \leq 2\sigma^2 \left\{ R_{t,g}^{\omega=1}(f, \Phi) + R_{t,g}^{\omega=0}(f, \Phi) + B_\Phi \text{IPM}_G \left( g_\Phi(\cdot \mid W_t = 1), g_\Phi(\cdot \mid W_t = 0) \right) - \log(2\pi\sigma^2) \right\}, \quad (22)$$

*where $g_\Phi(r)$ is the distribution in the representation space $\mathcal{R}$. The Integral Probability Metric (IPM) (Müller, 1997; Sriperumbudur et al., 2009) measures the dissimilarity between distributions.*

*Assuming strict overlap in treatment assignment, where $\delta \in (0, 0.5)$ such that $\delta < e(\mathbf{h}_t) < 1 - \delta$, then there exists constants $A_{t,g}, B_{t,g}$ such that the unweighted PEHE $\epsilon_{\text{PEHE}_t}$ is bounded:*

$$A_{t,g} \cdot \epsilon_{\text{PEHE}_{t,g}}(\hat{\tau}) \leq \epsilon_{\text{PEHE}_t}(\hat{\tau}) \leq B_{t,g} \cdot \epsilon_{\text{PEHE}_{t,g}}(\hat{\tau}).$$

The representation discrepancy in 22 quantifies the imbalance between treatment groups, measured using IPMs like the Wasserstein distance or Maximum Mean Discrepancy (Gretton et al., 2012). The connection between $\epsilon_{\text{PEHE}_t}$ and $\epsilon_{\text{PEHE}_{t,g}}$ indicates that minimizing the weighted PEHE can also minimize $\epsilon_{\text{PEHE}_t}$, thereby enhancing the reliability of CATEs estimation across the original population.

To justify the pertinence of our probabilistic model defined in Section 4.2, we will show how maximizing the likelihood term in the ELBO of Eq. (17) implies minimization of the risks $R_{t,g}^{\omega=1}(f, \Phi)$ and $R_{t,g}^{\omega=0}(f, \Phi)$.

**Proposition 12 (ELBO-Risk Connection)** *Assume stationarity of the approximated posterior of $\mathbf{Z}$ given sub-longitudinal data; that is, there exists $t_0$ such that for $t \geq t_0$, $q_\phi(\mathbf{Z} \mid \mathcal{D}_t) \approx q_\phi(\mathbf{Z} \mid \mathcal{D}_T)$. Given a finite sample $\mathcal{B} = \{\mathcal{D}_{iT} = \{W_{it}, Y_{it}, \mathbf{X}_{it}\}_{t=1}^{T}; i = 1, \ldots, |\mathcal{B}|\}$, we have the following approximation:*

$$\sum_{t=t_0}^{T} \mathbb{E}_{\mathbf{Z} \sim q_\phi(\cdot|\mathcal{D}_T)} \left[ \alpha(\mathbf{H}_t, W_t) \log p_\theta(Y_t \mid \mathbf{H}_t, W_t, \mathbf{Z}) \right] \approx -\frac{1}{|\mathcal{B}|} \sum_{t=t_0}^{T} \left\{ n_1^{(t)} R_{t,g}^{\omega=1}(f, \Phi) + n_0^{(t)} R_{t,g}^{\omega=0}(f, \Phi) \right\}, \quad (23)$$

*with*

$$R_{t,g}^{\omega}(f, \Phi) \approx -\frac{1}{n_\omega^{(t)}} \sum_{i \in \mathcal{B}, W_{it}=\omega} \mathbb{E}_{\mathbf{Z} \sim q_\phi(\mathbf{Z}|\mathcal{D}_{iT})} \left[ \alpha(\mathbf{H}_{it}, \omega) \log p_\theta(Y_{it} \mid \mathbf{H}_{it}, \omega, \mathbf{Z}) \right],$$

*where $n_\omega^{(t)}$ represents the number of instances in the batch $\mathcal{B}$ for which $W_{it} = \omega$.*

---

[2]Note that $p_\theta(\mathbf{H}_t, \mathbf{Z}) = p(\mathbf{Z} \mid \mathcal{D}_{\leq t-1}) p(\mathbf{H}_t)$ because $\mathbf{X}_t$ is d-separated from $\mathbf{Z}$ given $\mathcal{D}_{\leq t-1} = [Y_{\leq t-1}, \mathbf{X}_{\leq t-1}, W_{\leq t-1}]$.

As a result, maximizing the ELBO in Eq. (17) not only maximizes the weighted conditional likelihood over responses (the "reconstruction term") but also, as shown in Eq. (23), minimizes the weighted population risk. This, in turn, decreases the generalization bound in Theorem 11, reducing the error in treatment effect estimation. To further reduce this error, we include the IPM term from Theorem 11, regularizing the WELBO with the IPM term at each time step:

$$\mathcal{L}_{\text{IPM}}(\theta_\omega, \Phi) := \sum_{t=1}^{T} \text{IPM}_G \left( g_{\theta_\omega, \Phi}(\cdot \mid W_t = 1), g_{\theta_\omega, \Phi}(\cdot \mid W_t = 0) \right).$$

where $\theta_\omega$ are the parameters of the treatment network.

**Intuition: Stationarity of the inferred latent variables** We further justify the core assumption made in Proposition 12. The representation learning for the adjustment variables is achieved by approximating the posterior $p_\theta(\mathbf{z}|\mathcal{D}_T)$ with $q_\phi(\mathbf{z}|\mathcal{D}_T)$. Here, the missing baseline covariates are inferred by analyzing *all longitudinal data* $\{y_{\leq T}, \mathbf{x}_{\leq T}, \omega_{\leq T}\}$. However, since these covariates are *static* and *pre-response* variables, we should, ideally, be able to infer the exact substitute from a shorter longitudinal data $\{y_{\leq T'}, \mathbf{x}_{\leq T'}, \omega_{\leq T'}\}$ where $T' < T$, or from any temporal slice $\{y_{t_1:t_2}, \mathbf{x}_{t_1:t_2}, \omega_{t_1:t_2}\}$ with $t_2 > t_1$. Thus, ensuring a form of stationarity in our posterior approximation is crucial. To understand the importance of this property, consider estimating the Average Treatment Effect (ATE) for panel data at time $t$:

$$\tau_t := \mathbb{E}(Y_t(1) - Y_t(0)) = \mathbb{E}_{\mathbf{Z}, \mathbf{H_t}} \left[ \mathbb{E}_{Y_t|\mathbf{Z}, \mathbf{H_t}, W_t}[Y_t \mid \mathbf{Z}, \mathbf{H_t}, W_t = 1] - \mathbb{E}_{Y_t|\mathbf{Z}, \mathbf{H_t}, W_t}[Y_t \mid \mathbf{Z}, \mathbf{H_t}, W_t = 0] \right].$$

To compute $\tau_t$, we need to marginalize the conditional response over the joint distribution of covariates and latent adjustment variables, $p(\mathbf{h}_t, \mathbf{z}) = p(\mathbf{z} \mid \mathcal{D}_{\leq t})p(\mathbf{h}_t)$. Because we model $p_\theta(\mathbf{z}|\mathcal{D}_T)$, which depends on the *entire* history, there is, generally, no guarantee that $p(\mathbf{z} \mid \mathbf{h}_t) \approx p_\theta(\mathbf{z}|\mathcal{D}_T)$.

As a consequence of the stationarity assumption in Proposition 12, we introduce a penalty term in the form of the Wasserstein distance between consecutive posterior distributions $q_\phi(\mathbf{z}|\mathcal{D}_t)$ and $q_\phi(\mathbf{z}|\mathcal{D}_{t-1})$, penalizing significant variations as the data history grows. To preserve the model's capacity to capture time-varying dependencies, we begin regularization from $t_0 \gg 0$, typically $t_0 = \frac{T}{2}$:

$$\mathcal{L}_{\text{DistM}}(\phi) = \sum_{t=t_0}^{T} \mathbb{E}_{\mathcal{D}_t} W(q_\phi(\mathbf{z} \mid \mathcal{D}_t), q_\phi(\mathbf{z} \mid \mathcal{D}_{t-1})).$$

Assuming a Gaussian approximate posterior, the Wasserstein distance simplifies to

$$\mathcal{L}_{\text{DistM}}(\phi) = \sum_{t=t_0}^{T} \mathbb{E}_{\mathcal{D}_t} ||\mu_\mathbf{z}(\mathcal{D}_t) - \mu_\mathbf{z}(\mathcal{D}_{t-1})||_2^2 + \mathbb{E}_{\mathcal{D}_t} ||S_\mathbf{z}(\mathcal{D}_t) - S_\mathbf{z}(\mathcal{D}_{t-1})||_F^2,$$

where the penalty regularizes the first and second moments of the posterior distribution as the data history grows.

**Step 4: Total Loss Function and Training Strategy** The overall loss for the CDVAE model combines the ELBO from Eq. (17) with two additional terms: $\mathcal{L}_{\text{IPM}}$, which reduces covariate imbalance at each time step and addresses residual imbalance not corrected by weighting, and $\mathcal{L}_{\text{DistM}}$, which captures the global, static nature of adjustment variables. To push the Causal DVAE toward a near-deterministic regime, we treat the variance parameter $\sigma$ as learnable and fit it using the following loss function:

$$\mathcal{L}_{\text{tot}}(\theta, \theta_\omega, \phi, \Phi, \sigma) = -\text{ELBO}(\theta, \theta_\omega, \phi, \Phi, \sigma) + \lambda_{\text{IPM}}\mathcal{L}_{\text{IPM}}(\theta_\omega, \Phi) + \lambda_{\text{DistM}}\mathcal{L}_{\text{DistM}}(\phi). \quad (24)$$

Since the loss $\mathcal{L}_{\text{tot}}$ depends on the treatment parameters $\theta_\omega$, as the weighted ELBO is also a function of the propensity scores, we introduce an additional loss function $\mathcal{L}_W$, a binary cross-entropy loss to predict

treatment from confounders representation. To optimize CDVAE, we adopt an adversarial training strategy, simultaneously minimizing $\mathcal{L}_{\text{tot}}$ with respect to $\theta, \phi, \Phi$, and $\sigma$, while minimizing $\mathcal{L}_W$ with respect to $\theta_\omega$ (Algorithm 3, Appendix E):

$$\begin{cases} \min_{\theta,\phi,\Phi,\sigma} \mathcal{L}_{\text{tot}}(\theta, \theta_\omega, \phi, \Phi, \sigma), \\ \min_{\theta_\omega} \mathcal{L}_W(\theta_\omega, \Phi). \end{cases}$$

The adversarial nature of the training arises because, in one optimization step, we learn the representation $\Phi$ to predict the response while applying an IPM regularization to ensure balance between the weighted covariates. In the subsequent optimization step, for a fixed representation, we train the treatment classifier to predict the treatment. We use the Wasserstein distance as a specific case of IPM, with details on its computation provided in Algorithm 2, Appendix E.1. A computational complexity analysis of CDVAE is also provided in Appendix E.2.

**CDVAE: Model Specification**   We use a three-head neural network architecture for CDVAE to learn a shared representation between outcome and treatment models inspired by Shi et al. (2019). This architecture consists of two prediction heads for potential outcomes and one for treatment (Appendix G). We use a GRU (Cho et al., 2014) to learn a representation $\Phi(\mathbf{h}_{t+1})$ of the context history $\mathbf{h}_{t+1}$. The input of the outcome model $f_{\theta_y}$ is the shared representation $\Phi(\mathbf{h}_{t+1})$ and a sampled substitute $\mathbf{z}$ from $q_\phi(\mathbf{z} \mid \mathcal{D}_T)$. The outcome model $f_{\theta_y}$ is represented by two non-linear functions $f_{\theta_y^1}$ and $f_{\theta_y^0}$ for treatment assignments 1 and 0, respectively. The third head is a binary classifier $f_{\theta_\omega}$ estimating the propensity score. The inference model (encoder) $q_\phi(\mathbf{z} \mid y_{\leq T}, \mathbf{x}_{\leq T}, \omega_{\leq T})$ is modeled by a GRU, producing a hidden state that represents the sequence. Two non-linear mappings, $\mu_{\phi_2}$ and $\Sigma_{\phi_3}$, learn the mean and diagonal covariance matrix in the latent space.

## 5 Experiments

**Baselines**   In all our experiments, we compare CDVAE against the relevant baselines: RMSN, CRN, GNet, Causal Transformer, and Causal CPC. Each baseline is evaluated in three different settings: 1. The "base approach," where models are trained using only static and time-varying confounders. 2. The "substitute approach," where models are augmented with substitutes for the unobserved adjustment variables, obtained from CDVAE and represented by the mean of the approximated posterior. 3. The "oracle approach," where models are trained with the true adjustment variables.

**Hyperparameter Selection**   All models are fine-tuned using a grid search over hyperparameters, including architecture and optimizer settings. Model selection is based on the mean squared error (MSE) of factual outcomes on a validation set, which is also used as the criterion for early stopping. More details are in Appendix F.

**Adapting Baselines to Our Causal Settings**   All the considered baselines do not assume the observation of adjustment variables; however, they only consider time-varying and static confounders. It is not straightforward to incorporate adjustment variables into these models when observed. Naively augmenting their input with adjustment variables and treating them as static confounders often results in suboptimal performance or even worse outcomes compared to excluding the adjustment variables altogether. This is primarily due to two reasons: 1. Most baselines concatenate static confounders with time-varying ones at each time step. When high-dimensional adjustment variables are included, this dramatically increases the number of parameters, leading to overfitting. 2. When comparing different versions of the same baseline, it is important to ensure comparable complexity represented by the number of learnable parameters (Table 1). 3. Models such as CRN, Causal Transformer, and Causal CPC learn a balanced representation. Including adjustment variables naively as input may bias the learned representation towards these variables, as they are inherently balanced.

To ensure fair experimentation, we adapt the baselines as follows:

1. **CRN, Causal Transformer, and Causal CPC:** Instead of solely feeding the balanced representation to the outcome network, we augment it with static adjustment variables, similar to the CDVAE design, where the representation $\Phi(\mathbf{H}_t)$ is augmented with substitutes $\mathbf{Z}$.

2. **GNet:** Adjustment variables are only used to augment the representation fed to the outcome model, not the representation used to reconstruct the current covariates $\mathbf{X}_t$.

3. **RMSN:** Stabilized weights are learned solely as a function of $\mathbf{H}_t$, while the adjustment variables are concatenated with the representation of $\mathbf{H}_t$.

| Model | Params (k), synthetic data | Params (k), MIMIC-III data |
|---|---|---|
| CDVAE | 11.5 | 5.7 |
| Causal CPC (base) | 6.7 | 5.3 |
| Causal CPC (with substitute) | 7.1 | 5.7 |
| Causal CPC (oracle) | 14.3 | 7.5 |
| CT (base) | 18.7 | 16.3 |
| CT (with substitute) | 18.9 | 16.7 |
| CT (oracle) | 26 | 18.7 |
| G-Net (base) | 21.2 | 4.1 |
| G-Net (with substitute) | 21.5 | 4.3 |
| G-Net (oracle) | 28.5 | 5.7 |
| CRN (base) | 8.3 | 3.5 |
| CRN (with substitute) | 8.5 | 3.7 |
| CRN (oracle) | 14.2 | 4.9 |
| RMSN (base) | 14 | 6.7 |
| RMSN (with substitute) | 15.3 | 6.9 |
| RMSN (oracle) | 20.9 | 8.0 |

Table 1: Trainable parameters in thousands (k) for baselines in all configurations for the synthetic and semi-synthetic MIMIC-III data.

## 5.1 Synthetic Data Sets

**Generation** We simulate longitudinal data with a time length of $T = 75$ using an autoregressive model for both confounders and treatment assignments. The confounders $\mathbf{X}_t$ have a dimensionality $d_x = 100$. The outcome model includes unobserved adjustment variables $\mathbf{U}$, with dimensionality $d_u = 100$, generated from a Gaussian mixture model. A detailed description is provided in the Appendix C.1.1. The experiment is conducted with 5000 samples for training, 500 for validation, and 1000 for testing.

**Controlling the Effect of Unobserved Adjustment Variables** To assess the impact of unobserved variables $\mathbf{U}$, we vary the parameter $\gamma_{(1)}^{YU}$ (Appendix C.1.1), which determines the contribution of $\mathbf{U}$ in generating $Y_t(1)$. By design, $Y_t(1)$ increases with $\gamma_{(1)}^{YU}$, and the parameter $\gamma_{(1)}^{YU}$ is varied within the range $[0, 2.5]$ with a step size of 0.25, and we generate a longitudinal dataset at each level of $\gamma_{(1)}^{YX}$. We report the mean and standard deviation over 10 different seeds of PEHE in estimating the one-step-ahead ACATE, i.e., $\epsilon_{\mathrm{PEHE}_{T+1}}$, over the test individuals.

**Results** The Figure 3 shows the performances of all models given the three possible configurations. First, the error in estimating the ACATE increases substantially as the coefficient $\gamma_{(1)}^{YU}$ increases because the contribution of the unobserved static variables becomes considerably more important in the writing of the potential outcomes. However, CDVAE notably displays a slower increase in error and superior performance in all data configurations. Second, the errors decrease across all levels of unobserved heterogeneity when the baselines are provided with the substitutes learned features by CDVAE, highlighting the superiority of the substitute approach assisted by CDVAE over the base one. As expected, a baseline in the oracle approach generally performs better than with substitutes, especially at higher values of $\gamma_{(1)}^{YU}$, while the substitute approach provides near-oracle performance. Extended results are provided in Appendix C.2, which we complement by stressing performances by varying sequence length $T$ and number of simulated vitals $d_x$ in Appendix C.3.

## 5.2 Semi-Synthetic MIMIC-III Data

We adapt to our setting a semi-synthetic dataset constructed by Melnychuk et al. (2022) based on the MIMIC-III dataset (Johnson et al., 2016). The covariates are high-dimensional and include several vital measurements recorded in intensive care units. We investigate the effect of vasopressor treatments on blood pressure (response) for patients. The static covariates in the dataset (gender, ethnicity, and age) are included in the data construction as adjustment variables for the response. Detailed descriptions are provided in Appendix D.1. We conduct our experiments with 1400 patients for training, 200 patients for validation, and 400 patients for testing.

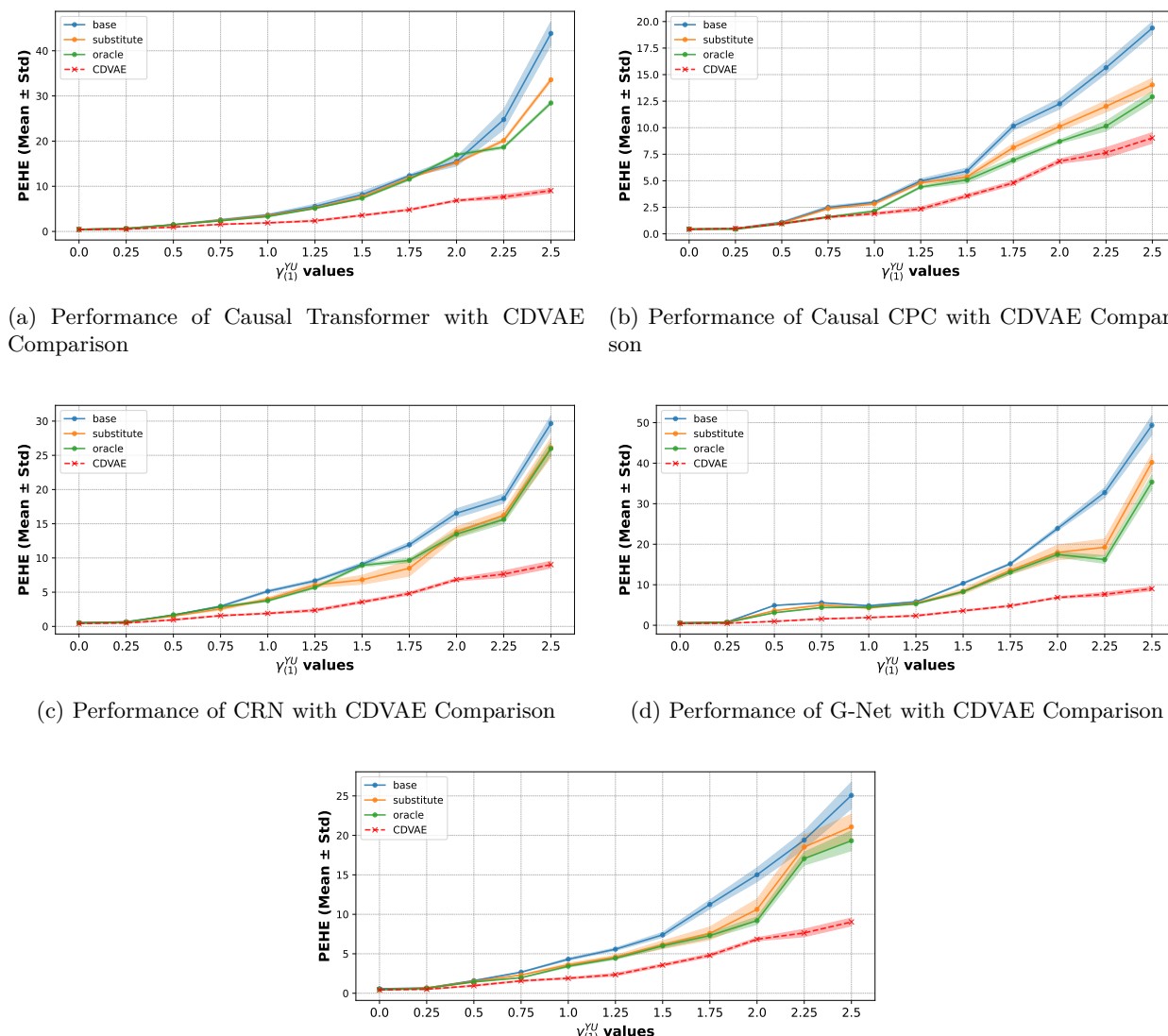

(a) Performance of Causal Transformer with CDVAE Comparison

(b) Performance of Causal CPC with CDVAE Comparison

(c) Performance of CRN with CDVAE Comparison

(d) Performance of G-Net with CDVAE Comparison

(e) Performance of RMSN with CDVAE Comparison

Figure 3: Evolution of PEHE in estimating ACATE for synthetic data across increasing levels of heterogeneity induced by adjustment variables **U**.

Similar to the synthetic data experiment, we evaluate all baselines using three approaches (base, with substitutes, and oracle), and report the PEHE in estimating the one-step-ahead ACATE for a patient trajectory.

**Results** Figure 4 reports the mean and standard deviation of PEHE for CDVAE and all baselines across all configurations. Similar to the synthetic data results, the base approach always performs worse than the substitute and oracle approaches. However, the substitute approach substantially enhances performance compared to the base approach, highlighting the relevance of the substitutes learned by CDVAE. Across all configurations, CDVAE achieves the best performance compared to all baselines. Extended results are provided in Appendix D.2.

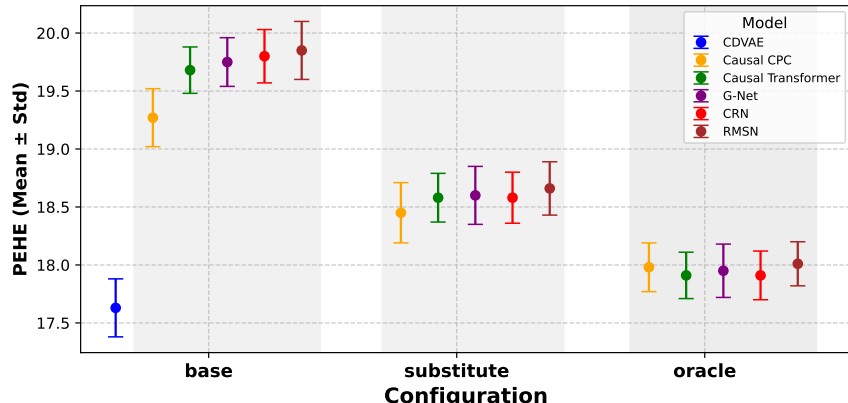

Figure 4: Results on the MIMIC-III data reported by PEHE and organized following the three possible configurations. Smaller is better.

## 6  Discussion

**Ablation Study**  To experimentally validate the design of CDVAE, we perform an ablation study by exploring several model configurations: the full CDVAE model with all components; CD-VAE without the IPM term ($\lambda_{\mathrm{IPM}} = 0$); CDVAE without the distribution matching term ($\lambda_{\mathrm{DistM}} = 0$); CDVAE without both the IPM term and distribution matching term ($\lambda_{\mathrm{IPM}} = 0, \lambda_{\mathrm{DistM}} = 0$); CDVAE without the IPM term and weighting ($\lambda_{\mathrm{IPM}} = 0$, w/o weighting); CDVAE ($\lambda_{\mathrm{IPM}} = 0, \lambda_{\mathrm{DistM}} = 0$, w/o weighting); and CDVAE with a fixed variance parameter ($\sigma = 1$).

We follow the same experimental protocol as in prior experiments, conducting the ablation for the synthetic dataset across all heterogeneity levels induced by $\gamma_{(1)}^{YU}$ and for the semi-synthetic MIMIC-III dataset. Across all configurations, the full CDVAE consistently outperforms the other models. Furthermore, removing individual components results in progressively higher errors, as shown in Tables 3 and 2. Interestingly, the configurations ($\lambda_{\mathrm{IPM}} = 0, \lambda_{\mathrm{DistM}} = 0$, without weighting) and ($\sigma = 1$) either perform comparably to or worse than the second-best model, Causal CPC, on both the synthetic dataset and the MIMIC-III dataset.

| CDVAE Configuration | PEHE |
|---|---|
| **Ours** | **17.63±0.25** |
| $\lambda_{\mathrm{IPM}} = 0$ | 18.38±0.26 |
| $\lambda_{\mathrm{DistM}} = 0$ | 18.21±0.26 |
| $\sigma = 1$ | 20.50±1.04 |
| $\lambda_{\mathrm{IPM}} = 0, \lambda_{\mathrm{DistM}} = 0$ | 18.45±0.23 |
| $\lambda_{\mathrm{IPM}} = 0$, **w/o weighting** | 18.97±0.28 |
| ($\lambda_{\mathrm{IPM}} = 0, \lambda_{\mathrm{DistM}} = 0$, **w/o weighting**) | 19.23±0.28 |

Table 2: CDVAE ablation study with semi-synthetic MIMIC-III data reported by PEHE. Smaller is better.

Table 3: Results on the synthetic data reported by PEHE. Smaller is better.

| CDVAE Configuration | $\gamma_{(1)}^{YU} = 0$ | $\gamma_{(1)}^{YU} = 0.25$ | $\gamma_{(1)}^{YU} = 0.5$ | $\gamma_{(1)}^{YU} = 0.75$ | $\gamma_{(1)}^{YU} = 1$ | $\gamma_{(1)}^{YU} = 1.25$ | $\gamma_{(1)}^{YU} = 1.5$ | $\gamma_{(1)}^{YU} = 1.75$ | $\gamma_{(1)}^{YU} = 2$ | $\gamma_{(1)}^{YU} = 2.25$ | $\gamma_{(1)}^{YU} = 2.5$ |
|---|---|---|---|---|---|---|---|---|---|---|---|
| **Ours** | **0.43±0.02** | **0.50±0.03** | **0.96±0.09** | **1.57±0.08** | **1.90±0.10** | **2.35±0.18** | **3.57±0.17** | **4.80±0.20** | **6.84±0.20** | **7.64±0.48** | **9.03±0.50** |
| $\lambda_{\mathrm{IPM}} = 0$ | 0.48±0.02 | 0.57±0.03 | 1.12±0.01 | 1.80±0.15 | 2.23±0.12 | 2.53±0.15 | 3.88±0.13 | 5.23±0.11 | 7.37±0.21 | 7.95±0.12 | 9.32±0.36 |
| $\lambda_{\mathrm{DistM}} = 0$ | 0.46±0.02 | 0.57±0.03 | 1.10±0.02 | 1.78±0.13 | 2.18±0.13 | 2.64±0.18 | 3.95±0.15 | 5.20±0.12 | 7.14±0.13 | 8.08±0.14 | 9.25±0.19 |
| $\sigma = 1$ | 0.81±0.40 | 0.66±0.07 | 2.17±0.53 | 5.73±1.51 | 9.27±3.27 | 10.67±2.02 | 12.66±2.16 | 14.28±4.23 | 15.63±4.04 | 18.23±5.71 | 17.19±5.71 |
| $\lambda_{\mathrm{IPM}} = 0, \lambda_{\mathrm{DistM}} = 0$ | 0.48±0.02 | 0.57±0.03 | 1.16±0.07 | 1.84±0.17 | 2.33±0.12 | 2.66±0.14 | 4.05±0.10 | 5.36±0.13 | 7.48±0.12 | 8.20±0.13 | 9.40±0.24 |
| $\lambda_{\mathrm{IPM}} = 0$, **w/o weighting** | 0.50±0.01 | 0.60±0.01 | 1.20±0.03 | 1.90±0.11 | 2.31±0.10 | 2.66±0.13 | 3.99±0.16 | 5.30±0.19 | 7.51±0.12 | 8.21±0.15 | 9.41±0.23 |
| $\lambda_{\mathrm{IPM}} = 0, \lambda_{\mathrm{DistM}} = 0$, **w/o weighting** | 0.50±0.02 | 0.62±0.02 | 1.27±0.03 | 1.98±0.10 | 2.39±0.09 | 2.85±0.14 | 4.19±0.13 | 5.59±0.13 | 7.61±0.15 | 8.41±0.14 | 9.78±0.21 |

**How CDVAE handles autoregressive order in practice.**  We require CMM($p$) to ensure identifiability of ACATE and validity of learned substitutes. In practice, we do not fix a lag $p$ but rely on two

mechanisms: 1. **Fading memory.** Recurrent predictors and fading-memory filters down-weight distant history exponentially. Classical results show that fading-memory maps can be uniformly approximated by finite-memory models: for any $\epsilon > 0$, there exists an order $p_\epsilon$ such that conditioning on the last $p_\epsilon$ lags is $\epsilon$-close to conditioning on the full history (Funahashi & Nakamura, 1993; Miller & Hardt, 2019; Gonon & Ortega, 2021). 2. **Bias from training.** Backpropagation through time biases RNNs toward short/medium context, with little gain from longer lags (Khandelwal et al., 2018).

To empirically substantiate these claims, we train CDVAE on synthetic data generated with $p_{\text{true}} = 10$. On test data, we compute counterfactual responses while masking outcomes to simulate CMM($p$) with $p \in \{4, 6, 8, 10, 12, 14, 16\}$. One-step residuals (conditioned on $\mathbf{H}_t$ and $\mathbf{Z}$) are tested for autocorrelation using Ljung–Box. Persistent autocorrelation or steadily improving accuracy with higher $p$ would suggest violation of CMM($p$). In all cases, Ljung–Box $p$-values up to lag 20 remain well above 0.05 (Table 5, e.g., $p = 4$: 0.085; $p = 10$: 0.109), so residual autocorrelation is not rejected. Meanwhile, one-step RMSE ($\sim 1.129$–$1.131$) and PEHE ($\sim 0.975$–$0.986$) are nearly flat across $p$ (Table 4). Thus, conditioning on short/medium windows suffices to remove serial dependence: increasing $p$ beyond 4–6 yields no material improvement. This agrees with (i) portmanteau tests showing no residual autocorrelation up to lag $h$, and (ii) the fading-memory approximation theorem together with the empirical bias of RNN training toward recent history (Table 6).

| Metric | $p = 4$ | $p = 6$ | $p = 8$ | $p = 10$ | $p = 12$ | $p = 14$ | $p = 16$ |
|---|---|---|---|---|---|---|---|
| $\text{RMSE}_{\text{1-step}}$ | 1.131 | 1.130 | 1.130 | 1.129 | 1.129 | 1.130 | 1.130 |
| $\text{PEHE}_{\text{1-step}}$ | 0.977 | 0.984 | 0.986 | 0.986 | 0.982 | 0.975 | 0.982 |

Table 4: One-step accuracy vs. masked autoregressive order $p$.

| Metric | $p = 4$ | $p = 6$ | $p = 8$ | $p = 10$ | $p = 12$ | $p = 14$ | $p = 16$ |
|---|---|---|---|---|---|---|---|
| $\min_{k \leq 20}$ p-value | 0.085 | 0.083 | 0.097 | 0.109 | 0.095 | 0.079 | 0.111 |
| # lags $\leq 20$ with $p < 0.05$ | 0 | 0 | 0 | 0 | 0 | 0 | 0 |
| Reject any? | No | No | No | No | No | No | No |

Table 5: Residual whiteness vs. masked order $p$ (Ljung–Box, joint up to $h = 20$).

| $p$ | Q(1) | p(1) | Q(5) | p(5) | Q(10) | p(10) | Q(20) | p(20) |
|---|---|---|---|---|---|---|---|---|
| 4 | 2.963 | 0.085 | 7.442 | 0.190 | 10.812 | 0.372 | 12.111 | 0.912 |
| 6 | 2.996 | 0.083 | 7.442 | 0.190 | 10.725 | 0.379 | 12.067 | 0.914 |
| 8 | 2.755 | 0.097 | 6.669 | 0.246 | 9.892 | 0.450 | 11.483 | 0.933 |
| 10 | 2.566 | 0.109 | 7.012 | 0.220 | 9.927 | 0.447 | 11.213 | 0.941 |
| 12 | 2.788 | 0.095 | 6.825 | 0.234 | 10.289 | 0.416 | 11.653 | 0.927 |
| 14 | 3.077 | 0.079 | 8.189 | 0.146 | 11.694 | 0.306 | 13.236 | 0.867 |
| 16 | 2.533 | 0.111 | 7.135 | 0.211 | 10.652 | 0.385 | 11.930 | 0.918 |

Table 6: Ljung–Box statistics (Q) and $p$-values at selected lags $k \in \{1, 5, 10, 20\}$.

**On the Near-Deterministic Behavior of CDVAE** For the theoretical results in Section 4.3 to hold, it is necessary to verify whether the variance parameter $\sigma$, as a learnable parameter, indeed decreases toward zero during training. In all experiments, $\sigma$ is initialized at one, and we show in Figure 5a its behavior during training on the synthetic dataset across all levels of $\gamma_{(1)}^{YU}$, as well as on the MIMIC-III dataset in Figure 5b. In all cases, the variance parameter stably decreases toward zero, validating the near-deterministic regime at the end of training. This supports the consistency result related to the treatment effect established in Theorem 9.

**CDVAE Sensitivity to unobserved confounding** To assess the sensitivity of CDVAE when sequential ignorability is violated and when adjustment variables $\mathbf{U}$ are unobserved, we mask four confounders from

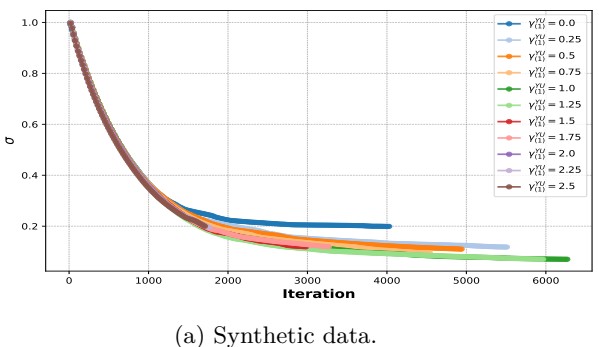 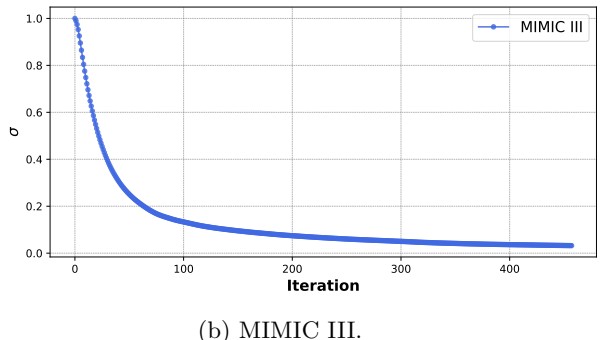

(a) Synthetic data.        (b) MIMIC III.

Figure 5: Evolution of variance parameter update during training for synthetic data (left) for each level of $\gamma_{(1)}^{YU}$ and MIMIC-III (right) averaged over 10 random initializations

the mimic experiment during model training, which are sodium, Glasgow Coma Scale total, cholesterol, and hemoglobin. We repeat the same experimental protocol as in the experiment of Figure 4 and show in Figure 7 the results of all models under the three configurations (base, substitute, and oracle). As expected, PEHE increases for all models across the three configurations, with an average increase of 2 points in PEHE, which is at least an 11% increase. CDVAE still outperforms baseline in base and substitutes configurations; however, baselines provided with true adjustment variables either provide comparable performances to CDVAE or slightly outperform (eg, CRN).

For synthetic data, we perform sensitivity analysis in the following way: We extend the simulator by partitioning covariates $\mathbf{X}_t = [\mathbf{X}_t^o, \mathbf{X}_t^h]$ into observed ones $\mathbf{X}_t^o$ and hidden ones $\mathbf{X}_t^h$ and summarizing the contribution of $\mathbf{X}_t^h$ into a hidden score. Sensitivity to unobserved confounding is controlled on the selection side by adding $\log(\Gamma)S_t$ to the treatment logit (Rosenbaum-$\Gamma$ style (Rosenbaum & Rubin, 1983)), and on the outcome side by arm-specific loadings $\beta_\omega S_t$ in the potential-outcome equations. Setting $\gamma = \beta_0 = \beta_1 = 0$ recovers the baseline DGP. Varying $\gamma$ yields families of datasets with controlled degrees of hidden selection bias and hidden effect modification by unobserved confounder (cf. Appendix C.1.2). Figure 6 shows how CDVAE, along with baselines in the "base" approach, degrades in performance (PEHE) as confounding level $\Gamma$ increases in $\{1, 1.5, \ldots, 5\}$. CDVAE still provides globally lower PEHE, but the performance margin narrows as $\Gamma$ increases.

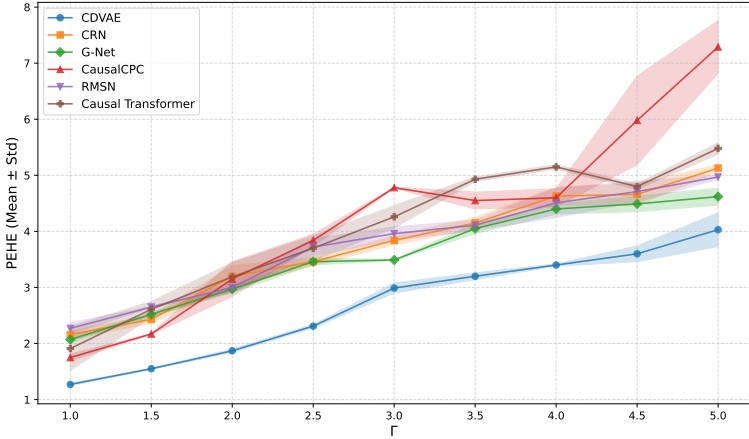

Figure 6: Sensitivity analysis on the synthetic data. We report the variation of PEHE under different levels of confounding $\Gamma$ for CDVAE and baselines under the "base" approach. Mean and standard deviation are computed for 10 different seeds. Smaller is better.

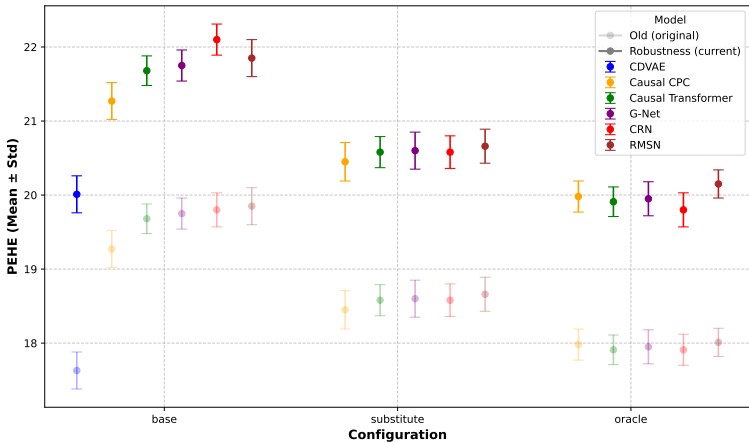

Figure 7: Results on the MIMIC-III data under violation of sequential ignorability. For comparison, results under sequential ignorability of Figure 4 are included in shadow. We report PEHE. Smaller is better.

**CDVAE Robustness to the Number of Components in the Prior** We demonstrate that the number of components $K$ in the Gaussian mixture prior does not substantially affect the PEHE. The chosen value, selected via random search, is $K = 7$, while the ground truth is $K = 8$. To test robustness, we evaluate CDVAE with values below and above this baseline, namely $K = 2, 5, 8, 11$. For the semi-synthetic MIMIC-III dataset, the true number of components is unknown, and the value selected via grid search is $K = 5$. To test robustness, we also evaluate $K = 2, 8, 11, 14$. Figure 9a shows the evolution of errors for the synthetic data across levels of $\gamma_{(1)}^{YU}$ for different values of $K$. While there are slight differences in performance means for CDVAE with varying $K$, almost all error bars overlap across all levels of $\gamma_{(1)}^{YU}$. The same observations hold for MIMIC-III, as depicted in Figure 9b. This highlights the robustness of CDVAE to the choice of $K$ for the prior and its low sensitivity compared to the components analyzed in the ablation study.

**Do the learned substitutes Z better capture the clustering structure in U (population structure)?** We assess this by computing the mutual information (MI) between learned representations and the true **U**-cluster labels across the sensitivity sweep $\Gamma \in \{1, 1.5, \ldots, 5\}$ (Figure 8). First, we notice that the MI for the baselines is positive. This is expected given the causal structure: the unobserved **U** affects the entire response series, and observed outcomes serve therefore as proxies for **U**, so encoders that use past responses inevitably preserve some cluster signal. CDVAE is more efficient because it learns a dedicated substitute **Z** (the variational posterior) and, under CMM($p$), explicitly exploits outcomes as high-signal proxies for **U**, thereby better capturing response heterogeneity driven by latent variables. The downward trend in baselines with larger $\Gamma$ is expected: we hold the contribution of **U** to $Y_t$ fixed while increasing $\Gamma$, which *amplifies the influence of missing time-varying confounders*, making outcomes less diagnostic about **U**. CDVAE nevertheless retains more MI because it learns a dedicated, sequence-level substitute $Z$ (near-deterministic variational posterior) that globally modulates the trajectory and is fed only to the outcome head, while baselines—though they inherit some MI via past outcomes acting as proxies—lack this explicit latent channel and therefore capture less of the population structure.

**Bayesian Model Assessment of CDVAE** We assess the quality of the conditional response fitting in our variational framework through a posterior predictive check (Rubin, 1984; Meng, 1994; Gelman et al., 1995) and similar to Bica et al. (2020b); Hatt & Feuerriegel (2024). For each time step $t$, let $\mathbf{y}_t^{\mathrm{obs}} \coloneqq \{y_{i,t}\}_{i=1}^{N_{\mathrm{val}}}$ represent the observed responses in the validation dataset. We generate $S$ replicated datasets of responses $\mathbf{y}_t^{\mathrm{rep}}(s) \coloneqq \{y_{i,t}^{\mathrm{rep}}(s)\}_{i=1}^{N_{\mathrm{val}}}$ for $s = 1, \ldots, S$, such that for each individual $i$ in the validation dataset, we draw $S$ samples of latent substitutes $\mathbf{z}_i(1), \ldots, \mathbf{z}_i(S) \sim q_\phi(\cdot \mid \mathcal{D})$. The approximate posterior serves as a proxy for the true (inaccessible) posterior $p(\mathbf{z} \mid \mathcal{D})$. Replicated responses are then generated from the fitted conditional distribution: $Y_t^{\mathrm{rep}}(s) \sim Y_t \mid y_{<t}, \mathbf{x}_{\leq t}, \omega_{\leq t}, \mathbf{z}_i(s)$.

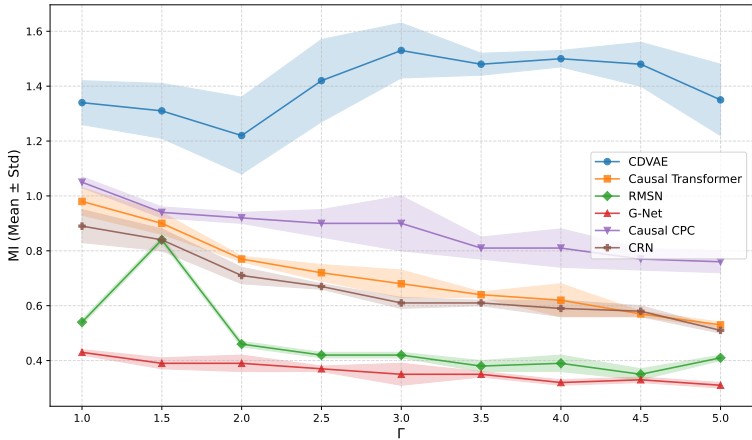

Figure 8: Mutual Information between learned representations and true cluster labels in $\mathbf{U}$ under a sensitivity analysis to missing confounding controlled by $\Gamma$. We report the mean and standard deviation over 10 different seeds. Higher is better.

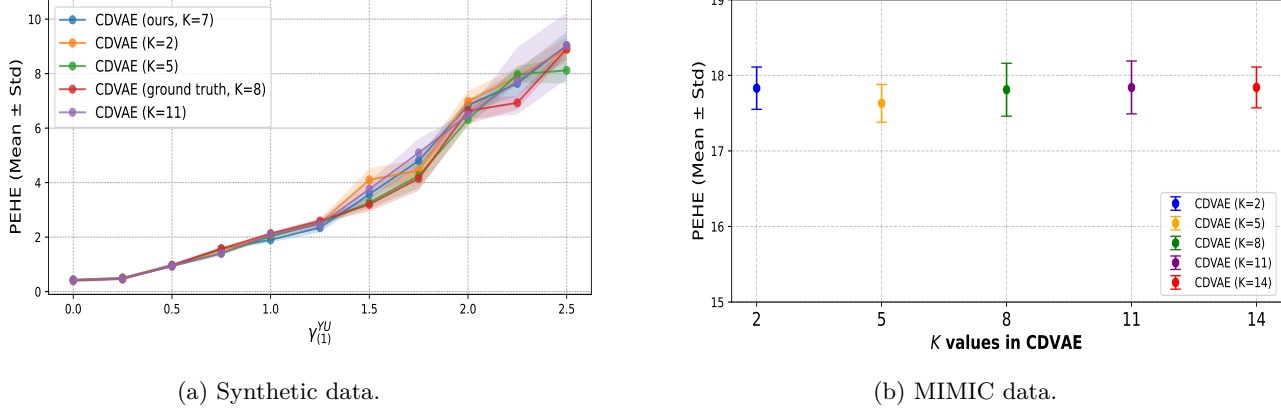

(a) Synthetic data.

(b) MIMIC data.

Figure 9: Results of CDVAE when varying the number of components $K$ of the prior for the synthetic data (left) and MIMIC-III (right) reported by PEHE. Smaller is better.

To compare the observed and replicated data, we define a statistic $\mathbb{T}$ based on the conditional log-likelihood:

$$\mathbb{T}(\mathbf{y}_t^{\text{obs}}) := \frac{1}{N_{\text{val}}} \sum_{i=1}^{N_{\text{val}}} \log p_\theta(y_{i,t} \mid \mathbf{h}_{it}, \omega_{i,t}, \mathbf{z}_i(s)),$$

and for replicated responses: $\mathbb{T}(\mathbf{y}_t^{\text{rep}}(s)) := \frac{1}{N_{\text{val}}} \sum_{i=1}^{N_{\text{val}}} \log p_\theta(y_{i,t}^{\text{rep}}(s) \mid \mathbf{h}_{it}, \omega_{i,t}, \mathbf{z}_i(s))$. We then define a posterior predictive p-value as:

$$p = \Pr(\mathbb{T}(\mathbf{y}_t^{\text{rep}}(s)) > \mathbb{T}(\mathbf{y}_t^{\text{obs}})),$$

which is approximated as:

$$p \approx \frac{1}{S} \sum_{s=1}^{S} \mathbb{1}\{\mathbb{T}(\mathbf{y}_t^{\text{rep}}(s)) > \mathbb{T}(\mathbf{y}_t^{\text{obs}})\}.$$

We compare these p-values over time to assess how closely the distribution of the conditional responses matches the distribution of the replicated responses. If the model accurately captures the conditional response distribution given the substitutes, the test statistics for the replicated data should be close to those for the observed data, ideally resulting in a p-value equal to 0.5. Figures 10a and 10b show the temporal evolution of

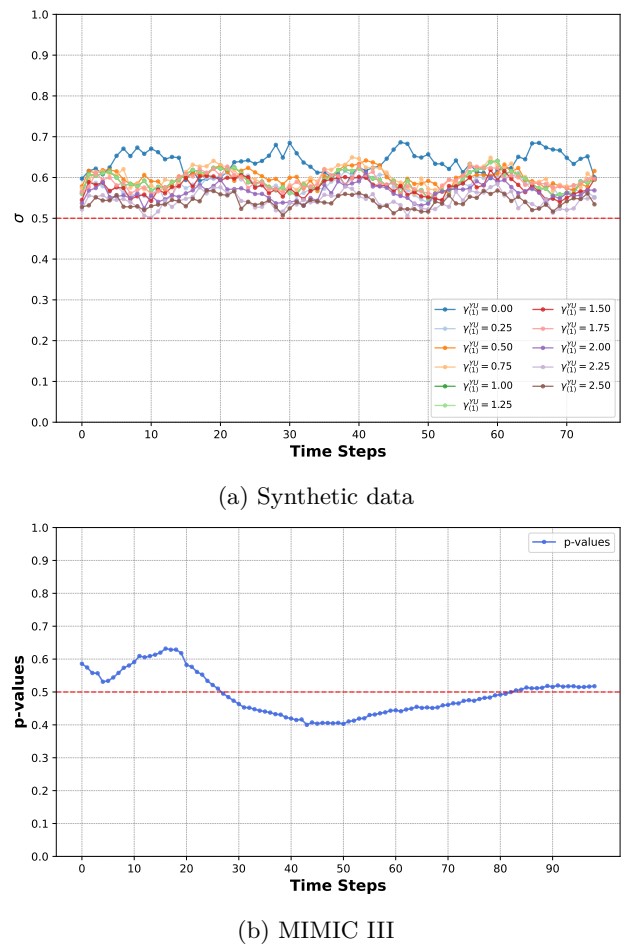

(a) Synthetic data

(b) MIMIC III

Figure 10: Evolution of CDVAE posterior predictive p-values for synthetic (top) and semi-synthetic MIMIC-III data (bottom). We report the average for 10 random initializations at each time step.

the p-values for both the synthetic data and semi-synthetic MIMIC-III. In general, all the p-values fluctuate inside the interval $[0.41, 0.59]$ except for the synthetic dataset with $\gamma^{YU}_{(1)} = 0.0$ because $\mathbf{U}$ no longer modifies the potential outcomes $(Y_t(1))_{t \geq 1}$. Interestingly, the bias in the p-values (Figure 10a) decreases as $\gamma^{YU}_{(1)}$ increases because CDVAE is designed to capture substantial heterogeneity in the responses due to the unobserved adjustment variables.

**Connection to the Deconfounder Theory** Our approach can also provide a remedy for a key inconsistency in the deconfounder theory (Lopez & Gutman, 2017; Ranganath & Perotte, 2018; Wang & Blei, 2019a;b). Crucially, while the inference model for the latent confounder is probabilistic, leading to a distribution over the substitute confounder, the theory relies on the assumption that the posterior distribution collapses into a Dirac delta distribution, implying that the confounder is estimated with certainty. This assumption simplifies the theoretical framework but introduces a fundamental incoherence between the stochastic nature of the inference models and the deterministic assumption in the theory (D'Amour, 2019b; Imai & Jiang, 2019; D'Amour, 2019a). There is a significant theoretical-application gap: In theory, the deconfounder assumes that the substitute confounder is estimated with certainty, while in practice, inference models are *stochastic* (Zhang et al., 2019; Bica et al., 2020b; Hatt & Feuerriegel, 2024), and the latent confounder is described by a distribution, introducing uncertainty. Our use of near-deterministic VAEs indirectly controls the posterior variance, finding a compromise where the posterior remains less diffuse yet not fully deterministic. This approach enables a more precise estimation of the substitute confounder without

assuming perfect certainty, addressing an inherent inconsistency. Applying this method within deconfounder theory holds independent interest beyond this paper's scope. Future research could explore integrating these improvements into the deconfounder framework, with near-deterministic VAEs for substitute confounders inference to bridge theory and practice.

**Comparing our approach to the deconfounder**    Unlike the deconfounder, our method does not require the consistency assumption for identifying treatment effects. Whereas the deconfounder framework models unobserved covariates as confounders affecting both treatment and outcome, we treat these unobserved covariates as adjustment variables, thus bypassing the need for the consistency assumption. A theoretical discussion of this distinction is in Appendix B.1. Furthermore, aligned with critiques of the deconfounder on the validity of inferred substitutes (D'Amour, 2019b; Imai & Jiang, 2019; D'Amour, 2019a; Ogburn et al., 2019), we investigate in Appendix B.2 whether the substitute adjustment variables $\mathbf{Z}$ might inadvertently capture "bad variables" that could bias causal effect estimation.

**On failure modes of CDVAE**    We now present the principal theoretical failure modes of CDVAE that may caution against its use in practice:

1. **Hidden confounding beyond $\mathbf{H}_t$.** We assume sequential ignorability; only *missing adjustment variables* $\mathbf{U}$ affect outcomes, but not treatments. If confounders are missing, identifiability fails. In a knife-edge case where these confounders are (i) time-invariant and (ii) satisfy the same CMM($p$) structure as $\mathbf{U}$, identifiability results (e.g., Theorem 4, Corollary 5, Theorem 6) might still hold. But this is implausible in practice, and crucially, our propensity model omits the latent variable $\mathbf{Z}$, introducing bias in treatment weighting.

2. **Time variation in the unobserved variables.** CDVAE treats $\mathbf{U}$ as *static* adjustment variables. If the true unobserved effect modifiers are time-varying, or if there are time-varying unobserved confounders, the latent substitute $\mathbf{Z}$ can no longer absorb all the relevant heterogeneity with a time-invariant representation, and both the identifiability argument and the sequential ignorability with augmented history may fail.

3. **Very long memory or infinite-order dependence.** Our identifiability results rely on a *finite* conditional Markov order $p$. If outcomes exhibit very long distributed-lag effects, or more strongly, if there is *no* finite order $m$ for which responses more than $m$ steps back are conditionally independent given current history $\mathbf{Z}$, then the process is effectively infinite-order. This violates CMM($p$) and undermines our ability to infer reliable substitutes for $\mathbf{U}$ from the outcome sequence, and ACATE may not be identifiable.

4. **Outcome support and heavy tails.** As a technical requirement, Theorem 6 assumes $\mathcal{Y}$ lies in a Borel subset of a compact interval. Heavy-tailed outcomes stretch this regularity condition, but it can often be mitigated by variance-stabilizing transforms or winsorization.

5. **Parameter nonstationarity/regime switching.** Even if a CMM($p$) holds structurally, our analysis presumes *fixed* parameters $\theta$. If the conditional response dynamics switch across regimes (e.g., latent or exogenous regime changes) so that, *even given* $\mathbf{Z}$, the effective parameters drift over time, the fixed-$\theta$ factorization in Assumption 3 breaks. A regime-switching or time-varying parameter extension would be needed; otherwise, causal validity and estimation stability can degrade.

**Conclusion**    In this paper, we proposed Causal DVAE, a novel framework for estimating treatment effects in high-dimensional, time-varying settings. By leveraging variational inference with robust regularization techniques, we introduced a principled approach to infer latent adjustment variables while ensuring identifiability and mitigating covariate imbalance. Our framework demonstrated theoretical guarantees for generalization and treatment effect consistency in the near-deterministic regime of VAEs—an overlooked property in the causal inference literature. Extensive empirical evaluations support the effectiveness of CDVAE on both synthetic and semi-synthetic datasets. Future work could focus on extending this framework to: 1. dynamic treatment regimes, for example, by incorporating the g-calculus (Robins, 1997; Vansteelandt & Joffe, 2014);

2. providing generalization bounds for treatment effects when they depend on a sequence of treatments (Lewis & Syrgkanis, 2021; Vankadara et al., 2022; Oh et al., 2022; Csillag et al., 2024); 3. exploring theoretically the bias-variance trade-off in the estimation of treatment effects in the near-deterministic regime.

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

# A    Proofs

## A.1    Identifiability: Treatment Effects and Unobserved Adjustment Variables

**Proof** [**CATE Identifiability**] Assuming the consistency, overlap, and ignorability assumptions (1, 3, 2), we demonstrate that the CATE, as defined in Eq. 2, is identifiable from the observed data distribution:

$$\tau_t = \mathbb{E}(Y_t|\mathbf{H}_t = \mathbf{h}_t, W_t = 1) - \mathbb{E}(Y_t|\mathbf{H}_t = \mathbf{h}_t, W_t = 0). \tag{25}$$

Establishing the identifiability of CATE requires showing that the two potential outcome expectations are identifiable. By the ignorability assumption (2), the potential outcome under treatment can be expressed as:

$$m_t^1(\mathbf{h}_t) = \mathbb{E}_{Y_t(1)|\mathbf{H}_t}(Y_t(1) \mid \mathbf{H}_t = \mathbf{h}_t) = \mathbb{E}_{Y_t(1)|\mathbf{H}_t, W_t}(Y_t(1) \mid \mathbf{H}_t = \mathbf{h}_t, W_t = 1).$$

Using the consistency assumption (1), the observed response can identify $Y_t(1)$ when conditioned on $W_t = 1$:

$$m_t^1(\mathbf{h}_t) = \mathbb{E}_{Y_t|\mathbf{H}_t, W_t}(Y_t \mid \mathbf{H}_t = \mathbf{h}_t, W_t = 1).$$

Similarly, we identify the expected potential outcome under no treatment:

$$m_t^0(\mathbf{h}_t) = \mathbb{E}_{Y_t|\mathbf{H}_t, W_t}(Y_t \mid \mathbf{H}_t = \mathbf{h}_t, W_t = 0).$$

The existence of these expectations is guaranteed by the overlap assumption. ∎

**Proof** [**Augmented CATE Identifiability**] Assuming that the adjustment variables $\mathbf{U}$ are observed, we demonstrate the identifiability of the Augmented CATE:

$$\tau_t(\mathbf{h}_t, \mathbf{u}) = \mathbb{E}(Y_t|\mathbf{H}_t = \mathbf{h}_t, \mathbf{U} = \mathbf{u}, W_t = 1) - \mathbb{E}(Y_t|\mathbf{H}_t = \mathbf{h}_t, \mathbf{U} = \mathbf{u}, W_t = 0).$$

The assumptions of consistency, overlap, and ignorability ensure that CATE is identifiable. Since $\mathbf{U}$ does not affect the treatment, the overlap assumption (3) remains valid. Further, $\mathbf{U}$ being independent of the treatment implies that the ignorability assumption holds when conditioning on $\mathbf{U}$:

$$Y_{it}(\omega) \perp\!\!\!\perp W_t|\mathbf{H}_t = \mathbf{h}_t \implies Y_{it}(\omega) \perp\!\!\!\perp W_t|\mathbf{H}_t = \mathbf{h}_t, \mathbf{U} = \mathbf{u} \quad \forall(\omega, \mathbf{h}_t, \mathbf{u}).$$

The remainder of the proof follows the CATE identifiability argument. ∎

**Proof** [**Theorem 6**] Let $\mathbf{Z}$ be a latent variable such that $\mathbf{Z} \sim CMM(p)$. Any static adjustment variables affecting all response series in the panel must be measurable with respect to $(\mathbf{Z}, \mathbf{H}_T)$. We assume weak regularity conditions on treatment and response domains.

**Assumption 13 (Regularity)** *The response domain $\mathcal{Y}$ is a Borel subset of a compact interval.*

The treatment domain $\mathcal{W} = \{0, 1\}$ is a Borel subset of $[0, 1]$. To prove the theorem, we need the following lemma:

**Lemma 14 (Kernels and Randomization (Kallenberg, 2021))** *Let $\mu$ be a probability kernel from a measurable space $S_1$ to a Borel space $S_2$. There exists a measurable function $f : S_1 \times [0, 1] \to S_2$ such that if $\vartheta$ is uniform on $[0, 1]$, then $f(s_1, \vartheta)$ is distributed as $\mu(s_1, \cdot)$.*

Suppose by contradiction the existence of $\mathbf{Z}\prime$ that is not measurable with respect to $(\mathbf{Z}, \mathbf{H}_T)$ and such that:

$$Y_{it}(\omega) \not\perp\!\!\!\perp \mathbf{Z_i}\prime | \mathbf{H}_{it}, \mathbf{Z}_i \quad \forall \omega, t$$

Let $t$ be an arbitrary time step in the panel data. By lemma 14, there exists a measurable function $f_t : \mathcal{H}_t \times \mathcal{W} \times \mathcal{Z} \times [0, 1] \to \mathcal{Y}$ such that:

$$Y_{it} = f_t(\mathbf{H}_{it}, W_{it}, \mathbf{Z}_i, \gamma_{it}), \quad \gamma_{it} \perp\!\!\!\perp (\mathbf{H}_{it}, W_{it}, \mathbf{Z}_i).$$

The conditional Markov property implies the independence of the following conditional distributions:

$$(Y_{it} \mid \mathbf{H}_{it}, \mathbf{Z}_i, W_{it} = \omega) \perp\!\!\!\perp (Y_{it\prime} \mid \mathbf{H}_{it\prime}, \mathbf{Z}_i, W_{it\prime} = \omega\prime).$$

Such that $t\prime$ verifies $|t - t\prime| > p$. We can thus conclude that:

$$(Y_{it}(\omega) \mid \mathbf{H}_{it}, \mathbf{Z}_i) \perp\!\!\!\perp (Y_{it\prime}(\omega\prime) \mid \mathbf{H}_{it\prime}, \mathbf{Z}_i).$$

Because :

$$(Y_{it}(\omega) \mid \mathbf{H}_{it}, \mathbf{Z}_i) = (Y_{it}(\omega) \mid \mathbf{H}_{it}, \mathbf{Z}_i, W_{it} = \omega)$$
$$= (Y_{it} \mid \mathbf{H}_{it}, \mathbf{Z}_i, W_{it} = \omega).$$

The first equality follows from the sequential ignorability of the theorem 4, and the second equality follows the consistency assumption. On the other hand, we also have from the CMM(p) that:

$$\gamma_{it\prime} \perp\!\!\!\perp Y_{it} \mid \mathbf{H}_{it}, \mathbf{Z}_i.$$

From which it follows using the fact that $\gamma_{it} \perp\!\!\!\perp (\mathbf{H}_{it}, W_{it}, \mathbf{Z}_i)$ and $Y_{it}(\omega) \mid \mathbf{H}_{it}, \mathbf{Z}_i = Y_{it} \mid \mathbf{H}_{it}, \mathbf{Z}_i, W_{it} = \omega$:

$$\gamma_{it\prime} \perp\!\!\!\perp Y_{it}(\omega) \mid \mathbf{H}_{it}, \mathbf{Z}_i$$

By using twice the lemma 14, we write:

$$\gamma_{it\prime} = h_t(\mathbf{Z}\prime_i, \eta_{it\prime}), \quad \eta_{it\prime} \perp\!\!\!\perp \mathbf{Z}\prime_i \tag{26}$$

And:

$$Y_{it}(\omega) = g_t(\mathbf{Z}\prime_i, \epsilon_{it}), \quad \epsilon_{it} \perp\!\!\!\perp \mathbf{Z}\prime_i. \tag{27}$$

Since $\mathbf{Z}\prime_i$ is not measurable with respect to $(\mathbf{Z}_i, \mathbf{H}_{it})$, then by equation 26 and equation 27:

$$\gamma_{it\prime} \not\perp\!\!\!\perp Y_{it} \mid \mathbf{H}_{it}, \mathbf{Z}_i$$

We have thus a contradiction. ∎

## A.2 Derivation of CDVAE Loss

**Proof [ELBO]** We provide proof that the bound in Eq. (12) is indeed the Evidence Lower Bound (ELBO) for the weighted conditional log-likelihood.

By the concavity of the logarithm, for every $t \in \{1, 2, \ldots, T\}$, we have:

$$\log p_\theta\left(y_t \mid \mathbf{h}_t, \omega_t\right) \geq \underbrace{\mathbb{E}_{\mathbf{Z}, C \sim q_\phi(\cdot, \cdot \mid \mathcal{D}_T)}\left[\log \frac{p_\theta\left(y_t, \mathbf{Z}, C \mid \mathbf{h}_t, \omega_t\right)}{q_\phi(\mathbf{Z}, C \mid \mathcal{D}_T)}\right]}_{(*)} \tag{28}$$

We use the identity $p_\theta(y_t, \mathbf{z}, c \mid \mathbf{h}_t, \omega_t) = p_\theta(y_t \mid \mathbf{z}, \mathbf{h}_t, \omega_t) p(\mathbf{z}, c)$, and the factorization of the approximate posterior $q_\phi(\mathbf{z}, c \mid \mathcal{D}_T) = q_{\phi_z}(\mathbf{z} \mid \mathcal{D}_T) q_{\phi_c}(c \mid \mathcal{D}_T)$, to write:

$$
\begin{aligned}
(*) &= \mathbb{E}_{\mathbf{Z}, C \sim q_\phi(\cdot, \cdot \mid \mathcal{D}_T)}\left[\log \frac{p_\theta(y_t \mid \mathbf{Z}, \mathbf{h}_t, \omega_t) \cdot p(C \mid \mathbf{Z}) p(\mathbf{Z})}{q_{\phi_z}(\mathbf{Z} \mid \mathcal{D}_T) \cdot q_{\phi_c}(C \mid \mathcal{D}_T)}\right] \\
&= \mathbb{E}_{\mathbf{Z}, C \sim q_\phi(\cdot, \cdot \mid \mathcal{D}_T)}\left[\log p_\theta(y_t \mid \mathbf{Z}, \mathbf{h}_t, \omega_t)\right] - \underbrace{D_{KL}(q_{\phi_z}(\mathbf{Z} \mid \mathcal{D}_T) q_{\phi_c}(C \mid \mathcal{D}_T) \parallel p(C \mid \mathbf{Z}) p(\mathbf{Z}))}_{(**)}
\end{aligned}
$$

The KL divergence term $(**)$ between the joint approximate posterior and the prior can be decomposed into the sum of two KL divergences:

$$
\begin{aligned}
(**) &= \sum_{c=1}^{K} \mathbb{E}_{\mathbf{Z} \sim q_\phi(\cdot, c \mid \mathcal{D}_T)} \log\left(\frac{p(c \mid \mathbf{Z}) p(\mathbf{Z})}{q_{\phi_z}(\mathbf{Z} \mid \mathcal{D}_T) q_{\phi_c}(C \mid \mathcal{D}_T)}\right) \\
&= \sum_{c=1}^{K} \mathbb{E}_{\mathbf{Z} \sim q_\phi(\cdot, c \mid \mathcal{D}_T)} \log\left(\frac{p(\mathbf{Z})}{q_{\phi_z}(\mathbf{Z} \mid \mathcal{D}_T)}\right) + \sum_{c=1}^{K} \mathbb{E}_{\mathbf{Z} \sim q_\phi(\cdot, c \mid \mathcal{D}_T)} \log\left(\frac{p(c \mid \mathbf{Z})}{q_{\phi_c}(C \mid \mathcal{D}_T)}\right) \\
&= D_{KL}\left(q_{\phi_z}(\mathbf{z} \mid \mathcal{D}_T) \parallel p(\mathbf{z})\right) + \mathbb{E}_{\mathbf{Z} \sim q_{\phi_z}(\cdot \mid \mathcal{D}_T)} D_{KL}\left(q_{\phi_c}(c \mid \mathcal{D}_T) \parallel p(c \mid \mathbf{Z})\right)
\end{aligned}
$$

We can now define the individual ELBO by performing a weighted sum over the log-likelihood terms and marginalizing over the full longitudinal data distribution:

$$L = \sum_{t=1}^{T} \mathbb{E}_{\mathcal{D}_T}\left[\alpha(\mathbf{H}_t, W_t) \log p_\theta(Y_t \mid \mathbf{H}_t, W_t)\right] \geq \mathbb{E}_{\mathcal{D}_T} \underbrace{\sum_{t=1}^{T} \mathbb{E}_{\mathbf{Z}, C \sim q_\phi(\cdot, \cdot \mid \mathcal{D}_T)}\left[\alpha(\mathbf{h}_t, \omega_t) \log \frac{p_\theta\left(y_t, \mathbf{Z}, C \mid \mathbf{h}_t, \omega_t\right)}{q_\phi(\mathbf{Z}, C \mid \mathcal{D}_T)}\right]}_{\text{ELBO}_0(\mathcal{D}_T; \theta, \phi)}$$

Given the developed expressions for $(*)$ and $(**)$, we can express $\text{ELBO}_0(\mathcal{D}_T; \theta, \phi)$ as:

$$\text{ELBO}_0(\mathcal{D}_T; \theta, \phi) = \sum_{t=1}^{T} \mathbb{E}_{\mathbf{Z} \sim q_{\phi_z}(\cdot \mid \mathcal{D}_T)}\left[\alpha(\mathbf{h}_t, \omega_t) \log p_\theta(y_t \mid \mathbf{h}_t, \omega_t, \mathbf{Z})\right] \tag{29}$$

$$- \left(\sum_{t=1}^{T} \alpha(\mathbf{h}_t, \omega_t)\right) \left\{D_{KL}(q_{\phi_z}(\mathbf{z} \mid \mathcal{D}_T) \parallel p(\mathbf{z})) + \mathbb{E}_{\mathbf{Z} \sim q_{\phi_z}(\cdot \mid \mathcal{D}_T)} D_{KL}(q_{\phi_c}(c \mid \mathcal{D}_T) \parallel p(c \mid \mathbf{Z}))\right\}.$$

The gap in our variational approximation is defined as the difference between the true weighted log-likelihood and the ELBO:

$$\mathbb{E}_{\mathcal{D}_T} \Delta_0(\mathcal{D}_T; \theta, \phi) \coloneqq L - \mathbb{E}_{\mathcal{D}_T} \text{ELBO}_0(\mathcal{D}_T; \theta, \phi)$$

The per-time-step gap in Eq. (28) can be rewritten as:

$$
\begin{aligned}
\log p_\theta\left(y_t \mid \mathbf{h}_t, \omega_t\right) - (*) &= \mathbb{E}_{\mathbf{Z}, C \sim q_\phi(\cdot, \cdot \mid \mathcal{D}_T)}\left[\log \frac{p_\theta\left(y_t \mid \mathbf{h}_t, \omega_t\right) q_\phi(\mathbf{Z}, C \mid \mathcal{D}_T)}{p_\theta\left(y_t, \mathbf{Z}, C \mid \mathbf{h}_t, \omega_t\right)}\right] \\
&= \mathbb{E}_{\mathbf{Z}, C \sim q_\phi(\cdot, \cdot \mid \mathcal{D}_T)}\left[\log \frac{q_\phi(\mathbf{Z}, C \mid \mathcal{D}_T)}{p_\theta\left(\mathbf{Z}, C \mid \mathcal{D}_t\right)}\right] \\
&= D_{KL}\left(q_\phi(\mathbf{Z}, C \mid \mathcal{D}_T) \parallel p_\theta\left(\mathbf{Z}, C \mid \mathcal{D}_t\right)\right)
\end{aligned}
$$

This last equation holds because

$$p_\theta(y_t, \mathbf{Z}, C \mid \mathbf{h}_t, \omega_t) = p_\theta(\mathbf{Z}, C \mid y_t, \mathbf{h}_t, \omega_t) \, p_\theta(y_t \mid \mathbf{h}_t, \omega_t),$$

and $\{y_t, \mathbf{h}_t, \omega_t\} = \{y_{\leq t}, \mathbf{x}_{\leq t}, \omega_{\leq t}\} = \mathcal{D}_t$. The individual gap is thus:

$$\Delta_0(\mathcal{D}_T; \theta, \phi) = \sum_{t=1}^{T} \alpha(\mathbf{h}_t, \omega_t) D_{KL}\left(q_\phi(\mathbf{Z}, C \mid \mathcal{D}_T) \parallel p_\theta(\mathbf{Z}, C \mid \mathcal{D}_t)\right)$$

$\blacksquare$

### A.3 Transfer of Ignorability under Invertible Maps

**Proposition 15** *Let $\Phi$ be an invertible representation function. Then, $Y_t(\omega) \perp\!\!\!\perp W_t|\Phi(\mathbf{H}_t)$ holds if and only if $Y_t(\omega) \perp\!\!\!\perp W_t|\mathbf{H}_t$. Moreover, $p(W_t = \omega|\Phi(\mathbf{h}_t)) > 0$ holds if and only if $p(W_t = \omega|\mathbf{h}_t) > 0$.*

**Proof** Assume $Y_t(\omega) \perp\!\!\!\perp W_t|\mathbf{H}_t$. For a non-invertible $\Phi$, we have, letting $\Phi^{-1}(\mathbf{r}) = \{\mathbf{h}_t : \Phi(\mathbf{h}_t) = \mathbf{r}\}$:

$$
\begin{aligned}
p(Y_t(\omega) \mid \omega_t, \mathbf{r}) &= \frac{\int_{\mathbf{h}_t \in \Phi^{-1}(\mathbf{r})} p(Y_t(\omega) \mid \omega_t, \mathbf{h}_t) p(\mathbf{h}_t \mid \omega_t) d\mathbf{h}_t}{\int_{\mathbf{h}_t \in \Phi^{-1}(\mathbf{r})} p(\mathbf{h}_t \mid \omega_t) d\mathbf{h}_t} \\
&= \frac{\int_{\mathbf{h}_t \in \Phi^{-1}(\mathbf{r})} p(Y_t(\omega) \mid \mathbf{h}_t) p(\mathbf{h}_t \mid \omega_t) d\mathbf{h}_t}{\int_{\mathbf{h}_t \in \Phi^{-1}(\mathbf{r})} p(\mathbf{h}_t \mid \omega_t) d\mathbf{h}_t},
\end{aligned}
\tag{30}
$$

where ignorability implies that, for general $\Phi$, $p(Y_t(\omega) \mid \omega_t, \mathbf{r}) \neq p(Y_t(\omega) \mid \mathbf{r})$. For an invertible $\Phi$, however, $p(Y_t(\omega) \mid \omega_t, \Phi(\mathbf{h}_t)) = p(Y_t(\omega) \mid \Phi(\mathbf{h}_t))$. Similarly, $p(W_t = \omega|\Phi(\mathbf{h}_t)) = p(W_t = \omega|\mathbf{h}_t)$. $\blacksquare$

### A.4 CDVAE in the Near-Deterministic Regime

**Proof** [Theorem 8]

**Part 1**: $lim_{s \to +\infty} p_{\theta_s}(y_{\leq T} \mid \mathbf{x}_{\leq T}, \omega_{\leq T}) = p(y_{\leq T} \mid \mathbf{x}_{\leq T}, \omega_{\leq T})$.

Define the conditional cumulative distribution function (CDF) of the response given covariates, for each $t$ as:

$$F_t(y_t \mid \mathbf{h}_t, \omega_t) = \int_{-\infty}^{y_t} p(y_t' \mid \mathbf{h}_t, \omega_t) dy_t'.$$

Define the mapping $F : \mathcal{Y}^T \to [0,1]^T$ as:

$$F(y_1, \ldots, y_T \mid \mathbf{x}_{\leq T}, \omega_{\leq T}) \coloneqq [F_1(y_1 \mid \mathbf{h}_1, \omega_1), \ldots, F_T(y_T \mid \mathbf{h}_T, \omega_T)].$$

The differential of the mapping is given by:

$$dF(y_1, \ldots, y_T \mid \mathbf{x}_{\leq T}, \omega_{\leq T}) = p(y_{\leq T} \mid \mathbf{x}_{\leq T}, \omega_{\leq T}) dy_{\leq T}.$$

Now, define the following mapping for the latent variables using the Darmois construction (Hyvärinen & Pajunen, 1999):

$$G_i(z_i \mid z_1, \ldots, z_{i-1}) = \int_{-\infty}^{z_i} p(z_i' \mid z_1, \ldots, z_{i-1}) dz_i'.$$

We then define the mapping $G : \mathbb{R}^{d_\mathbf{z}} \to [0,1]^T$ such that:

$$G(\mathbf{z}) \coloneqq [G_1(z_1), G_2(z_2 \mid z_1), \ldots, G_i(z_\mathbf{z} \mid z_1, \ldots, z_{d_\mathbf{z}-1})].$$

Since $d_{\mathbf{z}} \leq T$ by assumption, we can trivially augment the mapping $G$ to $\tilde{G}$ so that the image is defined on $[0,1]^{d_{\mathbf{z}}}$, i.e., $\tilde{G} : \mathbb{R}^{d_{\mathbf{z}}} \to [0,1]^T$ such that: $\tilde{G}(\mathbf{z}) := [G(\mathbf{z}), \underbrace{0, \ldots, 0}_{T-d_{\mathbf{z}}}]$. The differential is given by: $d\tilde{G}(\mathbf{z}) = p(\mathbf{z})d\mathbf{z}$.

We now define the mean and variance of the encoder as follows:

$$f_t(\mathbf{h}_t, \omega_t, \mathbf{z}; \theta_s^*) := [F^{-1}(\tilde{G}(\mathbf{z}) \mid \mathbf{x}_{\leq T}, \omega_{\leq T})]_t, \quad \sigma_s^* = \frac{1}{\sqrt{s}}.$$

There is a consistency issue to address with this definition. First, observe that the function $f_t(.; \theta_s^*)$ takes as input only data up to time step $t$, but the inverse of the cumulative CDF is defined given *the whole sequence* of $(\mathbf{x}_{\leq T}, \omega_{\leq T})$. We therefore need to verify the following lemma which holds for our definition of $f_t(.; \theta_s^*)$.

**Lemma 16** *Let $(y_{\leq T}, \mathbf{x}_{\leq T}, \omega_{\leq T})$ and $(y'_{\leq T}, \mathbf{x}'_{\leq T}, \omega'_{\leq T})$ be two distinct realizations of repeated measurements such that there exists $t_0$ for which there exist two identical sub-trajectories $\mathbf{x}_{\leq t_0} = \mathbf{x}'_{\leq t_0}$ and $\omega_{\leq t_0} = \omega'_{\leq t_0}$. Then, for every $t \leq t_0$, we have:*

$$[F^{-1}(\tilde{G}(\mathbf{z}) \mid \mathbf{x}_{\leq t}, \omega_{\leq t})]_t = [F^{-1}(\tilde{G}(\mathbf{z}) \mid \mathbf{x}'_{\leq t}, \omega'_{\leq t})]_t.$$

Now, we decompose the marginal probabilistic model:

$$\begin{aligned}
p_{\theta_s}(y_{\leq T} \mid \mathbf{x}_{\leq T}, \omega_{\leq T}) &= \int_{\mathbb{R}^{d_{\mathbf{z}}}} p_{\theta_s}(y_{\leq T}, \mathbf{z} \mid \mathbf{x}_{\leq T}, \omega_{\leq T})d\mathbf{z} \\
&= \int_{\mathbb{R}^{d_{\mathbf{z}}}} \prod_{t=1}^T p_{\theta_s}(y_t, \mathbf{z} \mid y_{<t}, \mathbf{x}_{\leq t}, \omega_{\leq t})p(\mathbf{z})d\mathbf{z} \\
&= \int_{\mathbb{R}^{d_{\mathbf{z}}}} \prod_{t=1}^T \mathcal{N}\left(y_t \mid [F^{-1}(\tilde{G}(\mathbf{z}) \mid \mathbf{x}_{\leq T}, \omega_{\leq T})]_t, (\sigma_s^*)^2\right) p(\mathbf{z})d\mathbf{z} \\
&= \int_{[0,1]^T} \prod_{t=1}^T \mathcal{N}\left(y_t \mid [F^{-1}(\xi \mid \mathbf{x}_{\leq T}, \omega_{\leq T})]_t, (\sigma_s^*)^2\right) d\xi \\
&= \int_{\mathcal{Y}^T} \prod_{t=1}^T \mathcal{N}\left(y_t \mid [y'_{\leq T}]_t, (\sigma_s^*)^2\right) p(y'_{\leq T} \mid \mathbf{x}'_{\leq T}, \omega'_{\leq T})dy'_{\leq T}.
\end{aligned}$$

Finally, we have:

$$\lim_{s \to +\infty} \int_{\mathcal{Y}^T} \prod_{t=1}^T \mathcal{N}\left(y_t \mid [y'_{\leq T}]_t, (\sigma_s^*)^2\right) p(y'_{\leq T} \mid \mathbf{x}'_{\leq T}, \omega'_{\leq T})dy'_{\leq T} = \int_{\mathcal{Y}^T} \prod_{t=1}^T \delta(y_t - y'_t)p(y'_{\leq T} \mid \mathbf{x}'_{\leq T}, \omega'_{\leq T})dy'_{\leq T}$$
$$= p(y_{\leq T} \mid \mathbf{x}_{\leq T}, \omega_{\leq T}).$$

**Part 2**: Proof of $\lim_{s \to +\infty} \Delta(\mathcal{D}_T; \theta_s, \phi_s) = 0$.
First, we give an explicit writing of the modified individual gap:

$$\begin{aligned}
\Delta(\mathcal{D}_T; \theta_s, \phi_s) &= L(\mathcal{D}_T; \theta_s, \phi_s) - \mathrm{ELBO}_0(\mathcal{D}_T; \theta_s, \phi_s) + \mathrm{ELBO}_0(\mathcal{D}_T; \theta_s, \phi_s) - \mathrm{ELBO}(\mathcal{D}_T; \theta_s, \phi_s) \\
&= \Delta_0(\mathcal{D}_T; \theta_s, \phi_s) + \mathrm{ELBO}_0(\mathcal{D}_T; \theta_s, \phi_s) - \mathrm{ELBO}(\mathcal{D}_T; \theta_s, \phi_s) \\
&\quad - \left[\sum_{t=1}^T \alpha(\mathbf{h}_t, \omega_t)\right] \mathbb{E}_{\mathbf{Z} \sim q_{\phi_s}(\cdot \mid \mathcal{D}_T)} D_{KL}(q_{\phi_s}(c \mid \mathcal{D}_T) \parallel p(c \mid \mathbf{Z})) \\
&\quad - \left[\sum_{t=1}^T \alpha_s(\mathbf{h}_t, \omega_t)\right] \log Z(q_{\phi_s}(\cdot \mid \mathcal{D}_T)) \\
&= \sum_{t=1}^T \alpha(\mathbf{h}_t, \omega_t) D_{KL}(q_{\phi_s}(\mathbf{z}, c \mid \mathcal{D}_T) \parallel p_{\theta_s}(\mathbf{z}, c \mid \mathcal{D}_t)).
\end{aligned}$$

The last equality holds because we assume $q_{\phi_s}(c \mid \mathcal{D}_T) = \pi_{\phi_s}(c \mid \mathcal{D}_T)$ and $\pi_{\phi_s}(c \mid \mathcal{D}_T)$ is a minimizer of

$$\min_{q_{\phi_s}(c|\mathcal{D}_T)} \mathbb{E}_{\mathbf{Z} \sim q_{\phi_s}(\cdot | \mathcal{D}_T)} D_{KL}(q_{\phi_s}(c \mid \mathcal{D}_T) \parallel p(c \mid \mathbf{Z})) = -\log Z(q_{\phi_s}(\cdot \mid \mathcal{D}_T)).$$

To show that $\lim_{s \to +\infty} \Delta(\mathcal{D}_T; \theta_s, \phi_s) = 0$, we will define $\phi_s$ such that for every $c \in \{1, \dots, K\}$ and $t \in \{1, \dots, T\}$:

$$\lim_{s \to +\infty} D_{KL}(q_{\phi_s}(\mathbf{z}, c \mid \mathcal{D}_T) \parallel p_{\theta_s}(\mathbf{z}, c \mid \mathcal{D}_t)) = 0.$$

Let us define $q_{\phi_s}(\mathbf{z} \mid \mathcal{D}_T)$ such that

$$f_{\mu_\mathbf{z}}(\mathcal{D}_T, \phi_s) = G^{-1}(F(y_1, \dots, y_{d_\mathbf{z}}] \mid \mathbf{x}_{\leq T}, \omega_{\leq T}))$$

and

$$f_{S_\mathbf{z}}(\mathcal{D}_T, \phi) = \sigma_s \sqrt{\tilde{\Sigma}_\mathbf{z}(\mathcal{D}_T, \phi)},$$

with $\tilde{\Sigma}_\mathbf{z}(\mathcal{D}_T, \phi)$ being the inverse of:

$$\mathrm{Jac}(F^{-1}(. \mid \mathbf{x}_{\leq T}, \omega_{\leq T}) \circ \tilde{G}(\mathbf{z})) \mathrm{Jac}(F^{-1}(. \mid \mathbf{x}_{\leq T}, \omega_{\leq T}) \circ \tilde{G}(\mathbf{z}))^\top.$$

Using Bayes' rule, we can write the true posterior as:

$$p_{\theta_s}(\mathbf{z}, c \mid \mathcal{D}_t) = \frac{p_{\theta_s}(y_{\leq T}, \mathbf{z}, c \mid \mathbf{x}_{\leq t}, \omega_{\leq t})}{p_{\theta_s}(y_{\leq t} \mid \mathbf{x}_{\leq t}, \omega_{\leq t})} = \frac{\prod_{l=1}^{t} \mathcal{N}\left(y_l \mid [F^{-1}(\tilde{G}(\mathbf{z}) \mid \mathbf{x}_{\leq T}, \omega_{\leq T})]_l, \sigma_s^2\right) p(\mathbf{z} \mid c) p(c)}{p_{\theta_s}(y_{\leq t} \mid \mathbf{x}_{\leq t}, \omega_{\leq t})}.$$

By the GMM assumption over the prior, we have $p(\mathbf{z} \mid c) = \mathcal{N}(\mathbf{z} \mid \mu_c, \Sigma_c)$. The approximate posterior is of the form:

$$q_{\phi_s}(\mathbf{z}, c \mid \mathcal{D}_T) = q_{\phi_s}(\mathbf{z} \mid \mathcal{D}_T) \pi_{\phi_s}(c \mid \mathcal{D}_T) = \mathcal{N}(\mathbf{z} \mid f_{\mu_\mathbf{z}}(\mathcal{D}_T, \phi_s), \sigma_s^2 \tilde{\Sigma}_\mathbf{z}(\mathcal{D}_T, \phi)) \pi_{\phi_s}(c \mid \mathcal{D}_T).$$

We now show convergence by performing a change of variables. Define $\mathbf{z}' = \sigma_s^{-\frac{t}{d_z}}(\mathbf{z} - \mathbf{z}^*)$. We analyze the behavior of the distributions $p'_{\theta_s}(\mathbf{z}', c \mid \mathcal{D}_t)$ and $q'_{\phi_s}(\mathbf{z}', c \mid \mathcal{D}_T)$.

We prove that

$$\frac{p'_{\theta_s}(\mathbf{z}', c \mid \mathcal{D}_t)}{q'_{\phi_s}(\mathbf{z}', c \mid \mathcal{D}_T)}$$

converges to a constant independent of $\mathbf{z}'$ as $s \to \infty$. Since both $q'_{\phi_s}(\mathbf{z}', c \mid \mathcal{D}_T)$ and $p'_{\theta_s}(\mathbf{z}', c \mid \mathcal{D}_t)$ are probability distributions, the constant must be 1. Therefore, the KL divergence between them converges to 0 as $s \to \infty$.

We have:

$$\frac{p'_{\theta_s}(\mathbf{z}', c \mid \mathcal{D}_t)}{q'_{\phi_s}(\mathbf{z}', c \mid \mathcal{D}_T)} = \frac{\mathcal{N}(\mathbf{z}^* + \sigma_s^{\frac{t}{d_z}} \mathbf{z}' \mid \mathbf{z}^*, \sigma_s^2 \tilde{\Sigma}_\mathbf{z}(\mathcal{D}_T, \phi)) \pi_{\phi_s}(c \mid \mathcal{D}_T) p_{\theta_s}(y_{\leq t} \mid \mathbf{x}_{\leq t}, \omega_{\leq T})}{\prod_{l=1}^{t} \mathcal{N}(y_l \mid f_l(\mathbf{h}_l, \omega_l, \mathbf{z}^* + \sigma_s^{\frac{t}{d_z}} \mathbf{z}'; \theta_s), \sigma_s^2) \mathcal{N}(\mathbf{z}^* + \sigma_s^{\frac{t}{d_z}} \mathbf{z}' \mid \mu_c, \Sigma_c) p(c)} \tag{31}$$

Now, let $A_t$ be a matrix such that its Moore–Penrose inverse $A^+$ satisfies:

$$f_{\leq t}(\mathbf{h}_{\leq t}, \omega_{\leq t}, \mathbf{z}^* + \sigma_s^{\frac{t}{d_z}} \mathbf{z}'; \theta_s) = A^+ f_{\leq T}(\mathbf{h}_{\leq T}, \omega_{\leq T}, \mathbf{z}^* + \sigma_s^{\frac{t}{d_z}} \mathbf{z}'; \theta_s), \quad y_{\leq t} = A^+ y_{\leq T}$$

That is, projecting onto $A^+$ selects the first $t$ responses. The matrix $A^+$ has the form:

$$A^+ = \begin{bmatrix} I_t & 0_{t \times (T-t)} \end{bmatrix}, \quad A = \begin{bmatrix} I_t \\ 0_{(T-t) \times t} \end{bmatrix}$$

By noticing that $[A^\intercal(\sigma_s^2 I_T)^{-1}A]^{-1} = \sigma_s^2 I_t$ We can therefore apply the mean rearranging formula (Petersen et al., 2008):

$$\mathcal{N}\left(A^+ y_{\leq T} \,\middle|\, A^+ f_{\leq T}(\mathbf{h}_{\leq t}, \omega_{\leq t}, \mathbf{z}^* + \sigma_s^{\frac{t}{d_z}} \mathbf{z}'; \theta_s), [A^\intercal(\sigma_s^2 I_T)A]^{-1}\right)$$

$$= \frac{\sqrt{\det(2\pi\sigma_s^2 I_T)}}{\sqrt{\det(2\pi\sigma_s^2 I_t)}} \mathcal{N}\left(y_{\leq T} \,\middle|\, f_{\leq T}(\mathbf{h}_{\leq T}, \omega_{\leq T}, \mathbf{z}^* + \sigma_s^{\frac{t}{d_z}} \mathbf{z}'; \theta_s), \sigma_s^2 I_T\right)$$

$$\frac{p'_{\theta_s}(\mathbf{z}', c \mid \mathcal{D}_t)}{q'_{\phi_s}(\mathbf{z}', c \mid \mathcal{D}_T)} = \frac{\mathcal{N}\left(\mathbf{z}^* + \sigma_s^{\frac{t}{d_z}} \mathbf{z}' \mid \mathbf{z}^*, \sigma_s^2 \tilde{\Sigma}_{\mathbf{z}}(\mathcal{D}_T, \phi)\right) \pi_{\phi_s}(c \mid \mathcal{D}_T) p_{\theta_s}(y_{\leq t} \mid \mathbf{x}_{\leq t}, \omega_{\leq T})}{\frac{\sqrt{\det(2\pi\sigma_s^2 I_T)}}{\sqrt{\det(2\pi\sigma_s^2 I_t)}} \mathcal{N}\left(y_{\leq T} \mid f_{\leq T}(\mathbf{h}_{\leq T}, \omega_{\leq T}, \mathbf{z}^* + \sigma_s^{\frac{t}{d_z}} \mathbf{z}'; \theta_s), \sigma_s^2 I_T\right) \mathcal{N}(\mathbf{z}^* + \sigma_s^{\frac{t}{d_z}} \mathbf{z}' \mid \mu_c, \Sigma_c) p(c)}$$

$$= (2\pi)^{\frac{t-d}{2}} \frac{\det(\tilde{\Sigma}_{\mathbf{z}})^{-\frac{1}{2}}}{\det(\Sigma_c)^{-\frac{1}{2}}} \exp\left\{-\frac{1}{2}\sigma_s^{2(\frac{t}{d_z}-1)} z'^\top \tilde{\Sigma}_z^{-1} z' + \frac{1}{2\sigma_s^2} \sum_{l=1}^T (y_l - f_l(\mathbf{h}_l, \omega_l, \mathbf{z}^* + \sigma_s^{\frac{t}{d_z}} \mathbf{z}'; \theta_s))^2 \right.$$

$$\left. + \frac{1}{2}(\mathbf{z}^* + \sigma_s^{\frac{t}{d_z}} \mathbf{z}' - \mu_c)^\top \Sigma_c^{-1} (\mathbf{z}^* + \sigma_s^{\frac{t}{d_z}} \mathbf{z}' - \mu_c)\right\}$$

$$\times \frac{\pi_{\phi_s}(c \mid \mathcal{D}_T) p_{\theta_s}(y_{\leq T} \mid \mathbf{x}_{\leq t}, \omega_{\leq t})}{p(c)}$$

By Noticing that

$$\sum_{l=1}^T (y_l - f_l(\mathbf{h}_l, \omega_l, \mathbf{z}^* + \sigma_s^{\frac{t}{d_z}} \mathbf{z}'; \theta_s))^2 = \|y_{\leq T} - F^{-1}(\tilde{G}(\mathbf{z}^* + \sigma_s^{\frac{t}{d_z}} \mathbf{z}') \mid \mathbf{x}_{\leq T}, \omega_{\leq T})\|_2^2$$

We apply a first-order Taylor expansion

$$y_{\leq T} - F^{-1}(\tilde{G}(\mathbf{z}^* + \sigma_s^{\frac{t}{d_z}} \mathbf{z}') \mid \mathbf{x}_{\leq T}, \omega_{\leq T}) \underset{s\to\infty}{\approx} -Jac(F^{-1}(\tilde{G}))(\mathbf{z}^*)(\sigma_s^{\frac{t}{d_z}} \mathbf{z}')$$

which implies by norm continuity that

$$\sum_{l=1}^T (y_l - f_l(\mathbf{h}_l, \omega_l, \mathbf{z}^* + \sigma_s^{\frac{t}{d_z}} \mathbf{z}'; \theta_s))^2 \underset{s\to\infty}{\approx} \|Jac(F^{-1}(\tilde{G}))(\mathbf{z}^*)(\sigma_s \mathbf{z}')\|_2^2 \underset{s\to\infty}{=} \sigma_s^{\frac{2t}{d_z}} z'^\top \tilde{\Sigma}_z^{-1} z'$$

Now, define the constant $\mathcal{C}st(\mathcal{D}_t; \sigma_s, \theta_s, \phi_s)$ with respect to $\mathbf{z}'$ as:

$$\mathcal{C}st(\mathcal{D}_t; \sigma_s, \theta_s, \phi_s) = (2\pi)^{\frac{t-d_z}{2}} \sqrt{\frac{\det(\Sigma_c)}{\det(\tilde{\Sigma}_{\mathbf{z}})}} \frac{\pi_{\phi_s}(c \mid \mathcal{D}_T) p_{\theta_s}(y_{\leq t} \mid \mathbf{x}_{\leq t}, \omega_{\leq t})}{p(c)}.$$

Thus,

$$\frac{p'_{\theta_s}(\mathbf{z}', c \mid \mathcal{D}_T)}{q'_{\phi_s}(\mathbf{z}', c \mid \mathcal{D}_T)} \underset{s\to\infty}{\approx} \frac{1}{2}(\mathbf{z}^* - \mu_c)\Sigma_c^{-1}(\mathbf{z}^* - \mu_c)^\top \mathcal{C}st(\mathcal{D}_t; \sigma_s, \theta_s, \phi_s) \exp\left\{-\sigma_s^{2(\frac{t}{d_z}-1)}\frac{1}{2} z'^\top \tilde{\Sigma}_z^{-1} z' + \sigma_s^{2(\frac{t}{d_z}-1)}\frac{1}{2} z'^\top z'\right\}$$

$$\underset{s\to\infty}{\approx} \frac{1}{2}(\mathbf{z}^* - \mu_c)\Sigma_c^{-1}(\mathbf{z}^* - \mu_c)^\top \mathcal{C}st(\mathcal{D}_t; \sigma_s, \theta_s, \phi_s).$$

To conclude the proof, we still need to show that $\lim_{s\to+\infty} \mathcal{C}st(\mathcal{D}_t; \sigma_s, \theta_s, \phi_s)$ exists and is finite because the model parameters still depend on the chosen variance. This was the main misapprehension of Dai & Wipf (2019) for a VAE in the static setting with an unimodal prior over the continuous latents $\mathbf{Z}$.

We have already shown that

$$\lim_{\sigma\to 0^+} p_{\theta_\sigma}(y_{\leq T} \mid \mathbf{x}_{\leq T}, \omega_{\leq T}) = p(y_{\leq T} \mid \mathbf{x}_{\leq T}, \omega_{\leq T}),$$

and therefore:

$$\lim_{s\to+\infty} \mathcal{C}\text{st}(\mathcal{D}_t; \sigma_s, \theta_s, \phi_s) = (2\pi)^{\frac{t-d_z}{2}} \sqrt{\frac{\det(\Sigma_c)}{\det(\tilde{\Sigma}_{\mathbf{z}})}} \cdot \frac{p(y_{\leq T} \mid \mathbf{x}_{\leq T}, \omega_{\leq T})}{p(c)} \cdot \lim_{s\to+\infty} \pi_{\phi_s}(c \mid \mathcal{D}_T)$$

We verify that $\lim_{s\to+\infty} \pi_{\phi_s}(c \mid \mathcal{D}_T)$ exists and is finite and not zero for all $c \in \{1, \dots, K\}$:

$$
\begin{aligned}
\pi_{\phi_s}(c \mid \mathcal{D}_T) &= \frac{\exp(\mathbb{E}_{\mathbf{Z}\sim q_{\phi_z}(\cdot|\mathcal{D}_T)} \log p(c \mid \mathbf{Z}))}{\sum_{c=1}^{K} \exp(\mathbb{E}_{\mathbf{Z}\sim q_{\phi_z}(\cdot|\mathcal{D}_T)} \log p(c \mid \mathbf{Z}))} \\
&= \frac{\exp(\int_{\mathcal{Z}} \log p(c \mid \mathbf{z}) \mathcal{N}\left(\mathbf{z} \mid \mathbf{z}^*, \sigma_s^2 \tilde{\Sigma}_{\mathbf{z}}(\mathcal{D}_T, \phi)\right) d\mathbf{z})}{\sum_{c=1}^{K} \exp(\int_{\mathcal{Z}} \log p(c \mid \mathbf{z}) \mathcal{N}\left(\mathbf{z} \mid \mathbf{z}^*, \sigma_s^2 \tilde{\Sigma}_{\mathbf{z}}(\mathcal{D}_T, \phi)\right) d\mathbf{z})} \\
&\xrightarrow[s\to\infty]{} \frac{\exp(\int_{\mathcal{Z}} \log p(c \mid \mathbf{z}) \delta(\mathbf{z} - \mathbf{z}^*) d\mathbf{z})}{\sum_{c=1}^{K} \exp(\int_{\mathcal{Z}} \log p(c \mid \mathbf{z}) \delta(\mathbf{z} - \mathbf{z}^*) d\mathbf{z})} \\
&= \frac{p(c \mid \mathbf{z}^*)}{\sum_{c=1}^{K} p(c \mid \mathbf{z}^*)}
\end{aligned}
$$

We have therefore proved that the ratio $\frac{p'_{\theta_s}(\mathbf{z}', c|\mathcal{D}_T)}{q'_{\phi_s}(\mathbf{z}', c|\mathcal{D}_T)}$ converges into a nonzero constant which finishes the proof of the theorem. ∎

**Proof of theorem 9.** We start by considering the ELBO:

$$
\begin{aligned}
\mathcal{L}(\mathcal{D}_T; \theta, \phi) = &-\sum_{t=1}^{T} \mathbb{E}_{\mathbf{Z}\sim q_\phi(\cdot|\mathcal{D}_T)} \left[\alpha(\mathbf{h}_t, \omega_t) \log p_\theta(y_t \mid \mathbf{H}_t, W_t, \mathbf{Z})\right] \\
&+ \left[\sum_{t=1}^{T} \alpha(\mathbf{h}_t, \omega_t)\right] D_{KL}\left(q_\phi(\mathbf{z} \mid \mathcal{D}_T) \parallel p(\mathbf{z})\right) \\
&- \left[\sum_{t=1}^{T} \alpha(\mathbf{h}_t, \omega_t)\right] \log Z(q_\phi(\cdot \mid \mathcal{D}_T)).
\end{aligned}
$$

By the positivity of the KL divergence and using the fact that $-\log Z(q_\phi(\cdot \mid \mathcal{D}_T))$ minimizes a positive functional (as explained in Eq. equation 16), we obtain:

$$
\begin{aligned}
\mathcal{L}(\mathcal{D}_T; \theta_\sigma^*, \phi_\sigma^*) \geq &-\sum_{t=1}^{T} \mathbb{E}_{\mathbf{Z}\sim q_\phi(\cdot|\mathcal{D}_T)} \left[\alpha(\mathbf{h}_t, \omega_t) \log\left(p_\theta\left(y_t \mid \mathbf{H}_t, W_t, \mathbf{Z}\right)\right)\right] \\
= &\sum_{t=1}^{T} \alpha(\mathbf{h}_t, \omega_t) \mathbb{E}_{\mathbf{Z}\sim q_\phi(\cdot|\mathcal{D}_T)} \left[\frac{1}{2} \log\left(2\pi\sigma^2\right) + \frac{1}{2\sigma^2}\left(y_t - f\left(\mathbf{H}_t, \mathbf{Z}, W_t\right)\right)^2\right] \\
= &\frac{1}{2\sigma^2}\left[\left(\sum_{t=1}^{T} \alpha(\mathbf{h}_t, \omega_t)\right) \sigma^2 \log\left(2\pi\sigma^2\right) + \sum_{t=1}^{T} \alpha(\mathbf{h}_t, \omega_t) \underbrace{\mathbb{E}_{\mathbf{Z}\sim q_\phi(\cdot|\mathcal{D}_T)}\left(y_t - f\left(\mathbf{H}_t, \mathbf{Z}, W_t\right)\right)^2}_{\delta_\sigma(t)}\right].
\end{aligned}
$$

Suppose there exists $t_0 \in \{1, 2, \dots, T\}$ such that $\lim_{\sigma\to0^+} \delta_\sigma(t_0) > 0$. Then we have $\lim_{\sigma\to0^+} \mathcal{L}(\mathcal{D}_T; \theta_\sigma^*, \phi_\sigma^*) = +\infty$, which is impossible since we assume $\mathcal{L}(\mathcal{D}_T; \theta_\sigma^*, \phi_\sigma^*)$ to be minimized. Therefore, we conclude that for all $t \in \{1, 2, \dots, T\}$, $\lim_{\sigma\to0^+} \delta_\sigma(t) = 0$.

Consequently, we have

$$\lim_{\sigma\to0^+} \mathbb{E}_{\epsilon\sim\mathcal{N}(0,I)}\left(y_t - f\left(\mathbf{H}_t, W_t, f_{\mu_\mathbf{z}}(\mathcal{D}_T; \phi_\sigma^*) + f_{S_\mathbf{z}}(\mathcal{D}_T; \theta_\sigma^*)\epsilon\right)\right)^2 = 0,$$

which implies that

$$\lim_{\sigma \to 0^+} f\left(\mathbf{H}_t, W_t, f_{\mu_{\mathbf{z}}}(\mathcal{D}_T; \phi_\sigma^*) + f_{S_{\mathbf{z}}}(\mathcal{D}_T; \theta_\sigma^*)\epsilon\right) = y_t, \quad \text{almost surely.}$$

In particular, we also have

$$\lim_{\sigma \to 0^+} f\left(\mathbf{H}_t, W_t, f_{\mu_{\mathbf{z}}}(\mathcal{D}_T; \phi_\sigma^*)\right) = y_t.$$

### A.5 Upper bound on weighted PEHE

**Proof** [**theorem** 11] To prove the theorem, we rely on the following key lemma, but first, we need to define further mathematical objects. We define the weighted population risk over the whole population as

$$R_{t,g}(f, \Phi) \coloneqq \mathbb{E}_{\mathbf{H}_t, W_t}\left[\alpha(\mathbf{H}_t, W_t)\,\ell_{f,\Phi}(\mathbf{H}_t, W_t)\right],$$

and the weighted population counterfactual risk as

$$R_{t,g}(f, \Phi)_{CF} \coloneqq \mathbb{E}_{\mathbf{H}_t, 1-W_t}\left[\alpha(\mathbf{H}_t, 1 - W_t)\,\ell_{f,\Phi}(\mathbf{H}_t, W_t)\right].$$

**Lemma 17**

$$R_{t,g}(f, \Phi) + R_{t,g}(f, \Phi)_{CF} \le R_{t,g}^1(f, \Phi) + R_{t,g}^0(f, \Phi) + B_\Phi\,\mathrm{IPM}_G\left(g_\Phi(\cdot \mid W_t = 1),\, g_\Phi(\cdot \mid W_t = 0)\right)$$

**Proof** The proof immediately follows from lemma 1 in Shalit et al. (2017). ∎

Next, to complete the proof, we define the expected potential outcome at time $t$, given the context history $\mathbf{H}_t = \mathbf{h}_t$ and latent $\mathbf{z}$ as

$$m_t^\omega(\mathbf{h}_t, \mathbf{z}) \coloneqq \mathbb{E}_{Y_t(\omega)|\mathbf{H}_t, \mathbf{Z}}(Y_t(\omega) \mid \mathbf{H}_t = \mathbf{h}_t, \mathbf{Z} = \mathbf{z}) \quad \omega \in \mathcal{W},$$

and we show the following lemma:

**Lemma 18** *Denote* $\hat{p}_\phi(\mathbf{h}_t, w, \mathbf{z}) \coloneqq p(\mathbf{h}_t, w)\,q_\phi(\mathbf{z} \mid \mathcal{D}_{\le t-1})$. *We can decompose* $R_{t,g}(f, \Phi)$ *and* $R_{t,g}(f, \Phi)_{\mathrm{CF}}$ *using our distribution assumptions as follows:*

$$\mathbb{E}_{\hat{p}_\phi(\mathbf{h}_t, w, \mathbf{z})}\left[\alpha(\mathbf{h}_t, w)\left(f(\mathbf{h}_t, w, \mathbf{z}) - m_t^w(\mathbf{h}_t, \mathbf{z})\right)\right] = 2\sigma^2\left(R_{t,g}(f, \Phi) - \tfrac{1}{2}\log(2\pi\sigma^2)\right) - \sum_{\omega \in \{0,1\}} \mathrm{Var}_{\hat{p}_\phi(\mathbf{h}_t, w, \mathbf{z})}(Y_t),$$

*and,*

$$\mathbb{E}_{\hat{p}_\phi(\mathbf{h}_t, w, \mathbf{z})}\left[\alpha(\mathbf{h}_t, 1-w)\left(f(\mathbf{h}_t, 1-w, \mathbf{z}) - m_t^{1-w}(\mathbf{h}_t, \mathbf{z})\right)\right] = 2\sigma^2\left(R_{t,g}(f, \Phi)_{\mathrm{CF}} - \tfrac{1}{2}\log(2\pi\sigma^2)\right) - \sum_{\omega \in \{0,1\}} \mathrm{Var}_{\hat{p}_\phi(\mathbf{h}_t, 1-w, \mathbf{z})}(Y_t).$$

**Proof** We have:

$$
\begin{aligned}
R_{t,g}(f, \Phi) &= \mathbb{E}_{\mathbf{H}_t, W_t}\left[\alpha(\mathbf{H}_t, W_t)\,\ell_{f,\Phi}(\mathbf{H}_t, W_t)\right] \\
&= \sum_{\omega \in \{0,1\}} \int_{\mathcal{H}_t \times \mathcal{Z} \times \mathcal{Y}} -\alpha(\mathbf{h}_t, \omega) \log p_\theta(y_t \mid \mathbf{h}_t, \omega, \mathbf{z})\, p(y_t \mid \mathbf{h}_t, \omega, \mathbf{z}) p(\mathbf{h}_t, \omega)\, q_\phi(\mathbf{z} \mid \mathcal{D}_{\le t-1}) dy_t d\mathbf{z} d\mathbf{h}_t \\
&= \sum_{\omega \in \{0,1\}} \int_{\mathcal{H}_t} \int_{\mathcal{Z}} \int_{\mathcal{Y}} \alpha(\mathbf{h}_t, \omega)\left\{\tfrac{1}{2}\log\left(2\pi\sigma^2\right) + \tfrac{1}{2\sigma^2}\left(y_t - f\left(\mathbf{h}_t, \mathbf{z}, \omega\right)\right)^2\right\} p(y_t \mid \mathbf{h}_t, \omega, \mathbf{z})\, p(\mathbf{h}_t, \omega)\, q_\phi(\mathbf{z} \mid \mathcal{D}_{\le t-1}) dy_t d\mathbf{z} d\mathbf{h}_t \\
&= \tfrac{1}{2}\log\left(2\pi\sigma^2\right) \sum_{\omega \in \{0,1\}} \int_{\mathcal{H}_t} \alpha(\mathbf{h}_t, \omega)\, p(\mathbf{h}_t, \omega) d\mathbf{h}_t d\omega + \tfrac{1}{2\sigma^2} \int_{\mathcal{H}_t} \int_{\mathcal{Z}} \int_{\mathcal{Y}} \alpha(\mathbf{h}_t, \omega)\Big\{\left(y_t - m_t^\omega(\mathbf{h}_t, \mathbf{z})\right) \\
&\quad - \left(f\left(\mathbf{h}_t, \omega, \mathbf{z}\right) - m_t^\omega(\mathbf{h}_t, \mathbf{z})\right)\Big\}^2 p(y_t \mid \mathbf{h}_t, \omega, \mathbf{z})\, p(\mathbf{h}_t, \omega)\, q_\phi(\mathbf{z} \mid \mathcal{D}_{\le t-1}) dy_t d\mathbf{z} d\mathbf{h}_t \\
&= \tfrac{1}{2}\log\left(2\pi\sigma^2\right) + \tfrac{1}{2\sigma^2} \sum_{\omega \in \{0,1\}} Var_{\hat{p}_\phi(\mathbf{h}_t, \omega, \mathbf{z})}(Y_t) + \int_{\mathcal{H}_t} \int_{\mathcal{Z}} \alpha(\mathbf{h}_t, \omega)(f\left(\mathbf{h}_t, \omega, \mathbf{z}\right) - m_t^\omega(\mathbf{h}_t, \mathbf{z}))^2 p(\mathbf{h}_t, \omega)\, q_\phi(\mathbf{z} \mid \mathcal{D}_{\le t-1}) d\mathbf{z} d\mathbf{h}_t
\end{aligned}
$$

In a similar way, we can show the decomposition related to $R_{t,g}(f,\Phi)_{CF}$. ∎

The remainder idea of the proof is to decompose the PEHE in such a way we can upper bound it with an expression including $R_{t,g}(f,\Phi)$ and $R_{t,g}(f,\Phi)_{CF}$ and then use lemma 17. We actually have:

$$
\begin{aligned}
\epsilon_{\text{PEHE}_{t,g}} &= \mathbb{E}_{\mathbf{H}_t \sim g}\mathbb{E}_{\mathbf{Z} \sim q_\phi(\mathbf{Z}|\mathcal{D}_{\leq t-1})}\left[(\tau(\mathbf{H}_t,\mathbf{Z}) - \hat{\tau}_{f,\Phi}(\mathbf{H}_t,\mathbf{Z}))^2\right] \\
&= \mathbb{E}_{\mathbf{H}_t \sim g}\mathbb{E}_{\mathbf{Z} \sim q_\phi(\mathbf{Z}|\mathcal{D}_{\leq t-1})}\left[\left(m_t^1(\mathbf{h}_t,\mathbf{z}) - m_t^0(\mathbf{h}_t,\mathbf{z}) - f(\mathbf{h}_t,1,\mathbf{z}) + f(\mathbf{h}_t,0,\mathbf{z})\right)^2\right] \\
&\overset{(1)}{\leq} 2\mathbb{E}_{\mathbf{H}_t \sim g}\mathbb{E}_{\mathbf{Z} \sim q_\phi(\mathbf{Z}|\mathcal{D}_{\leq t-1})}\left[(f(\mathbf{h}_t,1,\mathbf{z}) - m_t^1(\mathbf{h}_t,\mathbf{z}))^2\right] + 2\mathbb{E}_{\mathbf{H}_t \sim g}\mathbb{E}_{\mathbf{Z} \sim q_\phi(\mathbf{Z}|\mathcal{D}_{\leq t-1})}\left[(f(\mathbf{h}_t,0,\mathbf{z}) - m_t^0(\mathbf{h}_t,\mathbf{z}))^2\right] \\
&= \mathbb{E}_{\mathbf{H}_t|W_t=1}\mathbb{E}_{\mathbf{Z} \sim q_\phi(\mathbf{Z}|\mathcal{D}_{\leq t-1})}\left[\alpha(\mathbf{h}_t,1)(f(\mathbf{h}_t,1,\mathbf{z}) - m_t^1(\mathbf{h}_t,\mathbf{z}))^2\right] \\
&\quad + \mathbb{E}_{\mathbf{H}_t|W_t=0}\mathbb{E}_{\mathbf{Z} \sim q_\phi(\mathbf{Z}|\mathcal{D}_{\leq t-1})}\left[\alpha(\mathbf{h}_t,0)(f(\mathbf{h}_t,1,\mathbf{z}) - m_t^1(\mathbf{h}_t,\mathbf{z}))^2\right] \\
&\quad + \mathbb{E}_{\mathbf{H}_t|W_t=0}\mathbb{E}_{\mathbf{Z} \sim q_\phi(\mathbf{Z}|\mathcal{D}_{\leq t-1})}\left[\alpha(\mathbf{h}_t,0)(f(\mathbf{h}_t,0,\mathbf{z}) - m_t^0(\mathbf{h}_t,\mathbf{z}))^2\right] \\
&\quad + \mathbb{E}_{\mathbf{H}_t|W_t=1}\mathbb{E}_{\mathbf{Z} \sim q_\phi(\mathbf{Z}|\mathcal{D}_{\leq t-1})}\left[\alpha(\mathbf{h}_t,1)(f(\mathbf{h}_t,0,\mathbf{z}) - m_t^0(\mathbf{h}_t,\mathbf{z}))^2\right] \\
&= \mathbb{E}_{\mathbf{H}_t,W_t}\mathbb{E}_{q_\phi(\mathbf{Z}|\mathcal{D}_{\leq t-1})}[\alpha(\mathbf{H}_t,W_t)(f(\mathbf{H}_t,W_t,\mathbf{Z}) - m_t^{W_t}(\mathbf{H}_t,\mathbf{Z}))] \\
&\quad + \mathbb{E}_{\mathbf{H}_t,1-W_t}\mathbb{E}_{q_\phi(\mathbf{Z}|\mathcal{D}_{\leq t-1})}[\alpha(\mathbf{H}_t,1-W_t)(f(\mathbf{H}_t,W_t,\mathbf{Z}) - m_t^{W_t}(\mathbf{H}_t,\mathbf{Z}))] \\
&\overset{(2)}{=} 2\sigma^2\left\{R_{t,g}(f,\Phi) + R_{t,g}(f,\Phi)_{CF} - \log\left(2\pi\sigma^2\right)\right\} - \sum_{\omega \in \{0,1\}}(Var_{\hat{p}_\phi(\mathbf{h}_t,1-\omega,\mathbf{z})}(Y_t) + Var_{\hat{p}_\phi(\mathbf{h}_t,\omega,\mathbf{z})}(Y_t)) \\
&\overset{(3)}{\leq} 2\sigma^2\left\{R_{t,g}(f,\Phi) + R_{t,g}(f,\Phi)_{CF} - \log\left(2\pi\sigma^2\right)\right\} \\
&\overset{(4)}{\leq} 2\sigma^2\left\{R_{t,g}^1(f,\Phi) + R_{t,g}^0(f,\Phi) + B_\Phi \, \text{IPM}_G\left(g_\Phi(\cdot \mid W_t=1),\, g_\Phi(\cdot \mid W_t=0)\right) - \log\left(2\pi\sigma^2\right)\right\}
\end{aligned}
$$

The inequality $\overset{(1)}{\leq}$ follows from the property $(a-b)^2 \leq 2a^2 + 2b^2$. The equation $\overset{(2)}{=}$ follows from plugging in the equations from Lemma 18, the inequality $\overset{(3)}{\leq}$ follows from the positivity of the variance terms, and $\overset{(4)}{\leq}$ follows from Lemma 17. ∎

## A.6 Proof of Proposition 12

Using a Monte Carlo approximation, we can express the approximate factual risk as:

$$
R_{t,g}^\omega(f,\Phi) \approx \frac{1}{n_\omega^{(t)}}\sum_{i \in \mathcal{B}, W_{it}=\omega}\mathbb{E}_{\mathbf{Z} \sim q_\phi(\mathbf{Z}|\mathcal{D}_{i,\leq t-1})}\left[\alpha(\mathbf{h}_{it},\omega)\log\mathcal{N}(y_{it}; f(\Phi(\mathbf{h}_{it}),\mathbf{Z},\omega),\sigma^2)\right].
$$

By the stationarity assumption, for $t \geq t_0$, this approximation holds:

$$
R_{t,g}^\omega(f,\Phi) \approx \frac{1}{n_\omega^{(t)}}\sum_{i \in \mathcal{B}, W_{it}=\omega}\mathbb{E}_{\mathbf{Z} \sim q_\phi(\mathbf{Z}|\mathcal{D}_{iT})}\left[\alpha(\mathbf{h}_{it},\omega)\log\mathcal{N}(y_{it}; f(\Phi(\mathbf{h}_{it}),\mathbf{Z},\omega),\sigma^2)\right].
$$

On the other hand, the reconstruction term in the ELBO can be written as:

$$
\begin{aligned}
\sum_{t=t_0}^{T}\mathbb{E}_{\mathbf{Z} \sim q_\phi(\cdot|\mathcal{D}_T)}[\alpha(\mathbf{H}_t,W_t)\log p_\theta(Y_t \mid \mathbf{H}_t,W_t,\mathbf{Z})] &\approx \sum_{t=t_0}^{T}\frac{1}{|\mathcal{B}|}\sum_{i \in \mathcal{B}}\mathbb{E}_{\mathbf{Z} \sim q_\phi(\mathbf{Z}|\mathcal{D}_{iT})}\left[\alpha(\mathbf{h}_{it},\omega)\log\mathcal{N}(y_{it}; f(\Phi(\mathbf{h}_{it}),\mathbf{Z},\omega),\sigma^2)\right] \\
&\approx \frac{1}{|\mathcal{B}|}\sum_{t=t_0}^{T}\left\{\sum_{\omega \in \mathcal{W}}\sum_{i \in \mathcal{B}, W_{it}=\omega}\mathbb{E}_{\mathbf{Z} \sim q_\phi(\mathbf{Z}|\mathcal{D}_{iT})}\left[\alpha(\mathbf{h}_{it},\omega)\log\mathcal{N}(y_{it}; f(\Phi(\mathbf{h}_{it}),\mathbf{Z},\omega)\right.\right. \\
&\approx \frac{1}{|\mathcal{B}|}\sum_{t=t_0}^{T}\left\{-n_1^{(t)}R_{t,g}^1(f,\Phi) - n_0^{(t)}R_{t,g}^0(f,\Phi)\right\}.
\end{aligned}
$$

## B  Discussion of the validity of the Risk Factor Substitute

### B.1  Estimating $p(Y_t(\omega))$

In causal inference, when unobserved confounders are present, the conditional distribution of potential outcomes plays a critical role. If the adjustment variables $\mathbf{U}$ are observed, the potential outcome can be represented as:

$$p(Y_t(\omega)) = p(Y_t \mid \mathbf{h}_t, W_t = \omega) = \int p(y_t \mid \mathbf{h}_t, W_t = \omega, \mathbf{U} = \mathbf{u})p(\mathbf{u} \mid \mathbf{h}_t, W_t = \omega)d\mathbf{u}.$$

Due to the structural assumptions over the causal graph, the treatments $W_t$ and covariates $\mathbf{X}_t$ are d-separated from $\mathbf{U}$ given $\mathbf{H}_t$, implying:

$$p(\mathbf{u} \mid \mathbf{h}_t, W_t = \omega) = p(\mathbf{u} \mid \mathcal{D}_{t-1}),$$

where the conditional potential outcome is identifiable, but estimating it requires sampling from the posterior distribution of the adjustment variables, $p(\mathbf{u} \mid \mathcal{D}_{t-1})$.

When $\mathbf{U}$ is not observed, let $\mathbf{Z}$ be a latent variable satisfying the CMM(p). Under what conditions does the following equality hold?

$$\int p(y_t \mid \mathbf{h}_t, W_t = \omega, \mathbf{Z} = \mathbf{z})p(\mathbf{z} \mid \mathcal{D}_{t-1})\,d\mathbf{z} \overset{(*)}{=} \int p(y_t \mid \mathbf{h}_t, W_t = \omega, \mathbf{U} = \mathbf{u})p(\mathbf{u} \mid \mathcal{D}_{t-1})\,d\mathbf{u}$$
$$= p(Y_t(\omega) \mid \mathbf{h}_t). \tag{32}$$

This equation addresses the potential risk of non-uniqueness in the CMM assumption, which could lead to varying values on the left-hand side (LHS) of Eq. (32). However, our framework differs from the traditional deconfounder framework. Here, the identification of the conditional potential outcome $p(Y_t \mid \mathbf{h}_t)$ occurs *independently* of the latent variables, as sequential ignorability holds without requiring conditioning on them. The CMM assumption, under the condition $\mathbf{Z} \perp\!\!\!\perp \mathbf{X}_{\leq T}, W_{\leq T}$, ensures that no confounding backdoor paths are introduced by conditioning on $\mathbf{Z}$. Thus, using $\mathbf{Z}$ as a substitute for $\mathbf{U}$ presents no identifiability or consistency issues. This conclusion also extends to the marginal distribution of potential outcomes, which can be formally shown through a copula argument, as in D'Amour (2019a).

$$p(Y_t(\omega)) = \int p(Y_t(\omega) \mid \mathbf{H}_t = \mathbf{h}_t)p(\mathbf{h}_t)d\mathbf{h}_t,$$
$$p(Y_t(\omega)) = \int p(Y_t \mid \mathbf{H}_t = \mathbf{h}_t, W_t = \omega)p(\mathbf{h}_t)d\mathbf{h}_t,$$
$$p(Y_t(\omega)) = \int p(y_{\leq t}, \mathbf{x}_{\leq t}, \omega_{\leq t})\frac{p(\mathbf{h}_t)}{p(\mathbf{h}_t, \omega_t)}d\mathbf{h}_t, \tag{33}$$
$$p(Y_t(\omega)) = \int \int p(y_{\leq t}, \mathbf{z} \mid \mathbf{x}_{\leq t}, \omega_{\leq t})p(\mathbf{x}_{\leq t}, \omega_{\leq t})\frac{p(\mathbf{h}_t)}{p(\mathbf{h}_t, \omega_t)}d\mathbf{h}_t d\mathbf{z},$$
$$= \int \int c(y_{\leq t}, \mathbf{z} \mid \mathbf{x}_{\leq t}, \omega_{\leq t})p(y_t \mid \mathbf{h}_t, \omega_t)p(\mathbf{z} \mid \mathbf{x}_{\leq t}, \omega_{\leq t})p(\mathbf{h}_t)d\mathbf{h}_t d\mathbf{z}.$$

The copula $c(y_{\leq t}, \mathbf{z} \mid \mathbf{x}_{\leq t}, \omega_{\leq t})$ is restricted by the CMM(p) assumption. This constraint differs significantly from the deconfounder theory, where factor models often separate multiple treatments or treatment sequences and impose no such restrictions on the outcome model. Therefore, identifying potential outcomes in the deconfounder framework requires additional assumptions, as discussed in Wang & Blei (2019a;b). However, our framework enforces conditional separation of responses given past sequences of treatments and covariates, implying independence between the latent variables and treatment sequences. Finally, in practice, how can

we estimate $p(Y_t(\omega))$? We express this as:

$$
\begin{aligned}
p(Y_t(\omega)) &= \int p(y_t \mid \mathbf{H}_t = \mathbf{h}_t, W_t = \omega)p(\mathbf{h}_t)d\mathbf{h}_t, \\
&= \int \int p(y_t \mid \mathbf{h}_t, \omega, \mathbf{z})p(\mathbf{z} \mid \mathbf{h}_t, \omega)p(\mathbf{h}_t)d\mathbf{h}_t d\mathbf{z}, \\
&= \int \int p(y_t \mid \mathbf{h}_t, \omega, \mathbf{z})p(\mathbf{z} \mid \mathcal{D}_{t-1})p(\mathbf{h}_t)d\mathbf{h}_t d\mathbf{z}, \\
&\approx \int \int p_\theta(y_t \mid \mathbf{h}_t, \omega, \mathbf{z})q_\phi(\mathbf{z} \mid \mathcal{D}_T)p(\mathbf{h}_t)d\mathbf{h}_t d\mathbf{z}.
\end{aligned}
\tag{34}
$$

Here, $p_\theta(y_t \mid \mathbf{h}_t, \omega, \mathbf{z})$ is modeled by the outcome model (decoder), while $q_\phi(\mathbf{z} \mid \mathcal{D}_T)$ is the approximate posterior distribution of the latent variables, estimated using the observed sequence of data.

### B.2 Capturing "bad variables"

In this section, we examine whether the substitute adjustment variables $\mathbf{Z}$ can inadvertently capture "bad variables," that is, variables that may introduce bias into the estimation of causal effects.

**M-Colliders** One potential source of bias arises from M-colliders. If the substitute adjustment variables $\mathbf{Z}$ capture information not only about the true adjustment variables $\mathbf{U}$, but also about a collider variable $\mathbf{M}$, this could result in $\mathbf{Z} \not\perp W_{\leq T}$.

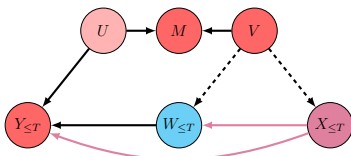

Figure 11: An example of an M-collider structure, where the variable $V$ influences both the treatment and the response, as indicated by the dashed arrows.

The key concern is whether $\mathbf{Z}$ could "blindly" capture information from the sequence of responses or treatments. Such cases may lead to open collider paths like $\mathbf{U} \to Y_t \leftarrow W_{\leq t}$ or $\mathbf{U} \to Y_t \leftarrow \mathbf{X}_{\leq t}$. These scenarios would violate the assumption that $\mathbf{Z} \perp W_{\leq t}$ and $\mathbf{Z} \perp \mathbf{X}_{\leq t}$.

In essence, the inclusion of colliders in the substitute adjustment variables can open "backdoor" paths, leading to biased estimates of the treatment effect. To prevent this, it is essential to ensure that the substitute variables $\mathbf{Z}$ do not inadvertently encode information about colliders that affect both the treatments and the responses.

## C Experiments on synthetic data: Details

### C.1 Description of the Simulation Model

#### C.1.1 Simulation under sequential ignorability

We simulate a longitudinal data set of time-length $T = 75$ by generating the time-varying variables autoregressively. Specifically, the confounders $\mathbf{X}_t$ at each time-step $t$ are generated in $\mathbb{R}^{d_x}$ with $d_x = 100$, and the dynamics are specified through an autoregression of order $p = 8$ plus a regression over the past treatment trajectory. Specifically, each dimension $j \in \{1, \ldots, d_x\}$ of $\mathbf{X}_t$ is defined as:

$$
\mathbf{X}_t^{(j)} = \frac{1}{p}\sum_{k=1}^{p}\gamma_k^{X,(t,j)}\mathbf{X}_{t-k}^{(j)} + \frac{1}{p}\sum_{k=1}^{p}\gamma_k^{XW,(t,j)}W_{t-k} + \epsilon_{t,j}^X,
$$

$$\gamma_k^{X,(t,j)} \sim \mathcal{N}(0,1) \quad \gamma_k^{XW,(t,j)} \sim \mathcal{N}(0,1) \quad \epsilon_t^X := \begin{bmatrix} \epsilon_{t,1}^X \\ \epsilon_{t,2}^X \\ \vdots \\ \epsilon_{t,d_x}^X \end{bmatrix} \sim \mathcal{N}(0,\Sigma_x).$$

To ensure dependence between confounders, the vector error $\epsilon_t^X$ is generated by a Gaussian distribution with a non-diagonal covariance matrix

$$\Sigma_x = \rho \mathbf{1}_{d_x} \mathbf{1}_{d_x}^\intercal + (1-\rho)\sigma^2 I_{d_x},$$

where $\rho = 0.5$ and $\sigma^2 = 0.3$.

The treatment $\omega_t$ is generated using a Bernoulli distribution with a probability $\sigma(\pi_t)$ defined with a logistic model to simulate the assignment mechanism. $\sigma(.)$ refers to the sigmoid function:

$$\pi_t = \frac{1}{p}\sum_{k=1}^p \gamma_k^{W,t} W_{t-k} + \frac{1}{d_x p}\sum_{k=1}^p \langle \gamma_k^{WX,t}, \mathbf{X}_{t-k}\rangle + \frac{1}{p}\sum_{k=1}^p \gamma_k^{WY,t} Y_{t-k} + \epsilon_W.$$

To ensure the variation of imbalance between treatment and control groups through time, we simulate the regression parameters as follows:

$$\gamma_k^{W,t}, \gamma_k^{WY,t} \sim \mathcal{N}(\sin(\frac{t}{\pi}), 0.01^2) \quad \gamma_k^{WX,t} \sim \mathcal{N}(\sin(\frac{t}{\pi}), 0.01^2)^{\otimes d_x} \quad \epsilon_W \sim \mathcal{N}(0, 0.01^2)$$

$$W_t \sim \mathcal{B}(\sigma(\pi_t))$$

To specify the outcome model, we first generate the unobserved adjustment variables $\mathbf{U}$ in $\mathbb{R}^{d_u}$ with $d_u = 100$ using a Gaussian mixture of three distributions:

$$\mathbf{U} \sim \frac{1}{K}\sum_{i=1}^K \mathcal{N}(\mu_i, \Sigma_u)$$

where $\mu_1, \ldots, \mu_K, \sim \mathcal{U}([-10, 10])^{\otimes d_u}$, and $\Sigma_u = 0.2\,\mathrm{diag}(\mathbf{1}_{d_u})$, and $K = 8$. Finally, we write the expression of the two potential outcomes as a function of the context history. The chosen specification is motivated by the classic mixed-effect approach. We introduce the random effects to have an individual reaction to the change in a covariate $\mathbf{X}_{t,j}$, and we model it by the Hadamard product between the covariate vector and the unobserved adjustment vector. The observed response is defined following the consistency assumption 1:

$$Y_t = Y_t(1)W_t + Y_t(0)(1 - W_t)$$

$$Y_t(\omega) = \frac{1}{d_U}\langle \gamma_\omega^{YU}, \mathbf{U}\rangle \sum_{k=1}^p W_{t-k} + \frac{1}{d_x p}\sum_{k=1}^p \langle \gamma_{k,\omega}^{YX,t}, \mathbf{X}_{t-k}\rangle + \frac{1}{p}\sum_{k=1}^p \gamma_{k,\omega}^{Y,t} Y_{t-k} + \epsilon_Y$$

$$\gamma_{k,\omega}^{YX,t} \sim \mathcal{N}(\gamma_\omega^{YX}, 0.01^2) \quad \gamma_\omega^{YU} \sim \mathcal{N}(\gamma_\omega^{YU}, 0.01^2) \quad \gamma_{k,(\omega)}^{Y,t} \sim \mathcal{N}(\gamma_\omega^Y, 0.1^2) \quad \epsilon_Y \sim \mathcal{N}(0, 0.01^2)$$

With $\gamma_1^{YX}, \gamma_1^Y = 0.0, 0.2$ and $\gamma_0^{YX}, \gamma_0^Y = 0.2, 0.1$ and $\gamma_0^{YU} = 0.1$ while $\gamma_1^{YU}$ is varied generate multiple datasets with different modification levels of treatment effect induced by $\mathbf{U}$.

### C.1.2 Simulation under hidden confounding

We partition the covariates into *observed* and *hidden* blocks:

$$\mathbf{X}_t = (\mathbf{X}_t^o, \mathbf{X}_t^h), \quad \mathbf{X}_t^o \in \mathbb{R}^{d_o}, \quad \mathbf{X}_t^h \in \mathbb{R}^{d_h}, \quad d_o + d_h = d_x.$$

We keep the block-diagonal covariance

$$\Sigma_x = \begin{pmatrix} \Sigma_{zz} & 0 \\ 0 & \Sigma_{hh} \end{pmatrix},$$

so that $\mathbf{X}_t^h$ is **not** a proxy for $\mathbf{X}_t^o$ at baseline.

Define the *hidden score* aggregated over the current $p$ steps:

$$S_t := \frac{1}{d_h} \sum_{j=1}^{d_h} \sum_{k=0}^{p-1} \mathbf{X}_{t-k}^{h,j}.$$

We inject the hidden score $S_t$ into the treatment logit with a selection-sensitivity parameter $\gamma$:

$$\pi_t = \underbrace{\pi_t^{(0)}}_{\text{baseline under ignorability}} + \gamma\, S_t, \quad W_t \sim \text{Bernoulli}\{\sigma(\pi_t)\}.$$

**Interpretation (Rosenbaum-$\Gamma$):** If $S_t$ is standardized, $\gamma = \log\Gamma$ where $\Gamma \geq 1$ is the maximum treatment odds ratio due to unobserved confounding.

We allow $S_t$ to load differently on the two potential outcomes via coefficients $\beta_0, \beta_1$:

$$Y_t(\omega) = Y_t^{(0)}(\omega) + \beta_\omega\, S_t + \tilde{\varepsilon}_t^{(\omega)}, \quad \omega \in \{0, 1\}.$$

The ITE is then

$$\tau_t = \left(Y_t^{(0)}(1) - Y_t^{(0)}(0)\right) + (\beta_1 - \beta_0)S_t + \left(\tilde{\varepsilon}_t^{(1)} - \tilde{\varepsilon}_t^{(0)}\right).$$

If $\beta_1 \neq \beta_0$, the hidden block changes the ITE, making PEHE sensitive to hidden confounding.

## C.2 Additional results

**Results of baselines on the synthetic datasets** The following Table 7 provides the detailed results responsible for Figures 3 related to baselines with the three different approaches across levels of $\gamma_{(1)}^{YU}$.

Table 7: Results on the synthetic data reported by PEHE. Smaller is better.

| Model | $\gamma_{(1)}^{YU}=0$ | $\gamma_{(1)}^{YU}=0.25$ | $\gamma_{(1)}^{YU}=0.5$ | $\gamma_{(1)}^{YU}=0.75$ | $\gamma_{(1)}^{YU}=1$ | $\gamma_{(1)}^{YU}=1.25$ | $\gamma_{(1)}^{YU}=1.5$ | $\gamma_{(1)}^{YU}=1.75$ | $\gamma_{(1)}^{YU}=2$ | $\gamma_{(1)}^{YU}=2.25$ | $\gamma_{(1)}^{YU}=2.5$ |
|---|---|---|---|---|---|---|---|---|---|---|---|
| **CDVAE (ours)** | **0.43±0.02** | **0.50±0.03** | **0.96±0.09** | **1.57±0.08** | **1.90±0.10** | **2.35±0.18** | **3.57±0.17** | **4.80±0.20** | **6.84±0.20** | **7.64±0.48** | **9.03±0.50** |
| **Causal CPC** | 0.43±0.01 | 0.50±0.03 | 1.08±0.08 | 2.49±0.14 | 2.98±0.09 | 4.98±0.21 | 5.91±0.29 | 10.15±0.38 | 12.25±0.49 | 15.65±0.56 | 19.39±0.59 |
| **Causal CPC (with substitute)** | 0.46±0.02 | 0.49±0.01 | 1.02±0.05 | 2.38±0.08 | 2.83±0.09 | 4.81±0.13 | 5.38±0.22 | 8.13±0.39 | 10.11±0.41 | 12.01±0.54 | 14.03±0.64 |
| **Causal CPC (oracle)** | 0.45±0.02 | 0.43±0.03 | 0.96±0.04 | 1.59±0.06 | 2.14±0.08 | 4.41±0.10 | 5.08±0.30 | 6.93±0.25 | 8.70±0.17 | 10.15±0.47 | 12.91±0.51 |
| **Causal Transformer** | 0.46±0.02 | 0.68±0.04 | 1.50±0.06 | 2.55±0.18 | 3.65±0.20 | 5.55±0.50 | 8.15±0.72 | 12.35±0.25 | 15.48±1.02 | 24.77±2.21 | 43.84±2.58 |
| **Causal Transformer (with substitute)** | 0.46±0.02 | 0.67±0.02 | 1.46±0.03 | 2.48±0.08 | 3.53±0.09 | 5.23±0.11 | 7.72±0.18 | 11.86±0.17 | 15.22±0.17 | 20.12±0.35 | 33.58±0.45 |
| **Causal Transformer (oracle)** | 0.46±0.02 | 0.60±0.03 | 1.48±0.03 | 2.35±0.06 | 3.30±0.07 | 5.11±0.09 | 7.34±0.13 | 11.55±0.25 | 16.98±0.29 | 18.64±0.31 | 28.45±0.33 |
| **G-Net** | 0.62±0.05 | 0.80±0.05 | 4.90±0.03 | 5.56±0.05 | 4.82±0.15 | 5.79±0.12 | 10.36±0.23 | 15.17±0.25 | 23.89±0.54 | 32.75±1.20 | 49.35±2.35 |
| **G-Net (with substitute)** | 0.56±0.04 | 0.75±0.01 | 3.61±0.02 | 4.99±0.20 | 4.27±0.18 | 5.50±0.15 | 8.34±0.64 | 13.55±0.97 | 17.97±1.83 | 19.25±2.04 | 40.21±2.10 |
| **G-Net (oracle)** | 0.48±0.02 | 0.69±0.03 | 3.10±0.05 | 4.36±0.08 | 4.45±0.12 | 5.28±0.17 | 8.28±0.23 | 13.10±0.50 | 17.47±0.65 | 16.22±0.95 | 35.35±1.77 |
| **CRN** | 0.53±0.02 | 0.68±0.03 | 1.63±0.04 | 2.94±0.11 | 5.14±0.17 | 6.66±0.19 | 9.08±0.25 | 11.93±0.37 | 16.54±0.65 | 18.68±0.67 | 29.66±1.12 |
| **CRN (with substitute)** | 0.48±0.01 | 0.65±0.01 | 1.51±0.02 | 2.56±0.18 | 3.98±0.21 | 6.05±0.35 | 6.81±0.65 | 8.50±1.16 | 13.81±0.76 | 16.23±0.73 | 26.11±1.41 |
| **CRN (oracle)** | 0.53±0.01 | 0.60±0.01 | 1.69±0.02 | 2.87±0.09 | 3.75±0.13 | 5.69±0.17 | 8.92±0.22 | 9.65±0.31 | 13.49±0.54 | 15.64±0.61 | 25.98±0.97 |
| **RMSN** | 0.57±0.02 | 0.67±0.02 | 1.60±0.03 | 2.67±0.05 | 4.31±0.15 | 5.58±0.17 | 7.40±0.32 | 11.25±0.57 | 15.01±0.89 | 19.41±1.13 | 25.07±0.97 |
| **RMSN (with substitute)** | 0.45±0.01 | 0.67±0.02 | 1.51±0.04 | 2.31±0.03 | 3.61±0.20 | 4.61±0.20 | 6.14±0.50 | 7.58±0.82 | 10.63±1.31 | 18.53±1.50 | 21.08±1.60 |
| **RMSN (oracle)** | 0.48±0.01 | 0.62±0.01 | 1.43±0.03 | 1.97±0.03 | 3.42±0.13 | 4.43±0.15 | 6.00±0.27 | 7.31±0.34 | 9.20±0.48 | 17.06±0.83 | 19.32±1.26 |

**Robustness to Number of Prior Components** The Table 8 gives detailed results summarized in Figure 9a that assess the sensitivity of CDVAE to the variation of the prior cluster numbers $K$.

Table 8: Results of CDVAE when varying the number of components $K$ of the prior. The study is conducted on the synthetic data and is reported by PEHE. Smaller is better.

| Model | $\gamma_{(1)}^{YU}=0$ | $\gamma_{(1)}^{YU}=0.25$ | $\gamma_{(1)}^{YU}=0.5$ | $\gamma_{(1)}^{YU}=0.75$ | $\gamma_{(1)}^{YU}=1$ | $\gamma_{(1)}^{YU}=1.25$ | $\gamma_{(1)}^{YU}=1.5$ | $\gamma_{(1)}^{YU}=1.75$ | $\gamma_{(1)}^{YU}=2$ | $\gamma_{(1)}^{YU}=2.25$ | $\gamma_{(1)}^{YU}=2.5$ |
|---|---|---|---|---|---|---|---|---|---|---|---|
| **CDVAE (ours)** | **0.43±0.02** | **0.50±0.03** | **0.96±0.09** | **1.57±0.08** | **1.90±0.10** | **2.35±0.18** | **3.57±0.17** | **4.80±0.20** | **6.84±0.20** | **7.64±0.48** | **9.03±0.50** |
| **CDVAE ($K=2$)** | 0.43±0.01 | 0.50±0.01 | 0.96±0.03 | 1.52±0.04 | 2.07±0.07 | 2.51±0.14 | 4.10±0.41 | 4.44±0.43 | 6.98±0.38 | 7.90±0.33 | 8.88±0.45 |
| **CDVAE ($K=5$)** | 0.42±0.01 | 0.48±0.02 | 0.93±0.02 | 1.41±0.11 | 2.02±0.03 | 2.52±0.06 | 3.27±0.22 | 4.25±0.45 | 6.32±0.25 | 7.97±0.31 | 8.12±0.48 |
| **CDVAE ($K=8$)** | 0.40±0.01 | 0.46±0.02 | 0.96±0.06 | 1.57±0.03 | 2.13±0.04 | 2.59±0.07 | 3.20±0.26 | 4.15±0.45 | 6.63±0.48 | 6.93±0.25 | 8.91±0.19 |
| **CDVAE ($K=11$)** | 0.42±0.02 | 0.47±0.01 | 0.94±0.03 | 1.40±0.05 | 2.09±0.07 | 2.47±0.22 | 3.76±0.28 | 5.09±0.52 | 6.48±0.14 | 7.73±1.232 | 9.02±1.22 |

## C.3 Computational Scalability

We include additional experiments varying both the sequence length $T$ and the covariate dimension $d_x$. Empirically, CDVAE's one-step PEHE remains essentially flat when $T$ more than doubles ($75 \to 175$), while it increases modestly (about $+57\%$) when $d_x$ quintuples ($100 \to 500$). This behavior matches the model's complexity profile: GRU unrolling makes both run-time and memory scale linearly in $T$, and the IPM term—computed on a fixed-size batch of 16-dimensional representations—costs $\mathcal{O}(b^2)$ in batch size $b$ but does not depend on $d_x$. Only the first dense projection and GRU input weights scale with $d_x$, explaining the near-linear slowdown.

| Metric | $T = 75$ | $T = 100$ | $T = 125$ | $T = 150$ | $T = 175$ |
|---|---|---|---|---|---|
| PEHE | $1.09 \pm 0.09$ | $0.98 \pm 0.15$ | $1.00 \pm 0.20$ | $0.98 \pm 0.21$ | $0.97 \pm 0.09$ |
| Wall-clock (min) | $35 \pm 3$ | $57 \pm 3$ | $82 \pm 2$ | $95 \pm 3$ | $105 \pm 3$ |

Table 9: Performance of CDVAE as sequence length $T$ increases.

| Metric | $d_x = 100$ | $d_x = 200$ | $d_x = 300$ | $d_x = 400$ | $d_x = 500$ |
|---|---|---|---|---|---|
| PEHE | $1.09 \pm 0.09$ | $1.51 \pm 0.03$ | $1.60 \pm 0.04$ | $1.71 \pm 0.05$ | $1.71 \pm 0.05$ |
| Wall-clock (min) | $35 \pm 3$ | $33 \pm 2$ | $34 \pm 2$ | $32 \pm 2$ | $33 \pm 2$ |

Table 10: Performance of CDVAE as covariate dimension $d_x$ increases.

# D Experiments on MIMIC-III data

## D.1 Description of the MIMIC-III semi-synthetic model Model

A cohort of 2,000 patients is extracted from the MIMIC-III dataset, with the simulation setup proposed by Melnychuk et al. (2022) extending the model introduced in Schulam & Saria (2017). Let $d_y$ denote the dimension of the outcome variable. For multiple outcomes, untreated outcomes, denoted as $\mathbf{Z}_t^{j,(i)}$ for $j = 1, \ldots, d_y$, are generated for each patient $i$ in the cohort. The generation process is defined as:

$$\mathbf{Z}_t^{j,(i)} = \underbrace{\alpha_S^j \mathbf{B}\text{-spline}(t) + \alpha_g^j g^{j,(i)}(t)}_{\text{endogenous}} + \underbrace{\alpha_f^j f_Z^j \left( \mathbf{X}_t^{(i)} \right)}_{\text{exogenous}} + \underbrace{\varepsilon_t}_{\text{noise}}, \tag{35}$$

where the B-spline $\mathbf{B}\text{-spline}(t)$ models the endogenous component, $g^{j,(i)}(\cdot)$ is sampled independently for each patient from a Gaussian process with a Matérn kernel, and $f_Z^j(\cdot)$ is sampled from a Random Fourier Features (RFF) approximation of a Gaussian process.

To introduce confounding into the assignment mechanism, current time-varying covariates are incorporated via a random function $f_Y^l(\mathbf{X}_t)$ and the average of a subset of the previous $T_l$ treated outcomes, $\bar{A}_{T_l}(\overline{\mathbf{Y}}_{t-1})$. For $d_a$ binary treatments $\mathbf{A}_t^l$, where $l = 1, \ldots, d_a$, the treatment assignment mechanism is modeled as:

$$p_{\mathbf{A}_t^l} = \sigma \left( \gamma_A^l \bar{A}_{T_l} \left( \overline{\mathbf{Y}}_{t-1} \right) + \gamma_X^l f_Y^l \left( \mathbf{X}_t \right) + b_l \right),$$

$$\mathbf{A}_t^l \sim \text{Bernoulli} \left( p_{\mathbf{A}_t^l} \right),$$

where $\sigma(\cdot)$ denotes the sigmoid function.

Static features $\mathbf{U}$, such as gender and ethnicity, which are categorical, are one-hot encoded. A random Singular Value Decomposition (SVD) transformation $f_E(\cdot)$ is then applied to $\mathbf{U}$, retaining all singular components. Subsequently, treatments are applied to the untreated outcomes using the following expression:

$$E^j(t) = \sum_{i=t-w^l}^{t} \frac{\min_{l=1,\ldots,d_a} 1\!\!1_{[\mathbf{A}_i^l=1]} p_{\mathbf{A}_i^l} \beta_{lj} + \left| \sum_{k=1}^{d_U} f_E(U)_k \right|}{(w^l - i)^2}, \tag{36}$$

where $w^l$ is the treatment window.

The final outcome combines the treatment effect and the untreated simulated outcome:

$$Y_t^j = Z_t^j + E^j(t). \tag{37}$$

## D.2 Additional results

**Results of baselines on the semi-synthetic MIMIC-III**   The following Table 11 provides the detailed results responsible for Figures 4 related to baselines with the three different approaches.

Table 11: Results on the MIMIC-III data reported by PEHE. Smaller is better.

| Model | PEHE |
|---|---|
| **CDVAE (ours)** | **17.63±0.25** |
| **Causal CPC** | 19.27±0.25 |
| **Causal CPC (with substitute)** | 18.45±0.26 |
| **Causal CPC (oracle)** | 17.98±0.21 |
| **Causal Transformer** | 19.68±0.20 ± |
| **Causal Transformer (with substitute)** | 18.58± 0.21 |
| **Causal Transformer (oracle)** | 17.91 ± 0.20 |
| **G-Net** | 19.75±0.21 |
| **G-Net (with substitute)** | 18.60±0.25 |
| **G-Net (oracle)** | 17.95± 0.23 |
| **CRN** | 19.80±0.23 |
| **CRN (with substitute)** | 18.58±0.22 |
| **CRN (oracle)** | 17.91± 0.21 |
| **RMSN** | 19.85±0.25 |
| **RMSN (with substitute)** | 18.66±0.23 |
| **RMSN (oracle)** | 18.01± 0.19 |

# E   Algorithmic Details

## E.1   Algorithms Description

In this appendix, we provide the algorithmic details for CDVAE. First, we describe the approximation of the Integral Probability Metric (IPM) term used in Eq. (24). The IPM term aims to reduce covariate imbalance between the treatment and control groups by quantifying the dissimilarity between their distributions. To calculate the IPM, we use the Sinkhorn-Knopp algorithm (Sinkhorn, 1967) and the Wasserstein distance computation algorithm (Algorithm 3 in Cuturi & Doucet (2014)). The Wasserstein distance computation (Algorithm 2) calculates the pairwise distances between data points and constructs a kernel matrix using a regularization parameter. It then computes the row and column marginals based on the weights assigned to each data point. The Sinkhorn-Knopp algorithm (Algorithm 1), integrated within the Wasserstein distance computation, is used to compute the optimal transport matrix. Finally, the Wasserstein distance is obtained by summing the products of the optimal transport matrix and the pairwise distances. Computing the Wasserstein distance at each time step and for every batch is computationally expensive. To accelerate training, we compute it for a subsample of the time indices, sampled randomly at each batch. Empirically, this approach is sufficient for maintaining model performance. Specifically, we sample 10% of the time indices at each batch, corresponding to $m = \lfloor \frac{T}{10} \rfloor$ time steps, as shown in Algorithm 3.

## E.2   Computational complexity analysis of CDVAE

We provide both a theoretical and a practical computational complexity analysis. The training complexity of CDVAE can be summarized in the following proposition.

**Proposition 19 (Per-epoch complexity)** *For a mini-batch of size $b$, sequence length $T$, raw covariate dimension $d_x$, GRU hidden size $H$, and fixed representation width $d_\phi(= 16)$, let $\varepsilon$ denote the entropic regularization strength in the Sinkhorn algorithm for the IPM term. Then, one training epoch of CDVAE*

---

**Algorithm 1** Sinkhorn-Knopp Algorithm

---

**Require:** Kernel matrix $K \in \mathbb{R}^{n_t \times n_c}$
**Require:** Row marginal vector $a \in \mathbb{R}^{n_t}$, Column marginal vector $b \in \mathbb{R}^{n_c}$
**Ensure:** Optimal transport matrix $T \in \mathbb{R}^{n_t \times n_c}$
 1: Initialize transport matrix $T^{(0)}$ with all entries set to 1
 2: Set iteration counter $k \leftarrow 0$
 3: **while** not converged **do**
 4:      Update row scaling vector $u \in \mathbb{R}^{n_t}$
 5:      $u_i \leftarrow \frac{a_i}{\sum_{j=1}^{n_c} K_{ij} T_{ij}^{(k)}}$
 6:      Update column scaling vector $v \in \mathbb{R}^{n_c}$
 7:      $v_j \leftarrow \frac{b_j}{\sum_{i=1}^{n_t} K_{ij} T_{ij}^{(k)}}$
 8:      Update transport matrix $T^{(k+1)}$
 9:      $T_{ij}^{(k+1)} \leftarrow \frac{u_i K_{ij} v_j}{\sum_{i'=1}^{n_t} \sum_{j'=1}^{n_c} u_{i'} K_{i'j'} v_{j'}}$
10:      Increment iteration counter $k \leftarrow k + 1$
11:      **if** convergence criterion met **then**
12:         **Return** $T^{(k+1)}$
13:      **end if**
14: **end while**

---

**Algorithm 2** Weighted Wasserstein Distance Computation

---

**Require:** Batch $\mathcal{B} = \{\{\omega_{it}, y_{it}, \mathbf{x}_{it}\}_{t=1}^T, i = 1, \ldots, |\mathcal{B}|\}$
**Require:** Representation learner $\Phi$
**Require:** Weights vectors $\alpha_\Phi(\mathbf{h}_t, \omega)$
**Require:** Regularization parameter $\lambda$
 1: **for** $t \in \{1, 2, \ldots, T\}$ **do**
 2:      Compute $n_t^{(t)} \leftarrow \sum_{i \in \mathcal{B}} W_{it}$, $n_c^{(t)} \leftarrow \sum_{i \in \mathcal{B}} (1 - W_{it})$
 3:      Compute pairwise distances matrix $M \in \mathbb{R}^{n_t^{(t)} \times n_c^{(t)}}$   $\triangleright$ $M_{ij}^{(t)} = \|\mathbf{H}_{it} - \mathbf{H}_{jt}\|_{L_2}$, $\forall i, j \in \mathcal{B}$; $W_{it} = 1$ and $W_{jt} = 0$
 4:      Initialize kernel matrix $K \in \mathbb{R}^{n_t^{(t)} \times n_c^{(t)}}$ such that $K_{ij}^{(t)} \leftarrow e^{-\lambda M_{ij}^{(t)}}$
 5:      Compute row marginal vector $a^{(t)} \in \mathbb{R}^{n_t}$ such that $a_i^{(t)} \leftarrow \frac{\alpha_\Phi(\mathbf{h}_{it}, 1)}{\sum_{k=1, W_{kt}=1} \alpha_\Phi(\mathbf{h}_{it}, 1)}$
 6:      Compute column marginal vector $b^{(t)} \in \mathbb{R}^{n_c}$ such that $b_j^{(t)} \leftarrow \frac{\alpha_\Phi(\mathbf{h}_{it}, 0)}{\sum_{k=1, W_{kt}=0} \alpha_\Phi(\mathbf{h}_{it}, 0)}$
 7:      Compute optimal transport matrix $T^{(t)} \in \mathbb{R}^{n_t^{(t)} \times n_c^{(t)}}$:
 8:         $T^{(t)} \leftarrow$ Sinkhorn-Knopp$(K^{(t)}, a^{(t)}, b^{(t)})$
 9:      Compute Wasserstein distance $D_t \leftarrow \sum_{i=1}^{n_t} \sum_{j=1}^{n_c} T_{ij}^{(t)} M_{ij}^{(t)}$
10: **end for**
11: **Return** $\sum_{t=1}^T D_t$

---

---

**Algorithm 3** Pseudo-code for training CDVAE

---

**Require:** Training Data $\mathcal{D}_T = \{\{w_{it}, y_{it}, \mathbf{x}_{it}\}_{t=1}^T, i = 1, \ldots, n\}$
**Require:** CDVAE parameters $\phi$, $\theta_y$, $\theta_\omega$, $\Phi$, $\sigma$, Optimizer parameters
 1: **for** $p \in \{1, \ldots, \text{epoch}_{\max}\}$ **do**
 2:     **for** batch $\mathcal{B} = \{\{w_{it}, y_{it}, \mathbf{x}_{it}\}_{t=1}^T, i = 1, \ldots, |\mathcal{B}|\}$ **do**
 3:         Compute approximate posterior $q_\phi(z \mid y_{\leq T}, \mathbf{x}_{\leq T}, \omega_{\leq T})$
 4:         Sample latent variables $z$ from $q_\phi(z \mid y_{\leq T}, \mathbf{x}_{\leq T}, \omega_{\leq T})$
 5:         Compute representation $\Phi(\mathbf{H}_t)$ for $t = 1, \ldots, T$.
 6:         Compute $\text{ELBO}(\theta, \phi, \Phi, \sigma)$.
 7:         Compute $\mathcal{L}_{\text{DistM}}(\phi)$.
 8:         Choose $t_1, \ldots, t_m \sim \mathcal{U}([1, T])$ compute IPM term as :

$$\mathcal{L}_{\text{IPM}} = \sum_{i=1}^m \text{IPM}_G \left( g_{\theta_\omega, \Phi}(\cdot \mid W_{t_i} = 1),\, g_{\theta_\omega, \Phi}(\cdot \mid W_{t_i} = 0) \right)$$

 9:         Compute total loss $\mathcal{L}_{\text{tot}} = \text{ELBO}(\theta, \phi, \Phi, \sigma) + \lambda_{\text{IPM}} \mathcal{L}_{\text{IPM}}(\theta_\omega, \Phi) + \lambda_{\text{DistM}} \mathcal{L}_{\text{DistM}}(\phi)$.
10:         Update parameters related to the total loss

$$[\theta, \phi, \Phi, \sigma] \leftarrow [\theta, \phi, \Phi, \sigma] - \mu \left( \frac{\partial \mathcal{L}_{\text{tot}}(\theta, \theta_\omega, \phi, \Phi, \sigma)}{\partial [\theta, \phi, \Phi, \sigma]} \right)$$

11:         Compute binary cross-entropy loss for the propensity network $\mathcal{L}_W(\theta_\omega, \Phi)$. and update parameters

$$\theta_\omega \leftarrow \theta_\omega - \mu_W \left( \frac{\partial \mathcal{L}_W(\theta_\omega, \Phi)}{\partial \theta_\omega} \right)$$

12:     **end for**
13: **end for**
14: **Return** Trained CDVAE model

---

*requires*

$$\underbrace{O\big(nT\,H(H+d_x)\big)}_{GRU\ forward/back\text{-}prop} + \underbrace{O\big(n\,d_x\,d_\phi\big)}_{\Phi\text{-}projection} + \underbrace{O\big(m\,b^2\log b/\varepsilon^2\big)}_{\substack{Sinkhorn\ IPM\\ m=\lceil 0.1T\rceil}},$$

*and stores $O(T\,H + n\,b)$ activations.*

**Discussion.**   The first two terms are *linear in the sequence length $T$* and *(near-)linear in the covariate dimension $d_x$* because the GRU unrolls once per time step and the projection layer is dense. The third term corresponds to the entropically regularized Sinkhorn solver; its cost is *quadratic only in the batch size $b$* and independent of $d_x$ once the feature map has been compressed to $d_\phi$.

Since we evaluate the IPM on only $m = \lceil 0.1T \rceil$ randomly chosen time points, the overall runtime grows linearly with $T$ and almost linearly with $d_x$ for realistic batches ($b \leq 1024$). The sole quadratic factor arises from the $b^2$ dependence inside Sinkhorn.

**Practical results.**   On the simulation dataset, Table 12 reports the mean training time (in minutes) of all models, averaged over 10 random seeds. All experiments were run on a single NVIDIA Tesla T4 GPU.

| Model | Training time (min) |
|---|:---:|
| CDVAE (ours) | $35 \pm 3$ |
| Causal CPC | $8 \pm 2$ |
| CT | $27 \pm 2$ |
| G-Net | $12 \pm 2$ |
| CRN | $7 \pm 2$ |
| RMSN | $6 \pm 2$ |

Table 12: Mean training times of different models on the simulation dataset (10 seeds, $1\times$ NVIDIA Tesla T4).

**Interpretation.**   CRN, RMSN, and CPC lack the IPM term and thus train faster, while the Causal Transformer incurs higher cost due to transfer blocks. CDVAE's additional complexity comes from explicit distributional balancing over $\Phi(\mathbf{H}_t)$, which we found necessary for stable ACATE estimation. The random-time subsampling of representations keeps the added cost manageable in practice.

## F   Models hyperparameters

We use Pytorch (Paszke et al., 2019) and Pytorch Lightening (Falcon & team, 2019) to implement CDVAE and all baselines. For the selection of hyperparameters, we fine-tuned our models using a random grid search. We use the weighted reconstruction error as the selection criterion for CDVAE. We do not use a metric related to the quality of estimating ACATEs as a selection criterion; a choice cannot be used in a real scenario because ACATEs are usually unavailable. It is still rather an open problem how to design a criterion for causal cross-validation or hyperparameters tuning for a causal model. Many proxies are used in the literature, including loss over the factual outcomes (Lim, 2018; Hassanpour & Greiner, 2019a; Bica et al., 2020a;b), one nearest-neighbor imputation (Shalit et al., 2017; Johansson et al., 2022), for the counterfactual outcome, influence functions (Alaa & Van Der Schaar, 2019), rank-preserving causal cross-validation (Schuler et al., 2018), Robinson residual decomposition (Nie & Wager, 2021; Lu et al., 2020). In this work, and since we intend to search for the best regularization parameters ($\lambda_{IPM}$, $\lambda_{MM}$) among other parameters related to CDVAE architecture and optimization, we chose not to consider the total loss as this may bias the choice of ($\lambda_{IPM}$, $\lambda_{MM}$) toward small values. We, therefore, use the weighted reconstruction error as an objective function for fine-tuning. Similarly, we fine-tune Causal CPC, Causal Transformer, CRN, and RMSM using the loss over the factual response as a criterion.

We report in the following tables the search space of hyperparameters for all baselines.

Table 13: Hyper-parameters search range for RMSN

| Model | Sub-model | Hyperparameter | Synthetic data | MIMIC III |
|---|---|---|---|---|
| RMSNs | Propensity Treatment Network | LSTM layers | 1 | 1 |
| | | Learning rate | $0.01, 0.005, 0.001, 0.0001$ | $0.01, 0.005, 0.001, 0.0001$ |
| | | Batch size | $32, 64, 128$ | $32, 64, 128$ |
| | | LSTM hidden units | $6, 8, \ldots, 12, 14$ | $4, 6, \ldots, 20$ |
| | | LSTM dropout rate | - | - |
| | | Max gradient norm | $0.5, 1, 2$ | $0.5, 1, 2$ |
| | | Early Stopping (min delta) | $0.001$ | $0.001$ |
| | | Early Stopping (patience) | 10 | 30 |
| | Propensity History Network | LSTM layers | 1 | 1 |
| | | Learning rate | $0.01, 0.005, 0.001, 0.0001$ | $0.01, 0.005, 0.001, 0.0001$ |
| | | Batch size | $32, 64, 128$ | $64, 128, 256$ |
| | | LSTM hidden units | $6, 8, \ldots, 12, 14$ | $4, 6, \ldots, 30$ |
| | | LSTM dropout rate | - | - |
| | | Early Stopping (min delta) | $0.001$ | $0.0001$ |
| | | Early Stopping (patience) | 10 | 30 |
| | Encoder | LSTM layers | 1 | 1 |
| | | Learning rate | $0.01, 0.005, 0.001, 0.0001$ | $0.01, 0.005, 0.001, 0.0001$ |
| | | Batch size | $32, 64, 128, 256$ | $32, 64, 128$ |
| | | LSTM hidden units | $6, 8, \ldots, 18, 20$ | $6, 8, \ldots, 18, 20$ |
| | | LSTM dropout rate | - | - |
| | | Early Stopping (min delta) | $0.001$ | $0.001$ |
| | | Early Stopping (patience) | 10 | 30 |
| | Decoder | LSTM layers | 1 | 1 |
| | | Learning rate | $0.01, 0.005, 0.001, 0.0001$ | $0.01, 0.005, 0.001, 0.0001$ |
| | | Batch size | $32, 64, 128, 256$ | $128, 512, 1024$ |
| | | LSTM hidden units | $6, 8, \ldots, 18, 20$ | $6, 8, \ldots, 18, 20$ |
| | | LSTM dropout rate | - | - |
| | | Max gradient norm | $0.5, 1, 2$ | $0.5, 1, 2$ |
| | | Early Stopping (min delta) | $0.001$ | $0.0001$ |
| | | Early Stopping (patience) | 10 | 30 |

Table 14: Hyper-parameters search range for CRN

| Model | Sub-model | Hyperparameter | Synthetic data | MIMIC III) |
|---|---|---|---|---|
| CRN | Encoder | LSTM layers | 1 | 1 |
| | | Learning rate | $0.01, 0.005, 0.001, 0.0001$ | $0.01, 0.005, 0.001, 0.0001$ |
| | | Batch size | $32, 64, 128, 256$ | $32, 64, 128$ |
| | | LSTM hidden units | $6, 8, \ldots, 18, 20$ | $6, 8, \ldots, 18, 20$ |
| | | LSTM dropout rate | - | - |
| | | BR size | $6, 8, \ldots, 18, 20$ | $6, 8, \ldots, 18, 20$ |
| | | Early Stopping (min delta) | $0.001$ | $0.001$ |
| | | Early Stopping (patience) | 10 | 30 |
| | Decoder | LSTM layers | 1 | 1 |
| | | Learning rate | $0.01, 0.005, 0.001, 0.0001$ | $0.01, 0.005, 0.001, 0.0001$ |
| | | Batch size | $128, 256, 512$ | $256, 512, 1024$ |
| | | LSTM hidden units | $6, 8, \ldots, 18, 20$ | $6, 8, \ldots, 18, 20$ |
| | | LSTM dropout rate | - | - |
| | | BR size | $6, 8, \ldots, 18, 20$ | $6, 8, \ldots, 18, 20$ |
| | | Early Stopping (min delta) | $0.001$ | $0.001$ |
| | | Early Stopping (patience) | 10 | 30 |

Table 15: Hyper-parameters search range for G-Net

| Hyperparameter | Cancer simulation | MIMIC III (SS) |
|---|---|---|
| LSTM layers | 1 | 1 |
| Learning rate | $0.01, 0.005, 0.001, 0.0001$ | $0.01, 0.005, 0.001, 0.0001$ |
| Batch size | $32, 64, 128$ | $32, 64, 128$ |
| LSTM hidden units | $6, 8, \ldots, 18, 20$ | $6, 8, \ldots, 18, 20$ |
| FC hidden units | $6, 8, \ldots, 18, 20$ | $6, 8, \ldots, 18, 20$ |
| LSTM dropout rate | - | - |
| R size | $6, 8, \ldots, 18, 20$ | $4, 6, \ldots, 30$ |
| MC samples | 50 | 50 |
| Early Stopping (min delta) | $0.001$ | $0.001$ |
| Early Stopping (patience) | 10 | 30 |

Table 16: Hyper-parameters search range for Causal Transformer

| Hyperparameter | Cancer simulation | MIMIC III (SS) |
|---|---|---|
| Transformer blocks | 1 | 1 |
| Learning rate | $0.01, 0.005, 0.001, 0.0001$ | $0.01, 0.005, 0.001, 0.0001$ |
| Batch size | $32, 64, 128$ | $32, 64, 128$ |
| Attention heads | 2 | 2 |
| Transformer units | $4, 6, \ldots, 20$ | $4, 6, \ldots, 20$ |
| LSTM dropout rate | - | - |
| BR size | $6, 8, \ldots, 18, 20$ | $4, 6, \ldots, 20$ |
| FC hidden units | $6, 8, \ldots, 18, 20$ | $4, 6, \ldots, 20$ |
| Sequential dropout rate | $0.1, 0.2, 0.3$ | $0.1, 0.2, 0.3$ |
| Max positional encoding | 15 | 20 |
| Early Stopping (min delta) | 0.001 | 0.001 |
| Early Stopping (patience) | 10 | 30 |

Table 17: Hyper-parameters search range for Causal CPC

| Model | Sub-model | Hyperparameter | Cancer simulation | MIMIC III (SS) |
|---|---|---|---|---|
| Causal CPC | Encoder | GRU layers | 1 | 1 |
| | | Learning rate | $0.01, 0.005, 0.001, 0.0001$ | $0.01, 0.005, 0.001, 0.0001$ |
| | | Batch size | $32, 64, 128$ | $64, 128, 256$ |
| | | GRU hidden units | $6, 8, \ldots, 18, 20$ | $6, 8, \ldots, 18, 20$ |
| | | GRU dropout rate | - | - |
| | | Local features (LF) size | $6, 8, \ldots, 18, 20$ | $4, 6, \ldots, 20$ |
| | | Context Representation (CR) size | $6, 8, \ldots, 18, 20$ | $4, 6, \ldots, 20$ |
| | | Early Stopping (min delta) | 0.001 | 0.001 |
| | | Early Stopping (patience) | 10 | 30 |
| | Decoder | GRU layers | 1 | 1 |
| | | Learning rate (decoder w/o treatment sub-network) | $0.01, 0.005, 0.001, 0.0001$ | $0.01, 0.005, 0.001, 0.0001$ |
| | | Learning rate (encoder fine-tuning) | $0.001, 0.0005, 0.0001, 0.00005$ | $0.001, 0.0005, 0.0001, 0.00005$ |
| | | Learning rate (treatment sub-network) | $0.05, 0.01, 0.005, 0.0001$ | $0.05, 0.01, 0.005, 0.0001$ |
| | | Batch size | $32, 64, 128$ | $32, 64, 128$ |
| | | GRU hidden units | CR size | CR size |
| | | GRU dropout rate | - | - |
| | | BR size | CR size | CR size |
| | | GRU layers (Treat Encoder) | 1 | 1 |
| | | GRU hidden units (Treat Encoder) | 6 | 6 |
| | | FC hidden units | $6, 8, \ldots, 18, 20$ | $4, 6, \ldots, 20$ |
| | | Random time indices (m) | 10% | 10% |
| | | Early Stopping (min delta) | 0.001 | 0.001 |
| | | Early Stopping (patience) | 10 | 30 |

Table 18: Hyper-parameters search range for CDVAE

| Model | Sub-model | Hyperparameter | Synthetic data | MIMIC III |
|---|---|---|---|---|
| CDVAE | Inference Network | GRU layers | 1 | 1 |
| | | GRU hidden units | $6, 8, \ldots, 12, 14$ | $4, 6, \ldots, 20$ |
| | | GRU dropout rate | - | - |
| | | Latent dim of $\mathbf{z}$ | 0.001 | 0.001 |
| | Propensity Network | FC hidden units | $6, 8, \ldots, 12, 14$ | $4, 6, \ldots, 30$ |
| | Representation Learner | LSTM layers | 1 | 1 |
| | | GRU hidden units | $6, 8, \ldots, 18, 20$ | $6, 8, \ldots, 18, 20$ |
| | | GRU dropout rate | - | - |
| | | Dimension of representation | $6, 8, \ldots, 18, 20$ | $4, 6, \ldots, 20$ |
| | Decoder | FC hidden units | $(dim(\mathbf{z}) + dim(\Phi(\mathbf{H}_t)))/2$ | $(dim(\mathbf{z}) + dim(\Phi(\mathbf{H}_t)))/2$ |
| | Global | Learning rate (w/o propensity network) | $0.01, 0.005, 0.001, 0.0001$ | $0.01, 0.005, 0.001, 0.0001$ |
| | | Learning rate (propensity network) | $0.01, 0.005, 0.001, 0.0001$ | $0.01, 0.005, 0.001, 0.0001$ |
| | | Batch size | $32, 64, 128, 256$ | $128, 512, 1024$ |
| | | Max gradient norm | $0.5, 1, 2$ | $0.5, 1, 2$ |
| | | Number of components in Prior | $2, 4, \ldots, 18, 20$ | $2, 4, \ldots, 20$ |

# G  The Neural Architecture of CDVAE

The extended neural architecture of CDVAE comprises multiple components which we did not explicit in Section 4.4. We first begin by detailing neural network functions related to the generative model. The Table 20 outlines the architecture for the Representation Learner $\Phi$ which encodes the context history, Table 19 presents the identical architecture for both $f_{\theta_y^1}$ and $f_{\theta_y^0}$ responsible for generating the two potential outcomes. Meanwhile, Table 21 illustrates the design of propensity network $e_{\theta_\omega}(.)$ built on the top of the shared representation. Lastly, Tables 22 depict the architecture used to learn both the mean and covariance matrix for the approximate posterior assumed to be Gaussian.

| Inputs: $\{\Phi(\mathbf{h}_t)\}_{1 \leq t \leq T}$, $\mathbf{z}$ |
|:---:|
| Concatenate: $[\Phi(\mathbf{h}_t), \mathbf{z}]_{1 \leq t \leq T}$ |
| Linear Layer |
| Weight Normalization |
| ELU |
| Linear Layer |
| Weight Normalization |
| Linear Layer |
| Output: $\{\hat{Y}_{t+1}(\omega)\}_{1 \leq t \leq T-1}$ |

Table 19: Architecture of the outcome model prediction, i.e., the decoder.

| Inputs: $\{y_t, \mathbf{x}_t, \omega_t\}_{1 \leq t \leq T}$ |
|:---:|
| Concat: $[y_t, \mathbf{x}_t, \omega_t]_{1 \leq t \leq T}$ |
| GRU layer |
| Linear Layer |
| Tanh |
| Outputs: $\{\Phi(\mathbf{h}_t)\}_{1 \leq t \leq T}$ |

Table 20: Architecture: representation learner $\phi$ of CDVAE

| Inputs: $\{\Phi(\mathbf{h}_t)\}_{1 \leq t \leq T}$ |
|:---:|
| Linear Layer |
| ELU |
| Sigmoid |
| Output: $\{\hat{W}_{t+1}(\omega)\}_{1 \leq t \leq T-1}$ |

Table 21: Architecture: propensity network $e_{\theta_\omega}(.)$.

| Inputs: $\{y_t, \mathbf{x}_t, \omega_t\}_{1 \leq t \leq T}$ | |
|:---:|:---:|
| Concat: $[y_t, \mathbf{x}_t, \omega_t]_{1 \leq t \leq T}$ | |
| GRU layer | |
| Linear Layer | Linear Layer |
| $\Sigma_{\phi_3}(\mathbf{g}_T)$ | $\mu_{\phi_2}(\mathbf{g}_T)$ |

Table 22: Inference network $q_\phi(\mathbf{z} \mid y_{\leq T}, \mathbf{x}_{\leq T}, \omega_{\leq T})$

