# OpenReview forum: "Causal Dynamic Variational Autoencoder for Counterfactual Regression in Longitudinal Data"
_TMLR — Accepted by TMLR_

### Review · Reviewer_CXvK · 2025-05-24

**Summary Of Contributions:**

Its been at least a couple of years since working in this particular space, so I defer to the other reviewers in terms of some of the more technically challenging components of this contribution. Nonetheless, I recall that much work at the intersection between causal inference and variational approaches concerned unobserved confounders. Whilst confounders present more problematically (in terms of bias), it is interesting to see the current authors tackling something a bit different: unobserved adjustment variables or, if I'm not mistaken, these are also referred to as unobserved risk factors. i.e. these are factors which affect the outcome sequence, but not the treatment assignment. If successful, such a method would be able to estimate the effect of treatment over time with more precision, as the method would be accounting for further sources of outcome heterogeneity.

The authors approach this problem with a dynamic variational autoencoder CDVAE. In addition, they prove theoretical validity under certain Markov assumptions, and demonstrate the effectiveness of the method empirically.

**Audience:**

Yes

**Claims And Evidence:**

Yes

**Requested Changes:**

*Questions:*

1) I see a lot of switching between ITE and CATE/ACATE. Whilst I found Section 3 very clear in terms of its definition of ACATE on page 5 (eqs. 2-4), I found the switch to ITE in 4.3 abrupt. I remember that Shalit et al.'s work on CFR net seemed to confuse CATE with ITE, and this co-occurred with a flurry of causal-ML papers which did the same thing. Can the authors clarify (a) the difference , and (b) perhaps reference the ITE in Section 3 if an ITE is indeed the quantity they seek to estimate? Without wanting to be patrionizing, by my understanding,  the expected value of an outcome for an individual E[E[Y_i|W=0, X_i]- E[Y_i|W=1, X_i]] is a CATE, where the condition achieves a specific stratification over an observed set of covariates, rather than an ITE. I assume the authors will explain that the transition from CATE to ITE is achieved via the near-deterministic regime, but I'm not 100% sure this is exactly what is happening.

2) Can the authors comment on their intuition about Figure 3 - why does CDVAE perform so much better than other methods in the oracle and substitute conditions?

3) To what extent is the method sensitive to consistency between the distribution of U and the choice of the prior for Z?


Minor:
- I wonder if the authors should include G-transformer https://arxiv.org/abs/2406.05504 as another comparative method in their related work section.

**Strengths And Weaknesses:**

1) The paper is very well written and easy to follow.

2) The idea is intuitive and the approach to proofs is thorough (although I am unable to comment on their correctness).

3) The latent substitutes/representations for the unobserved adjustment variables can be used in combination with existing methods. I think leveraging representations from one method to augment another is an obvious but often overlooked opportunity in many related works.

4) The results indicate consistent improvement of performance when utilising the additional representation.

5) The authors take a comprehensive approach to evaluations, including synthetic data, modified real-data, ablations, and a series of other checks. The comparisons include an accounting for the parameter counts associated with the modified comparison methods.

6) The authors include code attached as supplementary, although I have not evaluated/reproduced any of their experiments with it.

---

> ### Author Response · Authors · 2025-08-12
>
> Dear reviewer CXvK, thank you very much for your careful review and insightful comments. We kindly refer you to the following elements answering your requests.
>
> ## On the use of ITE/CATE.
> Thank you very much for this question. Many papers indeed, since Shalit's paper, used ITE and CATE terms almost interchangeably, especially papers falling into the theme of "counterfactual regression". Otherwise, with additional care, others used rather *Individualized Treatment Effect* to talk about CATE. We are happy to clarify further the difference in our work. Section 3 defines our target as the *Augmented CATE*:
>
> $$
> \tau\_{t}(\mathbf{h}\_{t}, \mathbf{u}) = \mathbb{E}(Y\_{t} \mid \mathbf{H}_t = \mathbf{h}_t, \mathbf{U} = \mathbf{u}, W\_{t} = 1) - \mathbb{E}(Y\_{t} \mid \mathbf{H}_t = \mathbf{h}_t, \mathbf{U} = \mathbf{u}, W\_{t} = 0).
> $$
>
> This is a *conditional expectation* and, therefore, still a CATE in its broader sense. In Section 4.3, we momentarily write “ITE” when we discuss plugging a *single realised vector* — either a draw $Z$ or its posterior mean $\mu_Z$ — into the learned conditional outcome model. That quantity
>
> $$
> \tau\_{t}(\mathbf{h}\_{it}, \mathbf{z}_i) = \mathbb{E}(Y\_{t}| \mathbf{H}\_{it} = \mathbf{h}\_{it}, \mathbf{Z}= \mathbf{z}_i, W\_{t} = 1) - \mathbb{E}(Y\_{t}| \mathbf{H}\_{t} = \mathbf{h}\_{it}, \mathbf{Z}= \mathbf{z}_i, W\_{t} = 0),
> $$
>
> is still an ACATE evaluated at the observed covariates of the subject $i$; it is *not* the fundamental (unobservable) individual treatment effect $Y\_{it}(1)-Y\_{it}(0)$. We invoke the near‐deterministic regime ($\sigma\to0^+$) precisely to guarantee — by Theorem 4.2 — that every realisation $\mathbf z_i$ collapses onto the same limit as the posterior mean. This addresses the “which $\mathbf Z$ do we choose?” concern and prevents posterior collapse observed in stochastic VAEs. You are completely right to doubt whether ITE is recovered in the near-deterministic regime, because it is actually not the case. The near-deterministic regime allows us to be sure that there is a one-to-one mapping between covariates (confounding plus adjustment) and ACATE, and thus allows, by extension, confounding theory to have consistency as a limit rather than as a starting hard assumption.
>
> We have now added:
> 1. Replace in 4.3 ITEs with *individualized ACATE* when discussing estimation for an individual $i$.
> 2. We think it would be better for the sake of clarity for any practitioner in the causal inference community to use in this paper the Individualized Treatment Effect (ITE). This suits this work better because we aim to estimate a treatment effect that is individualized as much as possible, given covariates. Standard CATE of Eq. 2. I am an individualized treatment effect and ACATE of Eqs 3–4 too, but it is more individualized given dependence on adjustment variables.
> 3. We also added a footnote in the Introduction to insist that we are not targeting the individual Treatment Effect: *"In this work, we do not target the fundamental and unobservable individual treatment effect $Y\_{it}(1)-Y\_{it}(0)$ but rather a treatment effect conditional on covariates whose heterogeneity is richer than that of a standard in the sense of ACATE."*
> ## Intuitions on why CDVAE performs better than other methods in the oracle and substitute conditions
>
> We explicitly adapted all baselines to avoid unfair advantages: we fed the learned substitute $\mathbf Z$ or oracle $\mathbf U$ *only* to the outcome head, matched parameter counts, and kept their balancing mechanisms (see “Adapting Baselines…”). Thus, the margin is *not* due to over-parameterizing competitors. Even so, CDVAE keeps an edge for three reasons:
>
> - **Joint latent–likelihood learning.** In the substitute setting, the same $\mathbf Z$ others consume is **co-trained** in CDVAE to maximize a weighted outcome likelihood, so $\mathbf Z$ is optimized to explain heterogeneity in $Y_t \mid (\mathbf H_t, W_t)$; using a fixed $\mathbf Z$ without co-training is weaker.
> - **Global static modifier.** CDVAE treats $\mathbf Z$ as a sequence-level latent that modulates all $t = 1{:}T$ outputs via the decoder, matching a static effect modifier with finite-order dynamics and leveraging deep latent-sequence models.
> - **Near-deterministic training preserves the latent signal.** Shrinking decoder variance avoids posterior collapse and yields a realization-invariant latent, stabilizing individualized ACATEs. Ablations (Tables 2–3) show fixing $\sigma = 1$ worsens PEHE, and baselines can surpass our model (e.g., Causal CPC on synthetic, Fig. 3b; multiple baselines on MIMIC, Fig. 4).

---

> ### Author Response · Authors · 2025-08-12
>
> ## On consistency between the distribution of U and the choice of the prior for Z
> We assessed sensitivity to the consistency between the prior $p(Z)$ and the true distribution of $U$ by varying the number of mixture components $K$ in the GMM prior well below and above the truth. On the synthetic data (true $K = 8$) we ran $K \in \{2, 5, 8, 11\}$; on MIMIC-III (unknown truth) we selected $K = 5$ via search and also tried $K \in \{2, 8, 11, 14\}$. Across all $\gamma_{(1)}^{YU}$ levels, PEHE curves are nearly indistinguishable — mean differences are small and error bars overlap almost everywhere — showing that CDVAE is robust to prior–truth mismatch in mixture complexity.
>
> We run a new experiment in which, in addition to missing adjustment variables, some time-varying confounders are missing in response to a request by Reviewer Z88p. We sweep the degree of unobserved confounding using a Rosenbaum-style sensitivity analysis, setting $\Gamma \in \{1, 1.5, \ldots, 5\}$ and letting the propensity change as a function of $\log \Gamma$. The contribution of $U$ to the outcomes is held fixed; increasing $\Gamma$ makes missing confounders more dominant in the outcome mechanism.
>
> We report here the **mutual information (MI)** between learned representation coordinates and the true cluster labels of $U$, averaged across latent dimensions. By “representation,” we mean learned $\mathbf{Z}$ for CDVAE and learned (balanced) representation for baselines.
>
> **Table – Mutual Information between learned representations and true cluster labels in $\mathbf{U}$ under a sensitivity analysis to missing confounding controlled by $\Gamma$. Bigger is better.**
>
> | **Model**          | $\Gamma=1$        | 1.5              | 2                | 2.5              | 3                | 3.5              | 4                | 4.5              | 5                |
> |--------------------|-------------------|------------------|------------------|------------------|------------------|------------------|------------------|------------------|------------------|
> | CDVAE              | $1.34 \pm 0.08$   | $1.31 \pm 0.10$  | $1.22 \pm 0.14$  | $1.42 \pm 0.15$  | $1.53 \pm 0.10$  | $1.48 \pm 0.04$  | $1.50 \pm 0.03$  | $1.48 \pm 0.08$  | $1.35 \pm 0.13$  |
> | Causal Transformer | $0.98 \pm 0.05$   | $0.90 \pm 0.04$  | $0.77 \pm 0.01$  | $0.72 \pm 0.03$  | $0.68 \pm 0.05$  | $0.64 \pm 0.01$  | $0.62 \pm 0.06$  | $0.57 \pm 0.01$  | $0.53 \pm 0.01$  |
> | RMSN               | $0.54 \pm 0.01$   | $0.84 \pm 0.01$  | $0.46 \pm 0.01$  | $0.42 \pm 0.01$  | $0.42 \pm 0.01$  | $0.38 \pm 0.02$  | $0.39 \pm 0.03$  | $0.35 \pm 0.02$  | $0.41 \pm 0.01$  |
> | G-Net              | $0.43 \pm 0.01$   | $0.39 \pm 0.02$  | $0.39 \pm 0.03$  | $0.37 \pm 0.01$  | $0.35 \pm 0.04$  | $0.35 \pm 0.01$  | $0.32 \pm 0.01$  | $0.33 \pm 0.01$  | $0.31 \pm 0.01$  |
> | CRN                | $0.89 \pm 0.06$   | $0.84 \pm 0.04$  | $0.71 \pm 0.03$  | $0.67 \pm 0.01$  | $0.61 \pm 0.02$  | $0.61 \pm 0.01$  | $0.59 \pm 0.03$  | $0.58 \pm 0.02$  | $0.51 \pm 0.01$  |
>
> **Why MI declines as $\Gamma$ increases.** In our sensitivity design, the contribution of $U$ to $Y$ is held fixed, while increasing $\Gamma$ amplifies the influence of missing time-varying confounders via a $\log \Gamma$ shift in propensities. As $\Gamma$ grows, outcomes become less informative about $U$ (lower signal-to-noise), so encoder representations recover less cluster information—especially for the baselines. CDVAE’s MI declines only mildly because $Z$ is learned as a global static modifier under the CMM($p$) model.
>
> **Interpretation.** CDVAE attains consistently higher MI than all adapted baselines. Some MI for the baselines is expected given the causal structure: $U$ affects the entire response series, and observed outcomes serve as proxies for $U$, so encoders that use past responses inevitably preserve some cluster signal. CDVAE is more efficient because it learns a dedicated substitute $Z$ (the variational posterior) and, under CMM($p$), explicitly exploits outcomes as high-signal proxies for $U$, thereby better capturing response heterogeneity driven by latent variables.

---

> ### Author Response · Authors · 2025-08-12
>
> ## (Minor) On G-Transformer
> G-transformer is a result of marrying G-Net and the causal transformer. It keeps the g-computation of G-Net but replaces its RNN structure with a transformer-based one. In the related work section, we added in the paragraph titled *"Causal Inference in Time-Varying Settings"*:
>
> > *"G-Transformer (Xiong et al., 2024) extends g-computation by using a Transformer to model covariate dynamics and Monte-Carlo rollouts for counterfactuals under dynamic regimes and is thus conceptually a combination of Causal Transformer and G-Net."*
>
> We implemented G-Transformer to inspect whether G-Net benefits from a transformer architecture in our setting. We adapted the transformer architecture of Causal Transformer to G-Net. We partially replicated the experiment over simulation data by varying the contribution $\gamma_{(1)}^{YU}$ of unobserved $\mathbf U$ factors in the treatment effect. We compare G-Net and G-Transformer in the following way:
>
> | **Model**       | 0                 | 0.25              | 0.5               | 0.75              | 1                 | 1.25              | 1.5               | 1.75              | 2                 | 2.25              | 2.5               |
> |-----------------|-------------------|-------------------|-------------------|-------------------|-------------------|-------------------|-------------------|-------------------|-------------------|-------------------|-------------------|
> | G-Net           | $0.62 \pm 0.05$   | $0.80 \pm 0.05$   | $4.90 \pm 0.03$   | $5.56 \pm 0.05$   | $4.82 \pm 0.15$   | $5.79 \pm 0.12$   | $10.36 \pm 0.23$  | $15.17 \pm 0.25$  | $23.89 \pm 0.54$  | $32.75 \pm 1.20$  | $49.35 \pm 2.35$  |
> | G-Transformer   | $2.47 \pm 0.03$   | $4.45 \pm 0.03$   | $12.09 \pm 0.04$  | $20.87 \pm 0.13$  | $31.22 \pm 0.12$  | $42.68 \pm 0.23$  | $55.19 \pm 0.36$  | $68.20 \pm 0.11$  | $81.80 \pm 0.11$  | $95.31 \pm 0.15$  | $110.21 \pm 0.35$ |
>
> The error in PEHE explodes quickly in G-Transformer as $\gamma_{(1)}^{YU}$ starts increasing, showing the harmful side of increasing the complexity of a model. In contrast, the number of observations is relatively modest (for training, 5000 in synthetic data and 1400 for MIMIC-III). To ensure a fair comparison of G-computation-based models, we keep the RNN architecture, i.e., the standard G-Net.

---

### Review · Reviewer_Z88p · 2025-06-02

**Summary Of Contributions:**

The paper introduces Causal Dynamic VAE (CDVAE) for estimating individual treatment effects (ITE) when there are unobserved static adjustment variables that affect the outcome sequence. Critical assumption is that these unobserved adjustment varibles do not confound the treatment.
The static (time-invariant) adjustment variables are somewhat justified through examples, such as genetic factors and environmen conditions.

The paper lays out the main challenges in working with longitudinal data and points out that the existing literature mostly addresses them under ``sequential ignorability’’ where all confounders are observed. For the missing covariates though, this common assumption is violated.
To address the challenge of missing covariates, the paper starts by answering the question: Under what conditions can a latent representation be a valid substitute for the unobserved adjustment variable such that the identifiability of CATE holds? After answering the question positively for a finite-order conditional Markov Process,  the paper addresses additional modeling points (e.g., using a Gaussian mixture model prior to account for the population structure caused by the unobserved adjustment variables).

An interesting component of the paper is studying VAE in the near-deterministic regime, and demonstrating that the treatment effect remains consistent regardless of the substitute. Studying this in causal inference is a nice contribution of this work, which has further implications regarding the related literature on deconfounders.

Finally,  the proposed method CDVAE is shown to outperform the baselines (adapted to the setting of the paper) for both synthetic and a semi-synthetic dataset. This includes showing that augmenting the input covariates with the inferred latent substitutes (of the unobserved adjustments) enhances the performance of the baseline approaches.

**Audience:**

Yes

**Claims And Evidence:**

Yes

**Requested Changes:**

One mid-level request (I’d prefer to see this added):
- Regarding the first weakness point. I suggest the authors to add some experiments with a small confounding effect. I wouldn’t require the method to be very robust or maintain great performance, but seeing how it fares in such assumption violations can be useful to guide practitioners.

Some minor questions and requests (none of them is too critical):
- Performance gap between CDVAE and the baselines: This gap seems to be much smaller in the semi-synthetic data (Figure 4) as opposed to synthetic data (Figure 3). Do you have an interpretation for this?
- How sensitive is your method to hyperparameters? I see from the tables in the appendix that grid search for each baseline and CDVAE seems comparable, so it’s a fair comparison. That being said, it’d be nice to see some interpretation on the effect of the grid search, e.g., is there a big drop in the performance when using sub-optimal hyperparameters?
- Some of the tables are too small, almost impossible to read without zooming (e.g., Table 3). I suggest enlarging such tables.

**Strengths And Weaknesses:**

Strengths:
- The paper investigates its key problem in a mostly comprehensive way: Clear writing, fair comparisons with respect to the existing literature and good explanations on how the considered setting differs from those.
- The key points and possible question marks about the framework are answered preemptively, with proper justifications and theoretical insights. For instance, Section 4.2 clearly explains how the proposed approach handles the common issues (e.g., selection bias and imbalance), Theorem 9 (realization invariant causal consistency) in Section 4.3. I haven’t gone through the proofs in the Appendix, but the treatment in the main paper seems solid to me.
- Albeit in a limited number of settings, CDVAE outperforms the baselines comfortably. Ablation studies also support the usefulness of each sub-component of the method.


Weaknesses:
- Uncounfedness is a significant assumption (even though it’s understandable). Exploring the sensitivity of the method to violations of this assumption via experiments is a missed opportunity.
- Besides the first point, I don’t see another striking weakness of the paper. Though I should admit that I’m not too familiar with the recent literature on this subject, so it’s not likely, but possible that I am missing some points.

---

> ### Author Response · Authors · 2025-08-12
>
> ## Challenging Sequential ignorability
> Thank you very much for this question. For synthetic data, we perform sensitivity analysis in the following way:  We extend the simulator by partitioning covariates $\mathbf{X}\_t = [\mathbf{X}\_t^o, \mathbf{X}\_t^h]$ into observed ones $\mathbf{X}\_t^o$ and hidden ones $\mathbf{X}\_t^h$ and summarizing the contribution of $\mathbf{X}\_t^h$ into a hidden score. Sensitivity to unobserved confounding is controlled on the selection side by adding $\log(\Gamma) S\_t$ to the treatment logit (Rosenbaum-$\Gamma$ style  (Rosenbaum \& Rubin, 1983)), and on the outcome side by arm-specific loadings $\beta\_{\omega} S\_t$ in the potential-outcome equations. Setting $\gamma = \beta\_0 = \beta\_1 = 0$ recovers the baseline DGP. Varying $\gamma$ yields families of datasets with controlled degrees of hidden selection bias and hidden effect modification by unobserved confounder (cf. Appendix C.1.2). Figure 6 shows how CDVAE, along with baselines in the "base" approach, degrades in performance (PEHE) as confounding level $\Gamma$ increases in $\{1,1.5, \dots, 5\}$. CDVAE still provides globally lower PEHE, but the performance margin narrows as $\Gamma$ increases.
>
> Furthermore, as a falsifiability test, we mask four confounders from the mimic experiment during model training, which are sodium, Glasgow Coma Scale total, cholesterol, and hemoglobin. We repeat the same experimental protocol as in the experiment of Figure 4 and show in Figure 7 the results of all models under the three configurations (base, substitute, and oracle). As expected, PEHE increases for all models across the three configurations, with an average increase of 2 points in PEHE, which is at least an 11\% increase. CDVAE still outperforms baseline in base and substitutes configurations; however, baselines provided with true adjustment variables either provide comparable performances to CDVAE or slightly outperform (eg, CRN).
>
> ## Performance gap between CDVAE and baselines: from synthetic to semi-synthetic MIMIC III data.
> Thank you for raising this concern. In the semi-synthetic MIMIC-III, we masked age, sex, and ethnicity. Compared to synthetic data, the performance gap has relatively shrunk because of a partial proxy leakage: in MIMIC-III, vital signs likely provide **noisy proxies** for masked demographics: prior work shows that models trained on vital signs alone can recover self-reported race, and that pulse-oximetry exhibits race-dependent measurement bias. Coupled with well-established age/sex differences in cardiovascular vitals and age-dependent lactate patterns in ICU cohorts [Avram2019, Sjoding2020, Cifkova2022, He2022, Velichkovska2022]. Baselines that see the full covariate vector therefore get *noisy proxies* for $\mathbf U$; in the fully synthetic data, the vitals were explicitly *uncorrelated* with $\mathbf U$, so this shortcut was not possible for baselines in the synthetic data, and hence the performance gap was much greater.
>
> ## Is CDVAE sensitive to architecture hyperparameters?
> As one concrete check, we varied one of the most important hyperparameters, which is the dimension of the learned substitutes $\mathbf Z$, $d_z$. One-step PEHE stayed in a tight band **1.05–1.30** with no monotone trend. Once $d_z$ is large enough to capture the intrinsic heterogeneity, performance plateaus; oversizing adds mild variance, undersizing adds mild bias, but there’s no catastrophic drop. Practically, our grid search with validation MSE and early stopping reliably lands in this flat region. Hence, CDVAE is not hypersensitive to hyperparameters.
>
> | $d_z$  | 12         | 14         | 16         | 18         | 20         | 22         |
> |--------|------------|------------|------------|------------|------------|------------|
> | CDVAE  | 1.09 ± 0.09 | 1.05 ± 0.19 | 1.21 ± 0.07 | 1.13 ± 0.06 | 1.30 ± 0.20 | 1.11 ± 0.07 |
>
> ## Table sizes
> Thank you for this remark. We updated the size of all figures and tables.

---

### Review · Reviewer_axn3 · 2025-08-04

**Summary Of Contributions:**

This paper proposes the Causal Dynamic Variational Autoencoder (CDVAE) for estimating Individual Treatment Effects (ITEs) in longitudinal data with unobserved adjustment variables. The key innovation is distinguishing between unobserved adjustment variables (affecting only outcomes) and traditional confounders (affecting both treatment and outcomes). CDVAE leverages a Dynamic VAE framework operating in a "near-deterministic regime" to learn latent representations of these adjustment variables, termed "substitutes."

The paper made several key contributions as follows:
(1) A principled approach to infer latent adjustment variables through Conditional Markov Models (CMM);
(2) Theoretical validity proofs ensuring treatment effect identifiability when conditioning on learned substitutes;
(3) Novel exploration of near-deterministic VAE regimes for causal inference, proving treatment effect consistency regardless of specific latent realizations;
(4) Empirical demonstrations showing CDVAE outperforms baselines and enables near-oracle performance when augmenting existing methods with learned substitutes;
(5) Comprehensive theoretical framework including generalization bounds and weighted risk minimization for covariate balance.

**Audience:**

Yes

**Claims And Evidence:**

Yes

**Requested Changes:**

There are a few aspects that might benefit from clarification as follows:
- Regarding the Adversarial Setting, could you provide more intuitive explanation of the adversarial training dynamics? The paper mentions simultaneously minimizing L_tot with respect to {θ, φ, Φ, σ} while minimizing L_W with respect to θ_ω, but the adversarial relationship isn't entirely clear. Is the adversarial aspect primarily about learning representations that predict outcomes well while being balanced across treatment groups? How does this differ from standard domain adaptation, and what prevents the representation learner from simply ignoring the treatment classifier?
- What are the primary failure modes or limitations of CDVAE? Under what conditions might the CMM(p) assumption be violated, and how would this manifest in practice? How sensitive is the method to the choice of the finite order p in the Conditional Markov Model?
- Additionally, could you elaborate on computational scalability  - how does CDVAE performance degrade with longer time series or higher-dimensional covariate spaces?

Some other minor questions:
- How does the choice of overlap weights affect performance compared to other weighting schemes like entropy weights or ATE weights?
- Could you provide more guidance on hyperparameter selection, particularly for λ_IPM and λ_DistM, in real-world applications where oracle performance is unknown?

**Strengths And Weaknesses:**

Some of the key strong points of the paper are as follows:
1. The paper provides rigorous theoretical grounding through the Conditional Markov Model assumption and proves that learned substitutes satisfy sequential ignorability and enable ACATE identifiability. The near-deterministic regime analysis is particularly practical, addressing a fundamental gap in existing deconfounder theory where probabilistic inference models conflict with deterministic theoretical assumptions.
2. CDVAE integrates multiple sophisticated components: weighted empirical risk minimization with overlap weights for selection bias, Gaussian Mixture Model priors for population structure, adversarial training for representation balance, and distribution matching penalties for latent variable stationarity. The technical handling of discrete latents through Bayes-optimal posteriors demonstrates computational awareness. The experimental design thoughtfully adapts baselines to ensure fair comparison while maintaining parameter complexity balance.
3. The experimental results are strong and demonstrate CDVAE's effectiveness across synthetic and semi-synthetic MIMIC-III datasets. Most importantly, the "substitute approach" shows that existing state-of-the-art methods achieve near-oracle performance when augmented with CDVAE's learned substitutes. This transferability represents significant practical value, as practitioners can enhance existing methods without complete framework replacement.

Given these strengths and the key contributions towards causal learning, there are a few ways the paper can be improved upon as follows:
1. The abstract and introduction could better motivate the specific focus on adjustment variables versus confounders. The distinction is crucial but may not be immediately clear to readers familiar with traditional causal inference literature. The introduction would benefit from concrete examples illustrating scenarios where adjustment variables are present but confounders are adequately observed, and why this setting is practically relevant across different domains.
2. The paper lacks detailed computational complexity analysis, which is critical given the sophisticated multi-component architecture. While individual algorithmic details appear in appendices, a comprehensive analysis of training time, memory requirements, and scalability compared to baselines is needed. The adversarial training procedure, IPM computation, and multi-objective optimization likely impose significant computational overhead that should be characterized.
3. The paper would benefit from more thorough discussion of when CDVAE might fail or perform poorly. The CMM assumption, while theoretically justified, may be restrictive in practice. The method's sensitivity to hyperparameter selection (λ_IPM, λ_DistM) and robustness to violation of the adjustment variable assumption deserve more attention. The connection to critiques of deconfounder theory suggests potential vulnerability to "bad variables" that could bias estimates.

---

> ### Author Response · Authors · 2025-08-12
>
> Dear reviewer axn3, we would like to first thank you a lot for your detailed and careful review of our work, and for your time and interest in our submission. Please find below our answers to your requests.
>
>
> ## Clarifying the difference between adjustment variables and confounders.
> Thanks for your suggestion. We have included a new paragraph titled "**Adjustment Variables vs. Confounders**" right after the paragraph "**Our Focus**". We included examples from healthcare, education, and online experimentation.
>
> The paragraph reads as follows: **Adjustment Variables vs. Confounders.** We distinguish confounders, pre-treatment variables that affect both treatment and outcome, whose omission biases estimates, from adjustment variables, pre-treatment variables that affect the outcome and may modify treatment effects but not treatment assignment (Hernán MA, 2020). Adjusting for the latter clarifies further treatment-effect heterogeneity, even when confounding variables are fully observed. For example, in healthcare, pharmacogenetic variants (VKORC1, CYP2C9) markedly alter warfarin (a blood thinner) response yet typically do not determine treatment assignment. Modeling these variants (observed or represented) improves precision and personalization of effects (McClain et al., 2008; Schwarz et al., 2008). In education, Baseline test scores and socioeconomic indicators are important adjustment variables and often are effect modifiers. Covariate adjustment in school-level RCTs thus substantially increases precision and clarifies subgroup treatment effects (Bloom et al., 2007; Steingrimsson et al., 2017). In online experiments/ marketing, pre-period behavior (preceding the A/B test) consists of important adjustment variables that do not affect the randomized assignment, yet adjusting for them leads to faster detection of heterogeneity in treatment effect and even a reduction of the required sample size to achieve a given statistical power (Deng et al., 2013; Benkeser et al., 2021).
>
> ## On failure modes of CDVAE
> Thank you for this very important remark. We summarize the failure mode in the following:
>
> **Hidden confounding beyond $H_t$.**
>
> Our theory assumes that, after conditioning on the observed history $H_t$, there are no unobserved *confounders* — only unobserved *adjustment variables* $\mathbf{U}$ that affect outcomes but not treatment. If unobserved confounders exist, the guarantees no longer hold. In a knife-edge case where those unobserved confounders are (i) static and (ii) satisfy exactly the same CMM(p) property as our adjustment variables, then the population statements (e.g., Theorem 4 and its consequences, Corollary 5 / Theorem 5) would still hold. Yet, two practical obstacles remain: it is unlikely that hidden confounders obey the *same* CMM(p) as $\mathbf{U}$; and, our propensity model is *not* augmented with the learned latent $\mathbf{Z}$. Thus, treatment weighting would still be biased in estimation.
>
> **Time variation in the unobserved variables.**
>
> CDVAE treats $\mathbf{U}$ as *static* adjustment variables. If the true unobserved effect modifiers are time-varying, or if there are time-varying unobserved confounders, the “static-$\mathbf{U}$” assumption is violated. In that case, the latent substitute $\mathbf{Z}$ can no longer absorb all the relevant heterogeneity with a time-invariant representation.
>
> **Very long memory or infinite-order dependence.**
>
> Our identifiability results rely on a *finite* conditional Markov order $p$: the effect of the distant past on $Y_t$ is mediated by the last $p$ lags together with $H_t$ and $\mathbf{Z}$. If outcomes exhibit very long distributed-lag effects, or more strongly, if there is *no* finite $m$ for which responses more than $m$ steps back are conditionally independent given current history and $\mathbf{Z}$, then the process is effectively infinite-order. This violates $\mathrm{CMM}(p)$ and undermines our ability to infer reliable substitutes for $\mathbf{U}$ from the outcome sequence; sequential ignorability with $\mathbf{Z}$ need not hold, and ACATE becomes unidentifiable under our conditions.
>
> **Outcome support and heavy tails.**
>
> Some proofs assume $\mathcal{Y}$ lies in a Borel subset of a *compact* interval. Heavy-tailed outcomes stretch this regularity condition. Practically, this can often be mitigated by variance-stabilizing transforms or winsorization, but it is a technical limitation worth noting.
>
> **Parameter nonstationarity/regime switching.**
>
> Even if a conditional Markov model holds, our analysis presumes fixed parameters $\theta$. If the conditional response dynamics switch across regimes so that, *even given* $\mathbf{Z}$, the effective parameters drift over time, the fixed-$\theta$ factorization used in the arguments breaks. A regime-switching or time-varying parameter extension would be needed; otherwise, causal validity and estimation stability can degrade.

---

> > ### Author Response · Authors · 2025-08-12
> >
> > ## Computational complexity analysis of CDVAE
> > We are happy to include both a theoretical and a practical computational complexity analysis. The training complexity of CDVAE can be summarized in the following proposition.
> >
> > **Proposition 1 (per-epoch complexity).**
> >
> > For a mini-batch of size $b$, sequence length $T$, raw covariate dimension $d_x$, GRU hidden size $H$, and fixed representation width $d_\phi (= 16)$, and let fix $\epsilon$ as the entropic regularization strength in the Sinkhorn algorithm for the IPM term. One training epoch of CDVAE requires
> >
> > $$ \mathcal{O}\\left(nT\\,H(H+d_x)\\right) +  \mathcal{O}\\left(n\\,d_x\\,d_\\phi\\right) +  \mathcal{O}\\left(m\\,b^{2}\\log b/\\varepsilon^{2}\\right)$$
> >
> > FLOPs and stores $O(T\,H+n\,b)$ activations, where
> > - $\mathcal{O} \\left(nT\\,H(H+d_x)\\right)$ — GRU forward/back-prop.
> > - $\mathcal{O}\\left(n\\,d_x\\,d_\\phi\\right)$ — $\\Phi$-projection.
> > - $\mathcal{O}\\left(m\\,b^{2}\\log b/\\varepsilon^{2}\\right)$ — Sinkhorn IPM, with $m=\\lceil 0.1T\\rceil$ and entropic regularization $\\varepsilon$.
> >
> > The first two terms are *linear in the sequence length* $T$ and *(near-)linear in the covariate dimension* $d_x$ because the GRU unrolls once per time-step and the projection layer is dense. The third term is the entropically regularized Sinkhorn solver, whose cost is *quadratic only in the batch size* $b$ and *independent of $d_x$* once the feature map has been compressed to $d_\phi$.
> >
> > Since we evaluate the IPM on just $m=\lceil0.1T\rceil$ randomly chosen time points, overall run-time grows linearly with $T$ and almost linearly with $d_x$ for realistic batches ($b \le 1024$); the sole quadratic factor is $b^2$ inside Sinkhorn.
> >
> > For the simulation dataset, we report the following table, which measures the mean clock training time of all models averaged over 10 different seeds (Processing unit is GPU – 1 × NVIDIA Tesla T4).
> >
> > | Model               | Training time (min) |
> > |---------------------|---------------------|
> > | **CDVAE (ours)**    | $35 \pm 3$          |
> > | **Causal CPC**      | $8 \pm 2$           |
> > | **CT**              | $27 \pm 2$          |
> > | **G-Net**           | $12 \pm 2$          |
> > | **CRN**             | $7 \pm 2$           |
> > | **RMSN**            | $6 \pm 2$           |
> >
> > CRN, RMSN, and CPC lack the IPM term and thus train faster, while the Causal Transformer incurs heavier sequence modeling due to transfer blocks. CDVAE’s additional cost buys explicit distributional balancing over $\Phi(\mathbf{H}_t)$, which we found necessary for stable ACATE estimation; the random-time subsampling of representations keeps the overhead manageable in practice.

---

> ### Author Response · Authors · 2025-08-12
>
> ## On adversarial training of CDVAE
> First of all, the adversarial design of optimization was not intended on its own. It first came from the following observations: the generalization bound and discussion on stationarity of inferred substitutes led to the loss
>
> $$ \\mathcal{L}\_{\\mathrm{tot}}(\\theta, \\theta\_{\\omega}, \\phi, \\Phi, \\sigma) = -\\mathrm{ELBO}(\\theta, \\theta\_{\\omega}, \\phi, \\Phi, \\sigma) + \\lambda\_{\\mathrm{IPM}}\\,\\mathcal{L}\_{\\mathrm{IPM}}(\\theta\_{\\omega}, \\Phi) + \\lambda\_{\\mathrm{DistM}}\\,\\mathcal{L}\_{\\mathrm{DistM}}(\\phi) $$
>
>
>
> The ELBO depends on $\theta_\omega$, which parametrizes the propensity score $e\_{\theta\_{\omega}}(\Phi(\mathbf{h}_t)) = p\_{\theta\_{\omega}}(W_t = 1 \mid \Phi(\mathbf{h}_t))$. ${\theta\_{\omega}}$ are fitted by minimizing the binary cross-entropy $\mathcal{L}\_{W}(\theta\_{\omega}, \Phi)$. Now, the most direct way would be to jointly fit $(\theta\_{\omega}, \Phi)$ with respect to the loss
>
> $$
> \mathcal{L}\_{\mathrm{tot}}(\theta, \theta\_{\omega}, \phi, \Phi, \sigma) + \mathcal{L}\_{W}(\theta\_{\omega}, \Phi).
> $$
>
> However, doing so raises many issues:
>
> 1. $\mathcal{L}\_{W}$ is computed over all observed time steps during training, similar to reconstruction loss in ELBO, but it has a very different scale from the rest of the terms in $\mathcal{L}\_{\mathrm{tot}}$. Coming up with a weighting scheme is difficult because it conflicts with sensitive hyperparameters in our modeling $\lambda\_{\mathrm{IPM}}, \lambda\_{\mathrm{DistM}}$. Even doing a warm-up phase where representation $\Phi(\cdot)$ is first fitted to be predictive of treatment using only $\mathcal{L}\_{W}(\theta\_{\omega}, \Phi)$ did not provide an improvement.
>
> 2. From a standard perspective, propensity score is computed conditioning on $\Phi(\mathbf{h}\_t)$, therefore a more correct way is to fit $\theta_\omega$ given a fixed $\Phi$ in order to extract any given signal from that particular $\Phi$. However since $\Phi$ is neither known nor fixed, a possible strategy to mimic this behavior is to *alternate* between an update of $\Phi$ and that of propensity parameters $\theta_\omega$ given a fixed $\Phi$.
>
> This led to
>
> $$
> \begin{cases}
> \min\_{\theta, \phi, \Phi, \sigma} \mathcal{L}\_{\mathrm{tot}}(\theta, \theta\_{\omega}, \phi, \Phi, \sigma), \\
> \min\_{\theta\_{\omega}} \mathcal{L}\_{W}(\theta\_{\omega}, \Phi).
> \end{cases}
> $$
>
> This multiobjective problem represents an adversarial problem because we have a Representation/outcome player $(\theta, \phi, \Phi, \sigma)$ whose goal pushes $\Phi$ toward balanced (treatment-invariant) features while the ELBO keeps $\Phi$ predictive of outcomes, and a Treatment player minimizes (binary cross-entropy) to predict treatment from the current representation. This is a domain classifier (treated vs. control) in the domain-adversarial training sense of [Ganin et al., 2016]. If $\Phi$ hides treatment signal well, $\mathcal{L}\_{W}$ rises; the classifier then adapts to any residual signal. The seminal work of [Ganin et al., 2016] uses a domain classifier (with gradient reversal) to obtain transfer across source/target domains. Instead of gradient reversal, the generalization bound provides us with an IPM term between treated and non-treated domains. At the same time, maximizing the conditional likelihood of treatment is necessary for the weighted ELBO. This creates a tug-of-war.
>
> Why can the representation not ignore the treatment classifier? Making $\Phi$ constant would drive IPM to 0 but would harm the ELBO (poor outcome fit). Even if $\Phi$ is not constant but overly balanced in the sense that IPM = 0, as pointed out in [Johansson et al., 2019; Zhang et al., 2020], this is likely harmful because we may lose confounding information in the space induced by $\Phi$, which itself will lead to poor outcome fit in ELBO. Secondly, the alternating optimization means that the treatment classifier cannot be ignored because the treatment classifier is re-optimized each round to pick up *any* residual treatment signal in $\Phi$. If $\Phi$ leaves treatment cues, $\mathcal{L}\_{W}$ drops; on the next step when we minimize $\mathcal{L}\_{\mathrm{tot}}$, $\Phi$ is pushed to erase them again (adversarial loop).

---

> ### Author Response · Authors · 2025-08-12
>
> **How CDVAE handles autoregressive order in practice.**
> A finite order in CMM is needed in theory to prove identifiability of ACATE when using the latent substitute as well as their causal validity (Corollary 5 and Theorem 6). In implementation we model $p_{\theta}(y_{t} \mid y_{<t}, \mathbf{x}\_{\leq t}, \omega_{\leq t}, \mathbf{z})$ with a GRU over $[ y_{<t}, \mathbf{x}\_{\leq t}, \omega_{\leq t}]$ and we concatenate its final state with the sampled $\mathbf{z}$. We do not fix a lag, instead, we rely implicitly on the following two mechanisms:
>
> 1. Recurrent predictors and fading memory filters exponentially down-weight distant history. Classical results show that fading memory maps can be uniformly approximated with finite memory models (finite orders). This means that for a given error $\epsilon$ there exists an order $p_\epsilon$ such that conditioning on the last lags is $\epsilon$-close to conditioning on the full history [Funahashi and Nakamura, 1993; Miller and Hardt, 2018; Gonon and Ortega, 2021].
> 2. Training with backpropagation through time biases RNNs toward short/medium context lengths, with negligible effect of considering additional lags [Khandelwal et al., 2018].
>
> To empirically substantiate these claims, we run the following diagnostic: we train CDVAE over datasets generated autoregressively on the outcome with $p_{\mathrm{true}} = 10$. Over the test data, we compute the counterfactual responses while masking entries in the outcome sequence to simulate a finite autoregressive order as in $\mathrm{CMM}(p)$. We do so for multiple values $p = 4, 6, 8, 10, 12, 14, 16$. On test data, we compute one-step outcome residuals after conditioning on $\mathbf{H}\_t$ and $\mathbf{Z}$. We run the Ljung–Box test to detect any remaining autocorrelation. Any persistent autocorrelation across the entire range of lags or a steady increase in the accuracy/precision with higher $p$ would indicate a violation of $\mathrm{CMM}(p)$.
>
> In tables, for every masked order $p \in \{4,6,8,10,12,14,16\}$, the Ljung–Box p-values for cumulative lags $h = 1, \dots, 20$ stay well above $0.05$ (e.g., $p=4$: 0.085; $p=10$: 0.109), so we fail to reject residual autocorrelation up to lag 20 at all tested $p$. Meanwhile, one-step RMSE on counterfactuals ($\sim 1.129$–$\sim 1.131$) and PEHE ($\sim 0.975$–$\sim 0.986$) are essentially flat across $p$. Together, this supports that conditioning on a short/medium window already removes serial dependence; increasing $p$ beyond $\sim 4$–6 gives no material gain, consistent with (i) portmanteau tests indicating no remaining autocorrelation in residuals up to $h$ lags, and (ii) theory that fading-memory systems can be uniformly approximated by finite-memory models (and that RNN training tends to weight recent context more heavily).
>
> **Table – One-step accuracy vs. masked autoregressive order $p$**
>
> | $p$ | RMSE$\_{\text{1-step}}$ | PEHE$\_{\text{1-step}}$ |
> |-----|-----------------------|------------------------|
> | 4   | 1.131   | 0.977|
> | 6   | 1.130 | 0.984|
> | 8   | 1.130 | 0.986|
> | 10  | 1.129 | 0.986|
> | 12  | 1.129| 0.982|
> | 14  | 1.130| 0.975|
> | 16  | 1.130| 0.982|
>
> **Table – Residual whiteness vs. masked order $p$ (Ljung–Box, joint up to $h=20$)**
>
> | $p$ | $\min_{k\le 20}$ p-value | # lags $\le 20$ with p $<0.05$ | Reject any? |
> |-----|--------------------------|--------------------------------|-------------|
> | 4   | 0.085                    | 0                              | No|
> | 6   | 0.083                    | 0                              | No|
> | 8   | 0.097                    | 0                              | No|
> | 10  | 0.109                    | 0                              | No|
> | 12  | 0.095                    | 0                              | No|
> | 14  | 0.079                    | 0                              | No          |
> | 16  | 0.111                    | 0                              | No          |
>
> **Table – Ljung–Box statistics (Q) and p-values at selected lags $k \in \{1, 5, 10, 20\}$**
>
> | $p$ | Q (lag 1) | p (lag 1) | Q (lag 5) | p (lag 5) | Q (lag 10) | p (lag 10) | Q (lag 20) | p (lag 20) |
> |-----|-----------|-----------|-----------|-----------|------------|------------|------------|------------|
> | 4   | 2.963     | 0.085     | 7.442     | 0.190     | 10.812     | 0.372      | 12.111     | 0.912      |
> | 6   | 2.996     | 0.083     | 7.442     | 0.190     | 10.725     | 0.379      | 12.067     | 0.914      |
> | 8   | 2.755     | 0.097     | 6.669     | 0.246     |  9.892     | 0.450      | 11.483     | 0.933      |
> | 10  | 2.566     | 0.109     | 7.012     | 0.220     |  9.927     | 0.447      | 11.213     | 0.941      |
> | 12  | 2.788     | 0.095     | 6.825     | 0.234     | 10.289     | 0.416      | 11.653     | 0.927      |
> | 14  | 3.077     | 0.079     | 8.189     | 0.146     | 11.694     | 0.306      | 13.236     | 0.867      |
> | 16  | 2.533     | 0.111     | 7.135     | 0.211     | 10.652     | 0.385      | 11.930     | 0.918      |

---

> ### Author Response · Authors · 2025-08-12
>
> ## Computational Scalability
> We are happy to include additional experiments where we first increase sequence length $T$ from 75 to 175, and in a second one, we vary the dimension of covariates $d_x$ from 100 to 500. Empirically, CDVAE’s one‐step PEHE stays almost flat as the sequence length $T$ more than doubles (75 $\rightarrow$ 175), whereas it grows modestly (approximately $+57\%$) when the raw covariate dimension $d_x$ quintuples (100 $\rightarrow$ 500). This behaviour matches the model’s complexity profile: GRU unrolling makes both run‐time and memory scale linearly in $T$, and the IPM term — computed on a fixed‐size batch of 16‐dimensional representations — costs $\mathcal{O}(b^2)$ in the batch size $b$ but does not grow with $d_x$. Only the first dense projection and the GRU input weights scale with $d_x$, hence the roughly linear slowdown.
>
> **Table – Performance of CDVAE over varying sequence length**
>
> | Sequence length | 75             | 100            | 125            | 150            | 175            |
> |-----------------|----------------|----------------|----------------|----------------|----------------|
> | PEHE            | $1.09 \pm 0.09$ | $0.98 \pm 0.15$ | $1.00 \pm 0.20$ | $0.98 \pm 0.21$ | $0.97 \pm 0.09$ |
> | Wall-clock (min) | $35 \pm 3$     | $57 \pm 3$     | $82 \pm 2$     | $95 \pm 3$     | $105 \pm 3$    |
>
> **Table – Performance of CDVAE over varying $d_{\mathrm{vitals}}$**
>
> | $d_{\mathrm{vitals}}$ | 100             | 200             | 300             | 400             | 500             |
> |-----------------------|-----------------|-----------------|-----------------|-----------------|-----------------|
> | CDVAE (base)          | $1.09 \pm 0.09$ | $1.51 \pm 0.03$ | $1.60 \pm 0.04$ | $1.71 \pm 0.05$ | $1.71 \pm 0.05$ |
> | Wall-clock (min)      | $35 \pm 3$      | $33 \pm 2$      | $34 \pm 2$      | $32 \pm 2$      | $33 \pm 2$      |
>
> ## (Minor) On the Choice of weighting
>
> To examine the effect of different weighting schemes on CDVAE performance, we partially reproduced the missing-confounding sensitivity analysis for $\Gamma = 1, 1.5, 2$. In general, Overlap weighting performs better than IPTW, the entropy method is competitive, but we found it to be less stable in performance (relatively higher standard deviation). We also note that entropy balancing (EB) adds non-trivial computational overhead. EB computes calibration weights by solving a **constrained convex program** that enforces moment balance. Furthermore, while entropy balancing can enforce exact moment balance on the learned representation $\Phi(\mathbf{H}_t)$ via convex calibration weights, it alters the optimization landscape relative to our likelihood–IPM objective and risks redundancy with the Wasserstein/IPM term that already aligns treated–control distributions in representation space.
>
> **Table – Performance of CDVAE under different weighting schemes across $\Gamma$ values**
>
> | **Weighting scheme** | $\Gamma=1$         | $\Gamma=1.5$       | $\Gamma=2$         |
> |----------------------|--------------------|--------------------|--------------------|
> | Overlap              | $1.09 \pm 0.09$    | $1.55 \pm 0.01$    | $1.87 \pm 0.03$    |
> | IPTW                 | $1.10 \pm 0.15$    | $1.65 \pm 0.02$    | $2.00 \pm 0.01$    |
> | Entropy              | $1.07 \pm 0.21$    | $1.43 \pm 0.08$    | $1.97 \pm 0.12$    |
>
> ## Selection of Hyperparameters
> The parameters were not chosen with knowledge of the Oracle performance. In fact, they were chosen as in a realistic setting: by grid search with model selection on factual validation error (MSE) and early stopping. This is standard practice in ITE/counterfactual learning because counterfactual outcomes are unobserved; recent work reports validation on factual error and early stopping for hyperparameter choice (RMSN, CRN, CT, Causal CPC). In practice, if sweeping over the hyperparameters provides indistinguishable validation MSE, we pick the one with the smaller IPM between control and treated representations.
>
> ## References
> [Funahashi and Nakamura, 1993] Funahashi, K.-i. and Nakamura, Y. (1993). Approximation of dynamical systems by continuous-time recurrent neural networks. Neural
> networks.
>
> [Ganin et al., 2016] Ganin, Y., Ustinova, E., Ajakan, H., Germain, P., Larochelle, H., Laviolette, F., Marchand, M., and Lempitsky, V. (2016). Domain-adversarial training of neural networks. The journal of machine learning research.
>
> [Gonon and Ortega, 2021] Gonon, L. and Ortega, J.-P. (2021). Fading memory echo state networks are universal. Neural Networks.
>
> [Khandelwal et al., 2018] Khandelwal, U., He, H., Qi, P., and Jurafsky, D. (2018). Sharp nearby, fuzzy far away: How neural language models use context. In Proceedings of the 56th Annual Meeting of the Association
> for Computational Linguistics.
>
> [Miller and Hardt, 2019] Miller, J. and Hardt, M. (2019). Stable recurrent models. In International Conference on Learning Representations

---

### Author Response · Authors · 2025-10-03
**Follow-up on Author Response**

Dear Reviewers,

We would like to kindly follow up regarding our submission number 4602.

We submitted our detailed responses addressing all reviewer comments on August 12. As the review guidelines mentioned that recommendations were expected by September 1, we wanted to check if you could share your feedback on whether our revisions satisfactorily address your concerns, or if any further clarifications are needed.

We sincerely appreciate your time and effort in reviewing our work.
Best regards,
Authors of TMLR submission 4602.

---

### Decision · Action_Editor_zoBN · 2025-10-04

**Recommendation:** Accept as is

**Additional Comments:**

The paper considers the problem of estimating treatment effect under heterogeneity. Under the assumption of no unobserved confounding, they aim to learn a latent representation for the adjustment variables---variables that only affect the outcome.
All reviewers appreciated the theoretical contributions in establishing identifiability. Its focus on longitudinal data is also welcome, as such data is widely available in practical domains. The study on VAE under near-deterministic regime and showing consistency therein can be of independent interest, especially for causal inference.
Overall, the paper has sufficient technical merit, and also reasonable empirical experiments.

A practical weakness is the assumption of no unobserved confounding, but that applies to most causal inference papers, so this is not a specific weakness for the paper.
I agree with R3 on the use of ITE. For the camera-ready, I recommend that the authors only use the term CATE and avoid use ITE throughout their paper.

**Audience:**

Yes

**Audience Explanation:**

Causal effect inference is a widely studied topic in ML, and this paper's focus on heterogeneous treatment effect will be of interest to many.

**Claims And Evidence:**

Yes

**Claims Explanation:**

The paper supports its claims very well. Its main claim is to account for effect modifiers/adjustment variables (that only affect the outcome) while estimating CATE. It provides sufficient theoretical justification---using a conditional markov process to represent the data and provide a proof for identifiability using their learnt adjustment variables.  Numerical experiments on synthetic and semi-synthetic data support the method's efficacy.